# A system dynamic model to quantify the impacts of water resources allocation on water-energy-food-society (WEFS) nexus

*Yujie Zeng*[1], *Dedi Liu*[1,2*], *Shenglian Guo*[1], *Lihua Xiong*[1], *Pan Liu*[1], *Jiabo Yin*[1,2],

*Zhenhui Wu*[1]

[1] State Key Laboratory of Water Resources and Hydropower Engineering Science,

Wuhan University, Wuhan 430072, China

[2] Hubei Province Key Lab of Water System Science for Sponge City Construction,

Wuhan University, Wuhan 430072, China

*Correspondence to*: Dedi Liu (dediliu@whu.edu.cn)

**Abstract:** Sustainable management of water-energy-food (WEF) nexus remains an urgent challenge, as interactions between WEF and human sensitivity and reservoir operation in the water system are typically neglected. This study proposes a new approach for modeling WEF nexus by incorporating human sensitivity and reservoir operation into the system. The co-evolution behaviors of the nexus across water, energy, food, and society (WEFS) were simulated using the system dynamic model. Reservoir operation was simulated to determine the water supply for energy and food systems by the Interactive River-Aquifer Simulation water resources allocation model. Shortage rates for water, energy, and food resulting from the simulations were used to qualify their impacts on the WEFS nexus through environmental awareness in society system. Human sensitivity indicated by environmental awareness can then adjust the co-evolution behaviors of the WEFS nexus through feedback loops. The proposed approach was applied to the mid-lower reaches of the Hanjiang river basin in China as a case study. Results indicate environmental awareness shows potential to capture human sensitivity to shortages from water, energy, and food systems. Parameters related to boundary conditions and critical values can dominate environmental awareness feedback to regulate socioeconomic expansion to maintain the integrated system from constant resources shortages. The annual average energy shortage rate thereby decreased from 17.16% to 5.80% by taking environmental awareness feedback, contributing to the sustainability of the WEFS nexus. Rational water resources allocation can ensure water supply through reservoir operation. The annual average water shortage rate decreased from 15.89% to 7.20% as water resources

allocation was considered. Threats from water shortage on the concordant
development of the WEFS nexus are significantly alleviated, particularly for the area
with limited regulating capacity of water project. Therefore, this study contributes to
the understanding of interactions across the WEFS systems and helps in improving
the efficiency of resources management.
**Keywords:** water-energy-food-society nexus; system dynamic; water resources
allocation; human sensitivity

## 1. Introduction

Water, energy, and food are indispensable resources for sustainable development
of society. With the growing population, urbanization, globalization, and economic
development, the expected global demands for water, food, and energy in 2030 will
increase by 40%, 50%, and 50%, respectively, compared to the 2010 levels
(Alexandratos and Bruinsma, 2012; Mckinsey & Company, 2009; International
Energy Agency, 2012). Resource scarcity will be exacerbated by the single-sector
strategy in traditional water, energy, and food management (El Gafy et al., 2017). To
increase resource use efficiency and benefits in production and consumption, taking
the inextricable interactions among sectors across water, energy, and food into rational
resources management has become an important strategy (Hsiao et al., 2007;
Vörösmarty et al., 2000). Considering these interactions, the water-energy-food (WEF)
nexus concept was first presented at the Bonn Conference in 2011 as an approach to
determine synergies and trade-offs between WEF sectors to support sustainable
development goals (Hoff, 2011).

Various methods have been proposed for integrated systems to quantify the

interactions in the WEF nexus. There are three main types of methods: system of
systems model (Eusgeld et al., 2011; Housh et al., 2015), agent-based model
(Bonabeau, 2002; Dawson et al., 2011), and system dynamic model (El Gafy, 2014;
Swanson, 2002). The system of systems model comprises several subsystems as a
holistic system to address the nexus by optimizing system behavior. The agent-based
model simulates the interactions between agents and environments as well as different
agents based on predefined rules obtained from long-term observations. These two
methods have been established to be capable of simulating the behaviors of an
integrated system. However, neither of them has emphasized feedback within the
integrated systems, which is considered an important driving force for nexus system
(Chiang et al., 2004; Kleinmuntz, 1993; Makindeodusola and Marino, 1989). The
results of these two methods for WEF security remain at risk. The system dynamic
model explicitly focuses on feedback connections between key elements in a model to
determine the co-evolution process and long-term characteristics of integrated
systems (Liu, 2019; Simonovic, 2002). Therefore, system dynamic model was
adopted in this study to simulate the co-evolution process of the nexus system.

System dynamic model has been widely used to analyze the WEF nexus

worldwide at different spatial scales, such as global (Davies and Simonovic, 2010;
Susnik, 2018), national (Laspidou et al., 2020; Linderhof et al., 2020), and basin-scale
(Purwanto et al., 2021; Ravar et al., 2020). Most of these models perform the

accounting and analysis of the WEF nexus, focusing only on the physical process, while rarely highlighting the social process that indicates human responses to the WEF nexus (Elshafei et al., 2014). As the connection between the WEF nexus and society is intensified under rapid socioeconomic development, both physical and social processes should be considered for the sustainability of the integrated system in the future (Di Baldassarre et al., 2015; Di Baldassarre et al., 2019).

To simultaneously capture the physical and social processes of the integrated system, human sensitivity was considered as a conceptual social state variable to identify environmental deterioration (Elshafei et al., 2014; Van Emmerik et al., 2014). Van Emmerik et al. (2014) developed a socio-hydrologic model to understand the competition for water resources between agricultural development and environmental health in the Murrumbidgee river basin (Australia). Li et al. (2019) developed an urban socio-hydrologic model to investigate future water sustainability from a holistic and dynamic perspective in Beijing (China). Feng et al. (2016) used environmental awareness to indicate community's attitude to influence the co-evolution behaviors of the water-power-environment nexus in the Hehuang region (China). These studies have contributed to effective resources management by incorporating both physical and social processes. However, potential threats to WEF security exist, as few of the current studies have simultaneously considered the impacts of reservoir operation in water system on the integrated system.

Reservoirs can adjust the uneven temporal and spatial distribution of available water resources and can ensure water supply to reduce water shortage (Khare et al.,

2007; Liu et al., 2019; Zeng et al., 2021; He et al., 2022). However, the available
water resources are typically adopted under historical natural water flow scenarios,
while reservoirs are seldom considered, or their operational rules are significantly
simplified in the WEF nexus. The assessment of water supply security based on the
WEF nexus should be improved. Thus, additional details regarding the reservoir
operation should be incorporated into the simulation of the WEF nexus.
The water resources allocation model can simultaneously incorporate reservoir
operation and water acquisition, and it has become an effective tool to quantitatively
assess the impacts of reservoir operation on water supply security, as well as WEF
security (Si et al., 2019; Zhou et al., 2019). Our study aims to establish a system
dynamic model for the water-energy-food-society (WEFS) nexus and assess the
impacts of reservoir operation on the WEFS nexus by integrating the water resources
allocation model into the integrated system. The reminder of this paper is organized as
follows: Section 2 introduces the framework for modeling the WEFS nexus and
assessing the impacts of water resources allocation on the WEFS nexus. Section 3
describes the methodologies applied in the mid-lower reaches of the Hanjiang river
basin in China, which is the study area. Section 4 presents the results of the
co-evolution process and the sensitivity analysis of the WEFS nexus. The impacts of
water resources allocation on the WEFS nexus have also been discussed. The
conclusions of this study are presented in Section 5.

## 2 Methods

System dynamic modeling (SDM) simulates the dynamics among different systems using nonlinear ordinary differential equations and dynamic feedback loops (Wolstenholme and Coyle, 1983; Swanson, 2002). SDM has become an efficient approach to facilitate the integrated analysis of sectors, processes, and interrelations among different system variables (Di Baldassarre et al., 2015; Simonovic, 2002). The SDM for assessing the WEFS nexus comprises four modules (shown in Figure 1): water system module, energy system module, food system module, and society system module.

In the water system module, socioeconomic water demand (i.e., municipal, rural, industrial, and agricultural water demand) and in-stream water demand are projected using the quota method and Tennant method (Tennant, 1976), respectively. The water demands and available water resources are further inputted into the water resources allocation model to determine the water supply and water shortage for every water use sector in each operational zone. The water supply for socioeconomic water use sectors and agricultural water shortage rates as outputs from the water system module are taken as the inputs of the energy system module and food system module to determine the energy consumption and food production, respectively. Considering the outputs of the energy and food system modules, the energy and food shortages can be estimated by comparing the planning energy availability and target food production, respectively. The function of the society module is to capture human sensitivity to

degradation in the WEF nexus (Elshafei et al., 2014). Environmental awareness is
considered as the conceptual social state variable to indicate human sensitivity (Van
Emmerik et al., 2014). Environmental awareness is composed of water shortage
awareness, energy shortage awareness, and food shortage awareness that are
determined by shortages of water, energy, and food, respectively. As environmental
awareness accumulates over its critical value, negative feedback on socioeconomic
sectors (i.e., population, GDP, and crop area) will be triggered to constrain the
increases in water demand, and further energy consumption, and food production to
sustain the WEFS nexus.

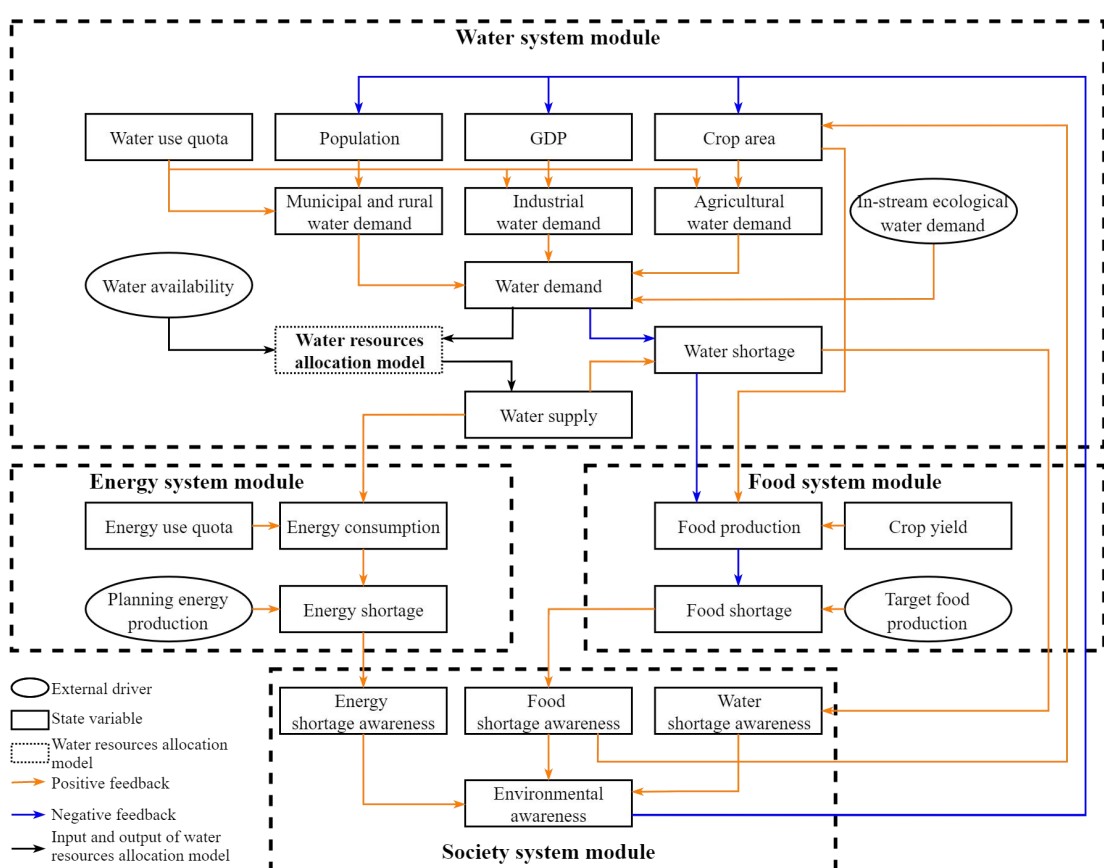

**Figure 1. Structure of WEFS nexus model and its feedbacks.**

## 2.1 Water System Module

### 2.1.1 Water Demand Projection

Water user comprises socioeconomic (also called off-stream) user and in-stream user. Socioeconomic water users can be classified into municipal, rural, industrial, and agricultural sectors. The quota method has been considered an efficient approach to project the annual socioeconomic water demand (Brekke et al., 2002). The amount of water demand for the socioeconomic users can be estimated using equation (1).

$$WD_{i,j}^t = WQ_{i,j}^t * A_{i,j}^t / U_{i,j}^t \tag{1}$$

where $WD_{i,j}^t$ is the amount of water demand for the $j$-th user in the $i$-th operational zone in the $t$-th year; $WQ_{i,j}^t$ denotes the water use quota unit of water user; $A_{i,j}^t$ is the amount of water units of water user; and $U_{i,j}^t$ represents the utilization rate of water user. The water quota units represent the amount of water consumption per capita in municipal and rural users, the amount of water consumption per ten thousand Yuan in industrial user, and the amount of net irrigation water per unit area in agricultural user, respectively. The amount of water units represents the projected population in municipal and rural users, projected GDP in industrial user, and projected irrigated area in agricultural user.

As population, GDP, crop area, and water use quota are prerequisites for water demand projection, the dynamic equations for these socioeconomic variables should be pre-determined. There are two types of methods which are popular in socioeconomic projection, Malthusian model (Bertalanffy, 1976; Malthus, 1798) and

Logistic model (Law et al., 2003), which are adopted for the socioeconomic
projection. The growth rate in original Malthusian model is constant (Malthus, 1798),
which is not consistent with previous studies that the socioeconomic expansion in the
future would slow down (He et al., 2017; Lin et al., 2016). Therefore, we used
exponential terms to simulate the evolution of socioeconomic variables, which
increases with decreasing rate. And feedback functions, as well as environmental
carrying capacities (indicating the maximum socioeconomic size that can be carried
by the system) of socioeconomic variables are adopted to constrain the evolution of
these socioeconomic variables through equations (2)–(4) (Feng et al., 2016;
Hritonenko and Yatsenko, 1999). Socioeconomic factors in original Logistic model
(Law et al., 2003) are prone to approach to their environmental carrying capacities,
while the constrains among subsystems in WEFS nexus are typically neglected, which
will lead over-sized socioeconomic projection. Therefore, feedback functions taken as
constraints from subsystems are adopted in equation (5)–(7) (Li et al., 2019; Wu et al.,

2022).


$$
\begin{cases}
\dfrac{dN_t}{dt} = r_{P,t} * N_t \\[2mm]
r_{P,t} = \begin{cases}
r_{P,0} * \kappa_P * \exp(-\varphi_P t) + f_1(E) & N_t \le N_{cap} \\
\mathrm{Min}(0, r_{P,0} * \kappa_P * \exp(-\varphi_P t) + f_1(E)) & N_t > N_{cap}
\end{cases}
\end{cases}
\tag{2}
$$


$$
\begin{cases}
\dfrac{dG_t}{dt} = r_{G,t} * G_t \\[2mm]
r_{G,t} = \begin{cases}
r_{G,0} * \kappa_G * \exp(-\varphi_G t) + f_2(E) & G_t \le G_{cap} \\
\mathrm{Min}(0, r_{G,0} * \kappa_G * \exp(-\varphi_G t) + f_2(E)) & G_t > G_{cap}
\end{cases}
\end{cases}
\tag{3}
$$


$$\begin{cases} \dfrac{dCA_t}{dt} = r_{CA,t} * CA_t \\ r_{CA,t} = \begin{cases} r_{CA,0} * \kappa_{CA} * \exp(-\varphi_{CA}t) + f_3(E, FA) & CA_t \le CA_{cap} \\ \text{Min}(0, r_{CA,0} * \kappa_{CA} * \exp(-\varphi_{CA}t) + f_3(E, FA)) & CA_t > CA_{cap} \end{cases} \end{cases} \tag{4}$$


$$\frac{dN_t}{dt} = N_t * (r_{P,0} * (1 - \frac{N_t}{N_{cap}}) + f_1(E)) \tag{5}$$


$$\frac{dG_t}{dt} = G_t * (r_{G,0} * (1 - \frac{G_t}{G_{cap}}) + f_2(E)) \tag{6}$$


$$\frac{dCA_t}{dt} = CA_t * (r_{CA,0} * (1 - \frac{CA_t}{CA_{cap}}) + f_3(E, FA)) \tag{7}$$

where $N_t$, $G_t$, and $CA_t$ are the population, GDP, and crop area in the $t$-th year,
respectively; $N_{cap}$, $G_{cap}$, and $CA_{cap}$ denote the environmental carrying capacities of
population, GDP, and crop area, respectively; $r_{P,0}$, $r_{G,0}$, and $r_{CA,0}$ represent the growth
rates of population, GDP, and crop area from historical observed data, respectively; $r_{P,t}$
$r_{G,t}$, and $r_{CA,t}$ are the growth rates of population, GDP, and crop area in the $t$-th year,
respectively; $\kappa_P*\exp(-\varphi_P t)$, $\kappa_G*\exp(-\varphi_G t)$, and $\kappa_{CA}*\exp(-\varphi_{CA}t)$ are used to depict the
impacts of social development on the evolution of population, GDP, and crop area,
respectively; $E$ is environmental awareness; $FA$ is food shortage awareness; and $f_1$, $f_2$,
and $f_3$ represent the feedback functions. The equations for $E$, $FA$, and feedback
functions are described in detail in Sections 2.4 and 2.5.
Water use quotas are also assumed to decrease with the social development
owing to the expansion economy (Blanke et al., 2007; Hsiao et al., 2007). As the
difficulties in saving water by technological advancement are increasing, the changing
rate of water use quota is decreasing in equation (8) (Feng et al., 2019).
$$\begin{cases} \dfrac{dWQ_{i,j}^{t}}{dt} = WQ_{i,j}^{t} * r_{qwu,t} \\ r_{qwu,t} = \begin{cases} r_{qwu,0} * \kappa_{qwu} * \exp(-\varphi_{qwu} t) & WQ_{i,j}^{t} > WQ_{i,j}^{min} \\ 0 & \text{else} \end{cases} \end{cases} \tag{8}$$

211 where $WQ_{i,j}^{t}$ denotes the water use quota of the *j*-th water user in the *i*-th operational

212 zone in the *t*-th year; $r_{qwu,0}$ and $r_{qwu,t}$ are the growth rates of water use quotas from

213 historical observed data and *t*-th year, respectively; $WQ_{i,j}^{min}$ is the minimum value of

214 water use quotas; and $\kappa_{qwu}*\exp(-\varphi_{qwu}t)$ is used to depict the water-saving effect of

215 social development on the evolution of water use quota.

216 **2.1.2 Water Resources Allocation**

217  Based on water availability and projected water demand, available water

218 resources can be deployed to every water use sector and in-stream water flows using a

219 water resources allocation model. The Interactive River-Aquifer Simulation (IRAS)

220 model is a rule-based water system simulation model developed by Cornell University

221 (Loucks, 2002; Zeng et al., 2021; Matrosov et al., 2011). The year is divided into

222 user-defined time step, and each time step is broken into user-defined sub-time step,

223 based on which water resources allocation conducts. The IRAS model was adopted

224 for water resources allocation owing to its flexibility and accuracy in water system

225 simulations.

226  As water system comprises water transfer, consumption, and loss components, it

227 is typically delineated by node network topology for the application of the water

228 resources allocation model. Reservoir nodes and demand nodes are the most

229 important elements in the node network topology, as they directly correspond to the

processes of water supply, acquisition, and consumption. The water shortage at the
demand node should first be determined based on its water demand and total water
supply. The total water supply comprises natural water inflow (i.e., local water
availability) and water supply from reservoir. In each sub-time step (except the first),
the average natural water inflow in the previous *sts*-1 sub-time steps is estimated as
the projected natural water inflow in the remaining sub-time steps using equation (9).
The water shortage can then be determined by deducting the demand reduction, total
real-time water inflow, and projected natural water inflow from water demand using
equation (10). The total water shortage rate can then be determined using equation

(11).

$$WE_{i,j}^{sts} = (\sum_{1}^{sts-1} WTSup_{i,j}^{sts} - \sum_{1}^{sts-1} WRSup_{i,j}^{sts}) * \frac{(Tsts - sts + 1)}{(sts - 1)} \tag{9}$$

$$WS_{i,j}^{sts} = \frac{WD_{i,j}^{ts}(1 - f_{red}) - \sum_{1}^{sts} WTSup_{in}^{sts} - WE_{i,j}^{sts}}{Tsts - sts + 1} \tag{10}$$

$$WSR_t = \frac{\sum_{i,j} WSR_{i,j}^t}{f_{red} * \sum_{i,j} WD_{i,j}^t} = \frac{\sum_{i,j} \sum_{ts} \sum_{sts} WS_{i,j}^{sts}}{f_{red} * \sum_{i,j} \sum_{ts} WD_{i,j}^{ts}} \tag{11}$$

where *ts* is the current time step; *Tsts* denotes the total number of the sub-time steps;
*sts* is the current sub-time step; $WE_{i,j}^{sts}$ represents the projected natural water inflow
for the *j*-th water use sector in the *i*-th operational zone; $WTSup_{i,j}^{sts}$ is the total water
supply; $WRSup_{i,j}^{sts}$ is the water supply from reservoir; $WD_{i,j}^{ts}$ is the water demand; $f_{red}$
is the demand reduction factor; $WS_{i,j}^{st}$ is the water shortage; $WSR_{i,j}^t$ is the water
shortage rate in the *t*-th year; and $WSR_t$ is the total water shortage rate.

The water shortage at the demand node requires water release from the

corresponding reservoir nodes according to their hydrological connections. The
amount of water released from the reservoir depends on the water availability for
demand-driven reservoirs and operational rules for supply-driven reservoirs,
respectively. The water release for the supply-driven reservoir is linearly interpolated
based on Figure 2 and equations (12)–(18). Additional details on the IRAS model can
be found in Matrosov et al. (2011).

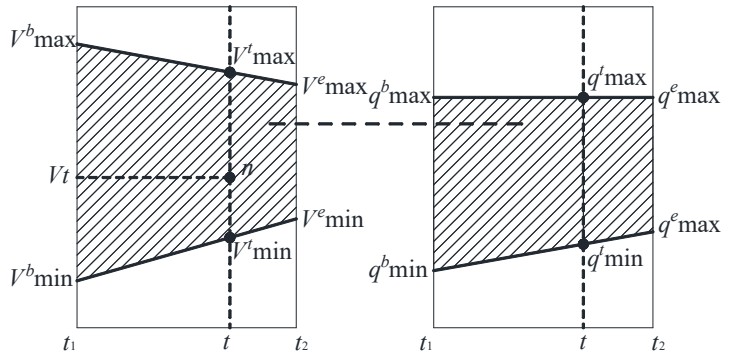

**Figure 2. Water release rule for supply-driven reservoir.**

$$P_t = (t - t_1)/(t_2 - t_1) \tag{12}$$

$$V_{max}^t = V_{max}^b * (1 - P_t) + V_{max}^e * P_t \tag{13}$$

$$V_{min}^t = V_{min}^b * (1 - P_t) + V_{min}^e * P_t \tag{14}$$

$$q_{max}^t = q_{max}^b * (1 - P_t) + q_{max}^e * P_t \tag{15}$$

$$q_{min}^t = q_{min}^b * (1 - P_t) + q_{min}^e * P_t \tag{16}$$

$$P_v = (V^t - V_{min}^t)/(V_{max}^t - V_{min}^t) \tag{17}$$

$$q^t = q_{min}^t * (1 - P_v) + q_{max}^t * P_v \tag{18}$$

where $t$, $t_1$, and $t_2$ are the current time, initial time, and end time in the period,
respectively; $P_t$ denotes the ratio of current time length to period length; $V_{max}^t$, $V_{min}^t$,
$V_{max}^b$, $V_{min}^b$, $V_{max}^e$, and $V_{min}^e$ represent the maximum and minimum storages at the
current time, beginning, and ending of the period, respectively; $q_{max}^t$, $q_{min}^t$, $q_{max}^b$,
$q_{\min}^{b}$,   $q_{\max}^{e}$, and   $q_{\min}^{e}$   denote the maximum and minimum releases, respectively; $P_v$
is the ratio of current storage; and $q_t$ is the current release.

## 2.2 Energy System Module

The energy system module focuses on the energy consumption during the water
supply process for socioeconomic water users to further investigate the energy
co-benefits of water resources allocation schemes (Zhao et al., 2020; Smith et al.,
2016). Energy consumption for water heating and water end-use was not included in
this study. Energy consumption is determined by the energy use quota and amount of
water supply for the water use sectors (Smith et al., 2016). As energy use efficiency
will be gradually improved with social development, the energy use quota is assumed
to decrease with decreasing rate. The trajectory of the energy use is formulated in
equation (19). The water supply for water use sectors derived from the water system
module is used to estimate energy consumption using equation (20). The energy
shortage rate will be further determined with planning energy availability using
equation (21).
$$\begin{cases} \dfrac{dEQ_{i,j}^{t}}{dt} = EQ_{i,j}^{t} * r_{e,t} \\ r_{e,t} = \begin{cases} r_{e,0} * \kappa_e * \exp(-\varphi_e t) & EQ_{i,j}^{t} > EQ_{i,j}^{min} \\ 0 & else \end{cases} \end{cases} \tag{19}$$

$$EC_t = \sum_{i,j} WTSup_{i,j}^{t} * EQ_{i,j}^{t} \tag{20}$$

$$ESR_t = \frac{ES_t}{EC_t} = \frac{EC_t - PEA_t}{EC_t} \tag{21}$$

where   $EQ_{i,j}^{t}$ is the energy use quotas of the $j$-th water user in the $i$-th operational zone
in the $t$-th year; $r_{e,0}$ and $r_{e,t}$ denote the growth rates of energy use quotas from
historical observed data and the $t$-th year, respectively; $EQ_{i,j}^{min}$ is the minimum value
of energy use quotas; $\kappa_e * \exp(-\varphi_e t)$ depicts the energy-saving effect of social
development; $EC_t$ is the total energy consumption; $WTSup_{i,j}^t$ is the total water
supply of the $j$-th water user in the $i$-th operational zone; $ES_t$ and $ESR_t$ are the
energy shortage and energy shortage rate, respectively; and $PEA_t$ is the planning
energy availability.

**2.3 Food System Module**

The food system module focuses on estimating the amount of food production.
As water is a crucial determinant for crop yield, the agricultural water shortage rate
can constrain the potential crop yield (French and Schultz, 1984; Lobell et al., 2009).
Owing to the technological advancements in irrigation, the amount of potential crop
yield is assumed to increase with decreasing rate, as indicated by equation (22). With
the target food production which has considered the local and exported food demands
of basin, the food shortage rate can then be estimated using equations (23) and (24).
$$\begin{cases} \dfrac{dCY_{i,j}^t}{dt} = CY_{i,j}^t * r_{pro,t} \\ r_{pro,t} = r_{pro,0} * \kappa_{pro} \exp(-\varphi_{pro} t) \end{cases} \tag{22}$$

$$FP_t = \sum_{i,j} CY_{i,j}^t * CA_{i,j}^t * (1 - WSR_{i,4}^t) \tag{23}$$

$$FSR_t = \frac{FS_t}{TFP_t} = \frac{TFP_t - FP_t}{TFP_t} \tag{24}$$

where $CY_{i,j}^t$ is the potential crop yields of the $j$-th crop in the $i$-th operational zone in
the $t$-th year; $r_{pro,\,0}$ and $r_{pro,\,t}$ are the growth rates of crop yields from historical
observed data and the $t$-th year, respectively; $\kappa_{pro} * \exp(-\varphi_{pro} t)$ depicts the impacts of
social development on the evolution of crop yield; $FP_t$ denotes the total food
production; $CA_{i,j}^t$ is the crop area; $WSR_{i,4}^t$ represents the water shortage rate of
agriculture sector; $FS_t$ and $FSR_t$ are the food shortage and food shortage rate,
respectively; and $TFP_t$ is the target food production.
**2.4 Society System Module**
The society system module is deployed to simulate the social process of the
integrated system. Environmental awareness and community sensitivity are two
primary terms of social state variables in socio-hydrologic modeling that indicate the
perceived level of threat to a community's quality of life (Roobavannan et al., 2018).
Environmental awareness describes societal perceptions of environmental degradation
within the prevailing value systems (Feng et al., 2019; Feng et al., 2016;
Roobavannan et al., 2018; Van Emmerik et al., 2014). Community sensitivity
indicates people's attitudes towards not only the environmental control, but also the
environmental restoration (Chen et al., 2016; Elshafei et al., 2014; Roobavannan et al.,
2018). As this study focuses on societal perceptions on environmental degradation,
environmental awareness based on the concept described in Van Emmerik et al. (2014)
was adopted as the social state variable. As water, energy, and food systems are
considered part of the environment in this study, environmental awareness is assumed
to be determined by the shortage rates of water, energy, and food. Environmental
awareness accumulates when the shortage rates of water, energy, and food exceed the
given critical values, but decreases otherwise. The dynamics of environmental

awareness can be described by equations (25)–(28).

$$\frac{dE}{dt} = \frac{dWA}{dt} + \frac{dEA}{dt} + \frac{dFA}{dt} \tag{25}$$

$$\frac{dWA}{dt} = \begin{cases} \eta_W * (\exp(\theta_W * (WSR - WSR_{crit})) - 1) & WSR > WSR_{crit} \\ -\omega_W * WA & WSR \leq WSR_{crit} \end{cases} \tag{26}$$

$$\frac{dEA}{dt} = \begin{cases} \eta_E * (\exp(\theta_E * (ESR - ESR_{crit})) - 1) & ESR \geq ESR_{crit} \\ -\omega_E * EA & ESR < ESR_{crit} \end{cases} \tag{27}$$

$$\frac{dFA}{dt} = \begin{cases} \eta_F * (\exp(\theta_F * (FSR - FSR_{crit})) - 1) & FDR \geq FDR_{crit} \\ -\omega_F * FA & FDR < FDR_{crit} \end{cases} \tag{28}$$

where $E$, $WA$, $EA$, and $FA$ are environmental awareness, water shortage awareness, energy shortage awareness, and food shortage awareness, respectively; $WSR$, $ESR$, and $FSR$ denote the shortage rates of water, energy, and food, respectively; $WSR_{crit}$, $ESR_{crit}$, and $FSR_{crit}$ represent the corresponding critical values of shortage rates, above which environmental deterioration can be perceived; $\eta_W$, $\eta_E$, and $\eta_F$ are the perception factors describing the community's ability to identify threats of degradation; $\theta_W$, $\theta_E$, and $\theta_F$ are the auxiliary factors for environmental awareness accumulation; and $\omega_W$, $\omega_E$, and $\omega_F$ denote the lapse factors that represent the decreasing rate of the shortage awareness of water, energy, and food, respectively.

**2.5 Respond Links**

Respond links are used to link society and water system modules through feedback. Respond links are driven by environmental awareness and food shortage awareness. The terms of feedback functions are based on the studies of Feng et al. (2019) and Van Emmerik et al. (2014), which have been established to have good performance and suitability, as they have been successfully applied to simulate the

human response to environmental degradation in the Murrumbidgee river basin
(Australia) and Hehuang region (China).

Environmental awareness increases with constant shortages in water, energy, and

food. As environmental awareness accumulates above its critical value, negative
feedback on socioeconomic factors is triggered (Figure 1). The growth of population,
GDP, and crop area will be constrained to alleviate the stress on the integrated system.
Notably, positive feedback on the expansion of crop area will be triggered to fill food
shortage as food shortage awareness exceeds its critical value (Figure 1). Although
food shortage awareness is part of environmental awareness, the negative feedback
driven by environmental awareness on crop area can only be triggered with the
prerequisite that food shortage awareness is below its threshold value, as food
production should first be assured. The respond links deployed by assuming feedback
functions are expressed in equations (29)–(31).
$$f_1(E) = \begin{cases} \delta_{rp}^E * (1 - \exp(\zeta_1 * (E - E_{crit})) & E > E_{crit} \\ 0 & else \end{cases} \tag{29}$$

$$f_2(E) = \begin{cases} \delta_{rg}^E * (1 - \exp(\zeta_2 * (E - E_{crit})) & E > E_{crit} \\ 0 & else \end{cases} \tag{30}$$

$$f_3(E, FA) = \begin{cases} \delta_{ra}^F * (\exp(\zeta_3^F * (FA - FA_{crit}) - 1) & FA > FA_{crit} \\ \delta_{ra}^E * (1 - \exp(\zeta_3^E * (E - E_{crit})) & FA < FA_{crit} \& E > E_{crit} \\ 0 & else \end{cases} \tag{31}$$

where $E_{crit}$ and $FA_{crit}$ are the critical values for environmental awareness and food
shortage awareness, respectively; $\delta_{rp}^E$, $\delta_{rg}^E$, and $\delta_{ra}^E$ denote the factors describing
feedback capability from environmental awareness; $\delta_{ra}^F$ is the factor describing
feedback capability from food shortage awareness; $\zeta_1$, $\zeta_2$, and $\zeta_3^E$ represent the
auxiliary factors for feedback functions driven by environmental awareness; and $\zeta_3^F$
is the auxiliary factor for feedback functions driven by food shortage awareness.

## 3 Case Study

### 3.1 Study Area

The Hanjiang river is the longest tributary of the Yangtze river. The total area of
the Hanjiang river basin is 159,000 km$^2$, divided into upper and mid-lower reaches
covering 95,200 and 63,800 km$^2$, respectively (shown in Figure 3). The Danjiangkou
reservoir is located at the upper boundary of the mid-lower reaches of the Hanjiang
river basin (MLHRB) and serves as the water source for the middle route of the
South–North water transfer project in China. Thus, the water availability in the
MLHRB is remarkably affected by the reservoir operation. In terms of energy, as the
population is large and the industry is developed in the MLHRB, the energy
consumption for urban water supply is high. For agriculture, as the land is flat and
fertile, MLHRB is considered an important grain-producing area, occupying one of
the nine major commodity grain bases in China (i.e., Jianghan plain) (Xu et al., 2019).

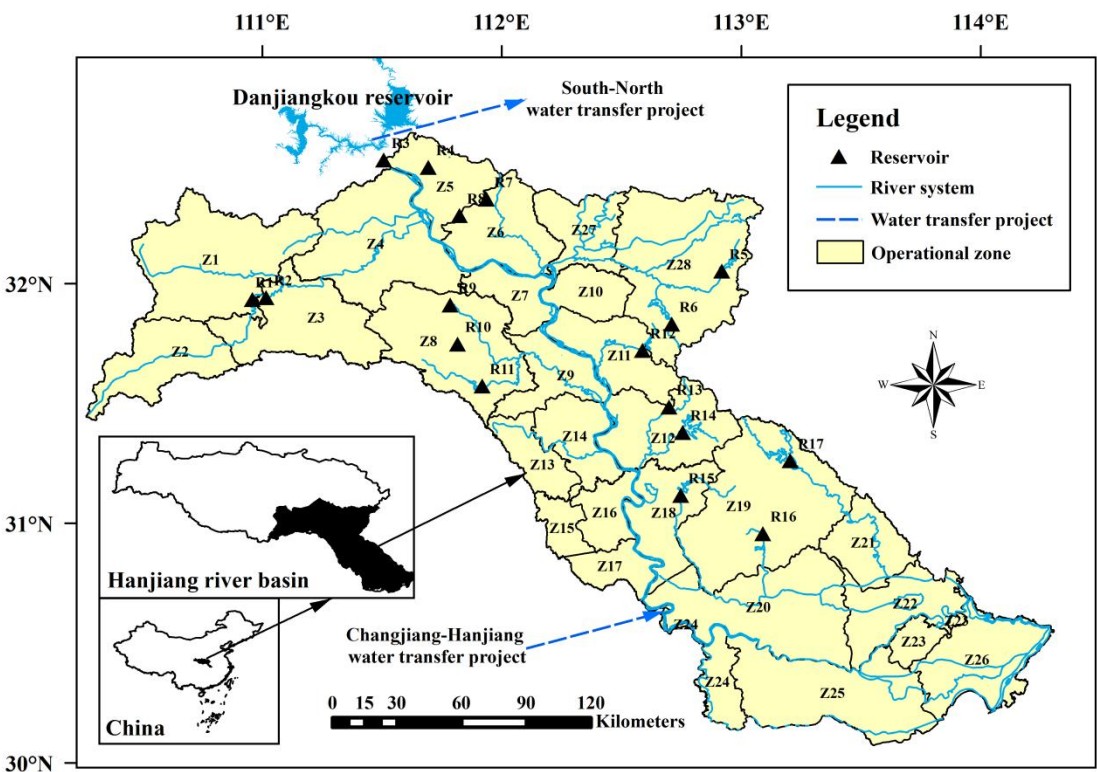

**Figure 3. Location of mid-lower reaches of Hanjiang river basin.**
However, owing to population expansion, rapid urbanization, and economic
development, the local demand for water, energy, and food is increasing enormously
(Zeng et al., 2021; Zhang et al., 2018). The contradictions between increasing demand
and limited resources will be intensified. Therefore, improving use efficiencies for
water, energy and food in MLHRB is urgent (Zhang et al., 2018; Liu et al., 2019). The
strictest water resources control system for water resources management policy, the
total quantity control of water consumed policy, and the energy-saving and
emission-reduction policy in China are implemented in the MLHRB to promote the
expansion of resource-saving technology and further improve the resource use
efficiencies in water, energy, and food systems. Therefore, the impacts of human
activities on the WEF nexus should be assessed to sustain the collaborative
development of the integrated system.
The socioeconomic data (i.e., population, GDP, and crop area) for water demand
projection were collected based on administrative units, whereas the hydrological data
were typically collected based on river basins. To ensure that the socioeconomic and
hydrological data are consistent in operational zones, the study area was divided into
28 operational zones based on the superimposition of administrative units and
sub-basins. Seventeen existing medium or large size reservoirs (the total storage
volume is 37.3 billion m$^3$) were considered to regulate water flows. Based on the
water connections between operational zones and river systems, the study area is
shown in Figure 4, including 2 water transfer projects (the South–North and
Changjiang–Hanjiang water transfer projects), 17 reservoirs, and 28 operational
zones.

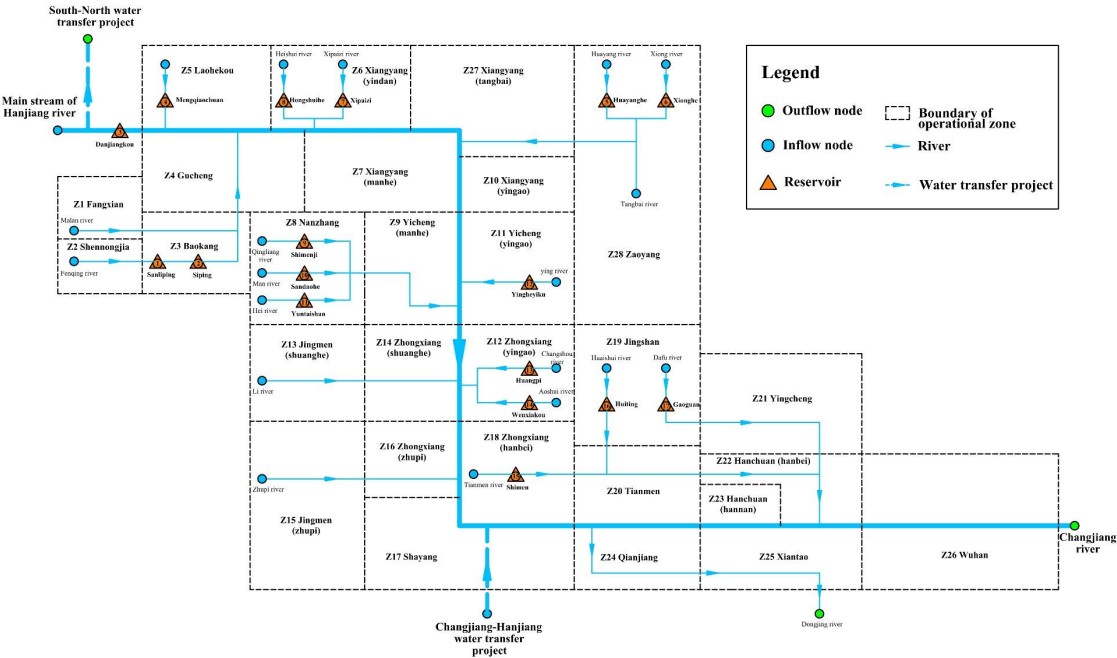


**Figure 4. Sketch of the water system for the mid-lower reaches of Hanjiang river basin.**

## 3.2 Data Sources

There are two main types of data: hydrological and socioeconomic data. The monthly historical discharge series of each operational zone and inflow of reservoirs from 1956 to 2016 were provided by the Changjiang Water Resources Commission (CWRC, 2016). The characteristics and operational rules of the 17 reservoirs listed in Table S1 in supplementary file were retrieved from the Hubei Provincial Department of Water Resources (HPDWR 2014). Socioeconomic data, including population, GDP, crop area, water use quota, energy use quota, and crop yield, during 2010–2019 were collected from the yearbooks of Hubei Province, which can be obtained from the Statistical Database of China's Economic and Social Development (http://data. cnki.net/). Notably, the agricultural water use quota was related to the annual effective precipitation frequency. Based on the precipitation frequency series during 1956–2016, four typical exceedance frequencies (i.e., P = 50%, 75%, 90%, and 95% are related to the wet, normal, dry, extreme dry years), were adopted to simplify agricultural water demand series. These historical data were further used to predict the future trajectories of the WEFS nexus.

## 4 Results and Discussion

The SDM was applied to the MLHRB. Specifically, water availability from 1956 to 2016 was adopted as the future water availability, while dynamic water demand was projected in water system module, both of which were inputted into water

resources allocation model. As the water resources allocation model in the water
system module took a monthly time step in the study (and the sub-time step was the
default value: 1 day), the annual water supply and water shortage were first
determined before being outputted to the energy system and food system modules,
respectively. The annual shortage rates of water, energy, and food were then used to
determine environmental awareness and further the feedback. Table 1 lists the initial
settings of the external variables for the integrated system. The co-evolutionary
behaviors of the WEFS nexus were analyzed as follows: (1) the system dynamic
model was calibrated using observed data, (2) co-evolution of the WEFS nexus was
then interpreted and analyzed, (3) the impacts of environmental awareness feedback
and water resources allocation on the WEFS nexus were discussed, and (4) sensitivity
analysis for WEFS nexus was tested.
**Table 1 Model initial condition setup.**

| Notation | Description | Unit | Value |
|---|---|---|---|
| $N_0$ | Population | million capita | 14.92 |
| $G_0$ | GDP | billion Yuan | 419 |
| $CA_0$ | Crop area | km$^2$ | 7,733 |
| $N_{cap}$ | ECC[a] of population | million capita | 20.00 |
| $G_{cap}$ | ECC of GDP | billion Yuan | 3,000 |
| $CA_{cap}$ | ECC of crop area | km$^2$ | 10,000 |
| $WQ_{\bullet,1}^0$, $WQ_{\bullet,1}^{min}$ | Initial and minimum municipal water use quota | m$^3$/(year*capita) | 56, 28 |

| | | | |
|---|---|---|---|
| $WQ_{\bullet,2}^{0}$, $WQ_{\bullet,2}^{min}$ | Initial and minimum rural water use quota | m³/(year*capita) | 25, 12.5 |
| $WQ_{\bullet,3}^{0}$, $WQ_{\bullet,3}^{min}$ | Initial and minimum industrial water use quota | m³/(10^4 Yuan) | 109, 54.5 |
| $WQ_{\bullet,4}^{0}$, $WQ_{\bullet,4}^{min}$ (P = 50%, 70%, 90%, and 95%) | Initial and minimum agricultural water use quota | million m³/km² | 0.77, 0.80, 0.90, 0.97 and 0.38, 0.40, 0.45, 0.49 |
| $EQ_{\bullet,j}^{0}$, $EQ_{\bullet,j}^{min}$ (j = 1, 2, 3, and 4) | Energy use quotas for municipal, rural, industry and agriculture sectors | kw*h/m³ | 0.29, 0.29, 0.29, 0[b] and 0.15, 0.15, 0.15 0 |
| $\sum_{j} CY_{\bullet,j}^{0}$ (j = 1, 2) | Crop yield | t/km² | 654 |
| $r_{P,0}$ | Growth rate of population | [-] | 0.003 |
| $r_{G,0}$ | Growth rate of GDP | [-] | 0.040 |
| $r_{CA,0}$ | Growth rate of crop area | [-] | 0.003 |
| $r_{qwu,0}$ | Growth rate of water use quota | [-] | -0.020 |
| $r_{e,0}$ | Growth rate of energy use quota | [-] | -0.004 |
| $r_{pro,0}$ | Growth rate of crop yield | [-] | 0.018 |
| PEA | Planning energy availability | [million kw*h] | 1,620 |
| TFP | Target food production | [million t] | 6,000 |

[a] ECC indicates the environmental carrying capacity. [b] As the primary source of water supply for agricultural use in
the study area is surface water, rather than groundwater, the energy consumption in the water supply process for
agricultural water use is negligible, and the energy use quota for agricultural water use is set as 0.

## 4.1 Model Calibration


As some parameters are adopted as auxiliary parameters, which are not equipped
with exactly physical definitions, there is no independent empirical data to calibrate
them. Therefore, by reviewing previous studies (Feng et al., 2019; Feng et al., 2016;
Van Emmerik et al., 2014) and expert knowledge, we evaluated the order of
magnitudes and rational boundaries for these parameters. An initial parameter
sensitivity analysis was then adopted to screen out the insensitive parameter, which
provided distinguishing 13 insensitive and 21 sensitive parameters. As the insensitive
parameters are not able to remarkably alter the system, the empirical values in
previous studies (Feng et al., 2019; Feng et al., 2016) were adopted. The sensitive
parameters in the model were then calibrated based on the observed data, and the
calibrated values are presented in Table S2 in supplementary file. The Nash–Sutcliffe
Efficiency (NSE) coefficient and percentage bias (PBIAS) (Krause et al., 2005; Nash
and Sutcliffe, 1970) were used to calibrate the model. When the NSE was >0.7 and
absolute value of PBIAS was <15%, the modeling performance was considered
reliable. The simulated state variables, including annual water demand, energy
consumption, food production, population, GDP, and crop area, were compared with
their observed values during 2010–2019. As shown in Table 2, the NSEs range from
0.74 to 0.97, and the corresponding PBIASs are from -4.2% to 5.2%, indicating that
both Malthusian model and Logistic model can effectively fit the observed data of
WEFS nexus.
**Table 2 NSE and PBIAS of state variables.**

| Model | Indicator | Water demand | Energy consumption | Food production | Population | GDP | Crop area |
|---|---|---|---|---|---|---|---|
| Malthusian model | NSE | 0.91 | 0.74 | 0.79 | 0.97 | 0.86 | 0.94 |
| | PBIAS (%) | -0.7 | 1.9 | -0.6 | -4.2 | 0.2 | -0.8 |
| Logistic model | NSE | 0.79 | 0.74 | 0.82 | 0.94 | 0.85 | 0.96 |
| | PBIAS (%) | -1.0 | 2.0 | -0.2 | 5.2 | 0.3 | -0.1 |

It's worth noting that the observed data can only cover the initial phase of WEFS nexus co-evolution. The environmental awareness stays at a low level and the feedback is not triggered, which indicates that feedback driven by high-level environmental awareness hasn't been calibrated yet. However, as environmental awareness is a subjective variable, there are no empirical data to calibrate it, which requires more evidences to show adaptive human response to environmental awareness. Hepburn et al. (2010) have reviewed studies on environmentally related human behavioral economics. Substantial studies indicate that environmental awareness is considered as an important factor in modelling socioeconomic decisions and policies for water, energy and food systems (Li et al., 2019; Li et al., 2021; Lian et al., 2018; Rockson et al., 2013; Xiong et al., 2016). For instance, Xiong et al. (2016) investigated the evolution newspaper coverage of water issues in China based on water-related articles in a major national newspaper, *People's Daily*. They found that economic development was the primary target of China before 2000. With the conflict between water demand and supply being intensified, concerns about water security arisen in the newspaper since 2000, which indicated that environmental awareness

towards water shortage emerged. Related policies (e.g., the strictest water resources
control system for water resources management policy in China) were thereby
implemented to constrain the over-speed socioeconomic expansion and further ensure
water security. Therefore, the established model still has potential to simulate the
co-evolution of WEFS nexus.

**4.2 Co-evolution of WEFS Nexus**

The calibrated system dynamic model was used to examine the properties of the
integrated system by simulating the co-evolution of state variables in the WEFS nexus.
Figure 5 shows the trajectories of population; GDP; crop area; water demand; energy
consumption; food production; shortage rates for water, energy, and food; awareness
for water shortage, energy shortage, and food shortage; and environmental awareness
during 2010–2070.

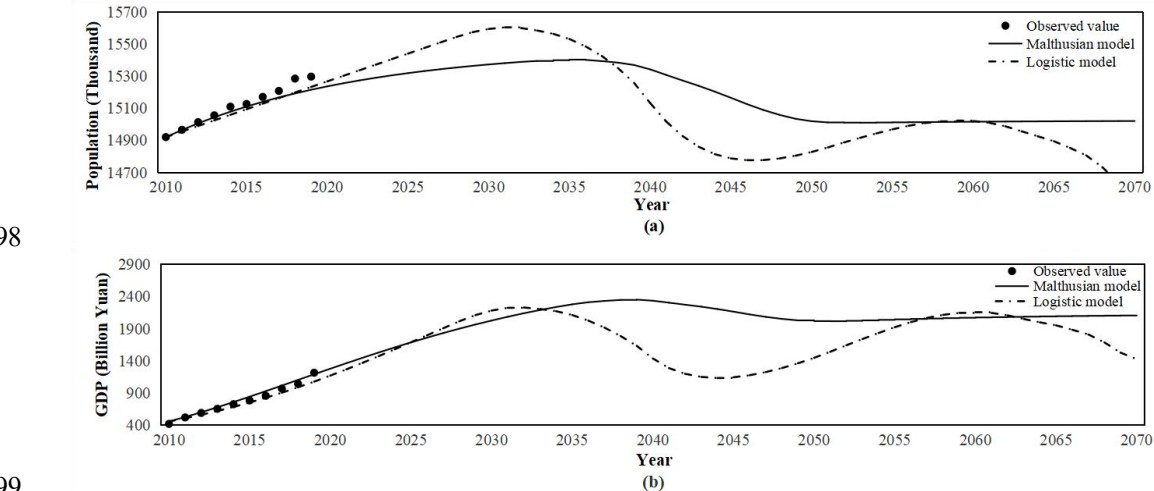

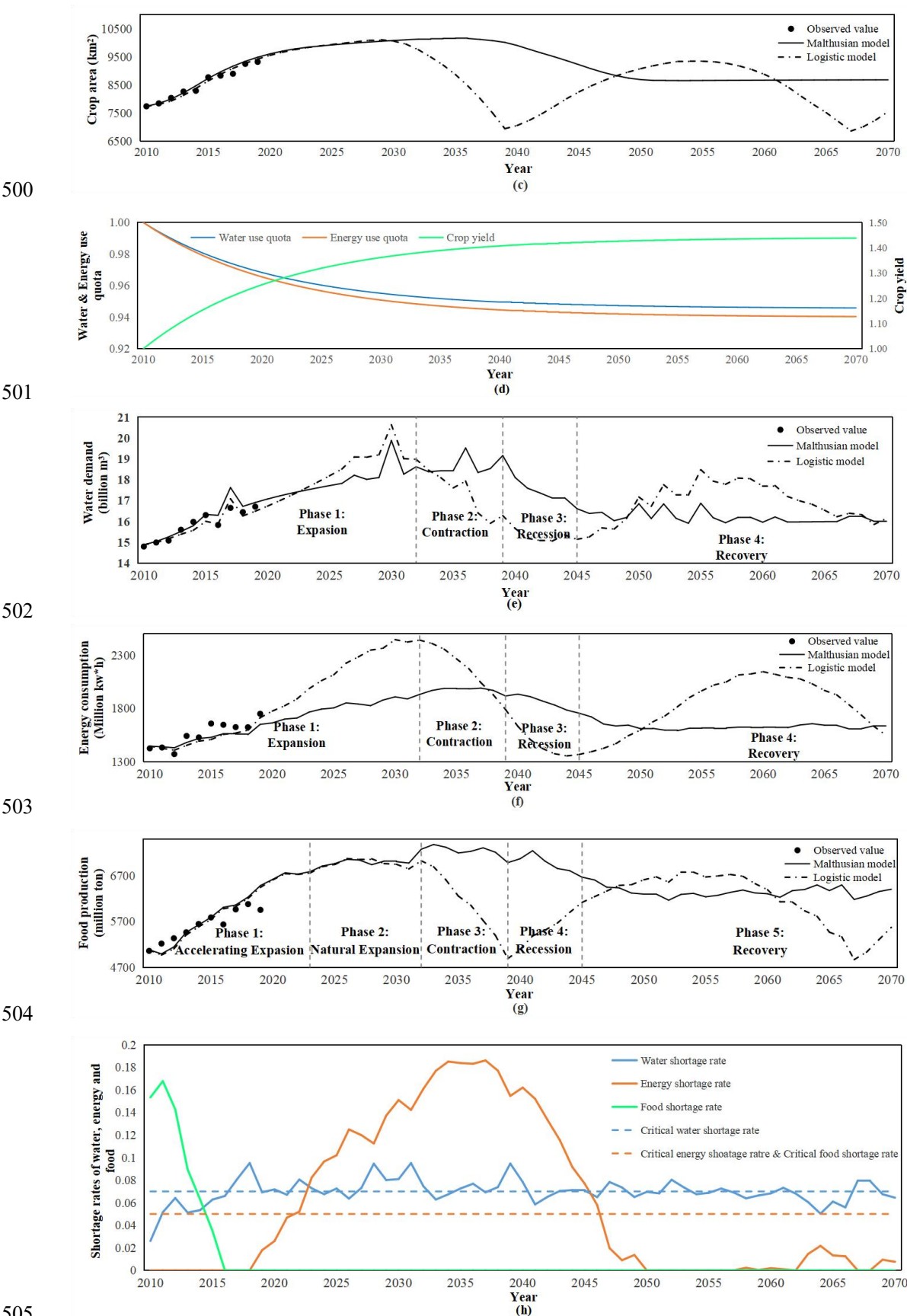







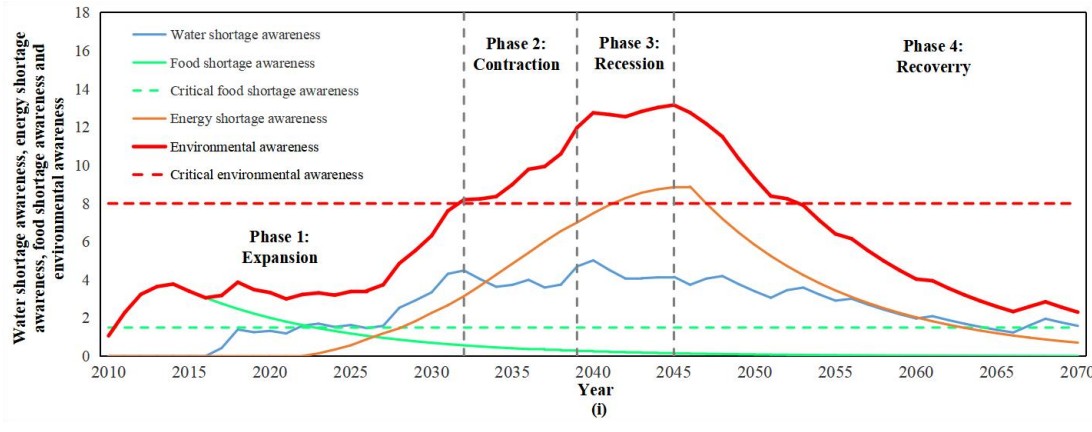


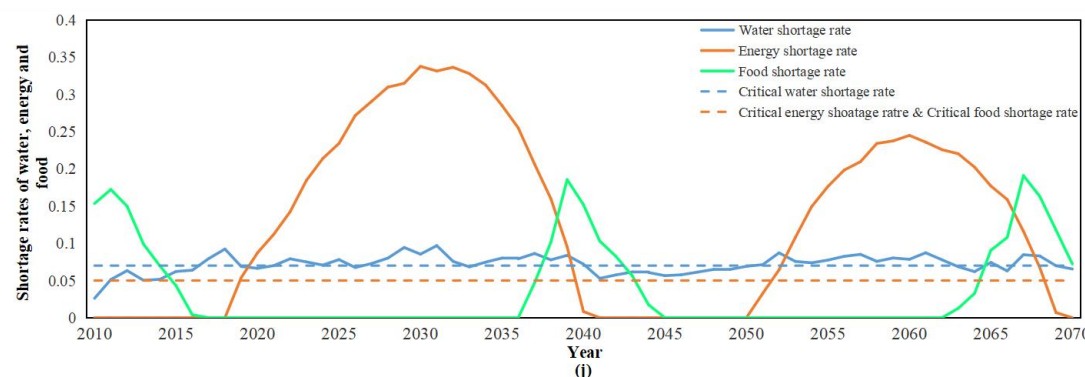


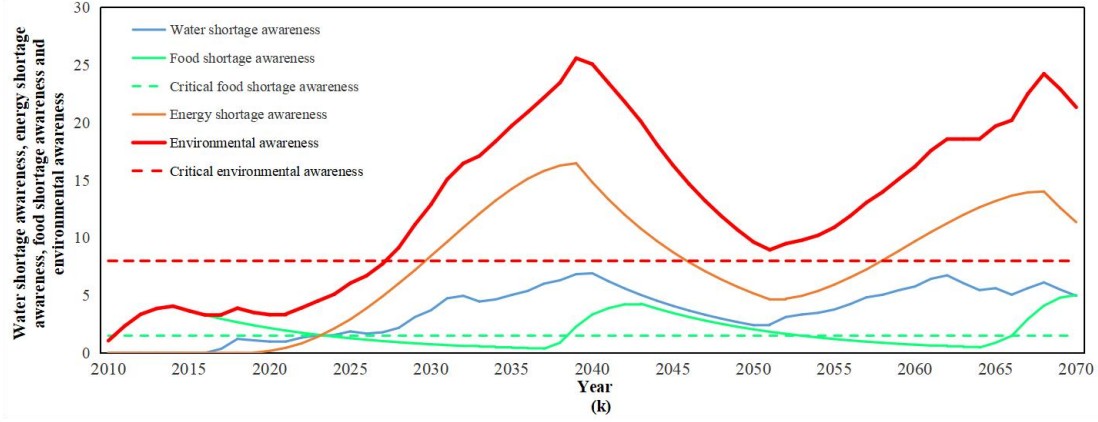


**Figure 5. Trajectories of state variables in WEFS nexus: (a) population; (b) GDP; (c) crop area; (d) percentage variations (compared with initial values) of water use quota, energy use quota, and crop yield; (e) water demand; (f) energy consumption; (g) food production; (h) shortage rates of water, energy, and food in Malthusian model; (i) water shortage awareness, energy shortage awareness, food shortage awareness, and environmental awareness in Malthusian model; (j) shortage rates of water, energy, and food in Logistic model; (k) water shortage awareness, energy shortage awareness, food shortage awareness, and environmental awareness in Logistic model.**

Based on the trajectory of environmental awareness, the co-evolution processes

of water demand and energy consumption in Malthusian model were divided into four

phases: expansion, contraction, recession, and recovery, which was consistent with the
results in Feng et al. (2016) and Elshafei et al. (2014).. Food production was divided
into five phases based on the trajectory of food shortage awareness: accelerating
expansion, natural expansion, contraction, recession, and recovery. The four phases in
the co-evolution process for water demand and energy consumption can be interpreted
as follows.

With environmental awareness below its critical value, the negative feedback on

socioeconomic sectors is not triggered, and water demand, as well as energy
consumption, increases rapidly, which is defined as expansion phase (2010–2032). In
the beginning of co-evolution, the water and energy demands can be satisfied by
water and energy availability. The shortage rates of water and energy were typically
below their critical values (Figure 5 (h)), and thus, shortage awareness of water and
energy remained at a low level as shown in Figure 5 (i). Despite food shortage struck
the system in the beginning, the shortage rate of which was 0.153 and more than its
critical value 0.05, the environmental awareness led by food shortage awareness was
still within its critical value 8.0. Therefore, environmental awareness feedback wasn't
triggered to constrain socioeconomic sectors, and water demand, as well as energy
consumption, thereby keeps increasing.

As environmental awareness exceeds its critical value, negative feedback on

socioeconomic sectors is triggered, and water demand and energy consumption is
constrained, which is defined as contraction phase (2033–2039). Although quotas for
water use and energy use decreased (Figure 5 (d)) with technological advancement,
water demand and energy consumption kept lowly increasing owing to the continuous
socioeconomic expansion (Figure 5 (a), (b), and (c)). Shortage rates of water and
energy remained over their critical values (Figure 5 (h), and (i)), leading the increases
of water shortage awareness and energy shortage awareness, and further
environmental awareness. Consequently, environmental awareness exceeded its
critical value in 2033 and continued to increase. Negative feedback on socioeconomic
sectors was triggered and strengthened. Water demand and energy consumption
gradually increased with decreasing rate and reached their maximum values of 19.2
billion $m^3$ and 1,916 million kw*h, respectively, at the end of the contraction phase.
As environmental awareness accumulates to the maximum value, water demand,
and energy consumption decrease significantly, which is defined as recession phase
(2040–2045). Environmental awareness feedback indeed constrained water demand
and energy consumption, which decreased but still exceeded local water and energy
carrying capacities. Therefore, as the shortage rates of water and energy remained
exceeding their critical values (Figure 5 (h)), environmental awareness continued
accumulating and reached the maximum value of 13.2 at the end of the recession
phase, thereby decreasing water demand and energy consumption.
As environmental awareness gradually decreases below its critical value, water
demand and energy consumption decrease slightly and then tend to stabilize, which is
defined as recovery phase (2046–2070). With continuous decline of socioeconomic
sectors, water demand and energy consumption gradually decreased within their
carrying capacities. The shortage rates of water and energy have then decreased to
below their critical values since 2047, resulting in the decreases in water shortage
awareness and energy shortage awareness (Figure 5 (h) and (i)). As the environmental
awareness decreased below its critical value, negative feedback was removed, and the
integrated system tended to stabilize.
The co-evolution process of food production can be interpreted in the similar
way. It's worth noting that the accelerating expansion phase (2010–2022) is unique
for food production. As the food production cannot satisfy the target value in the
beginning of co-evolution, food shortage emerged and led the increase of food
shortage awareness (Figure 5 (h), and (i)). With food shortage awareness increasing
over its critical value, positive feedback on crop area was triggered, and further
accelerated the increase of food production.
For Logistic model, socioeconomic sectors kept increasing in the initial phase.
The rapid socioeconomic expansion was slowed down until the negative feedback
driven by environmental awareness was triggered. With the increasing environmental
awareness,  socioeconomic  recession  was  followed.  Since  the  decreasing
socioeconomic sectors were much lower than their environmental capacities and
feedback driven by environmental awareness was weakening, the variables turned to
increase again to approach to their environmental capacities, and rolled in cycles.
One of the major differences between results of Malthusian model and Logistic
model is that state variable evolution in logistic model fluctuates remarkably and
performs periodicity. However, it's worth noting that the socioeconomic expansion in
the future will slow down and tend to stabilization (He et al., 2017; Lin et al., 2016),
the growth rate of which will thereby decrease as time goes. Moreover, the economic
development in the study area is also expected to gradually grow and then remains
stable according to the Integrated Water Resources Planning of Hanjiang River Basin
(CWRC, 2016). As the periodic fluctuation for WEFS nexus evolution through
Logistic model is not consistent with the slowed socioeconomic expansion in
foreseeable future and cannot fitly satisfy the planning in the study area, Logistic
model is not adopted. Malthusian model can fitly meet the demand mentioned above,
which is thereby applied for further analysis on WEFS nexus in our study.
**4.3 Impacts of Environmental Awareness Feedback and Water Resources**
**Allocation on WEFS Nexus**

To determine the potential impacts of environmental awareness feedback and

water resources allocation on the WEFS nexus, four scenarios were set, the
description of which is provided in Table 3. The *Ecrit* and *FAcrit* under scenario II
were set as 10,000 to ensure that the feedback cannot be triggered in the study, and the
*WSRcrit* in scenarios III and IV were set as 0.15 to avoid the explosion of water
shortage awareness. The other parameters in scenarios II, III, and IV were consistent
with the calibrated values of scenario I, as listed in Table S2. Scenarios I and II and
scenarios III and IV were used to investigate the impacts of environmental awareness
feedback and water resources allocation on the WEFS nexus, respectively. The
average annual values of water demand, energy consumption, food production, and
shortage rates for water, energy, and food are listed in Table 4. Figure 6 shows the
trajectories of key state variables of the integrated system, including water demand;
energy consumption; food production; shortage rates for water, energy, and food;
awareness of water shortage, energy shortage, and food shortage; and environmental
awareness.
**Table 3 Scenario description for assessing the impacts of environmental awareness feedback**
**and water resources allocation on WEFS nexus.**

| Scenario | Environmental awareness feedback | Water resources allocation | Parameter setting |
|---|---|---|---|
| I | Yes | Yes | Calibrated values |
| II | No | Yes | *Ecrit*, *FAcrit*: 10,000; others: calibrated values |
| III | Yes | Yes | *WSRcrit*: 0.15; others: calibrated values |
| IV | Yes | No | *WSRcrit*: 0.15; others: calibrated values |

**Table 4 Average annual values for the state variables in WEFS nexus.**

| Scenario | Water demand (billion m³) | Energy consumption (million kw*h) | Food production (million t) | Water shortage rate | Energy shortage rate | Food shortage rate |
|---|---|---|---|---|---|---|
| I | 16.94 | 1,710 | 6,519 | 7.03% | 5.80% | 1.07% |
| II | 17.66 | 1,930 | 6,248 | 7.44% | 17.16% | 1.74% |
| III | 17.29 | 1,761 | 6,638 | 7.20% | 8.25% | 1.08% |
| IV | 14.36 | 884 | 6,344 | 15.89% | 0.00% | 3.08% |


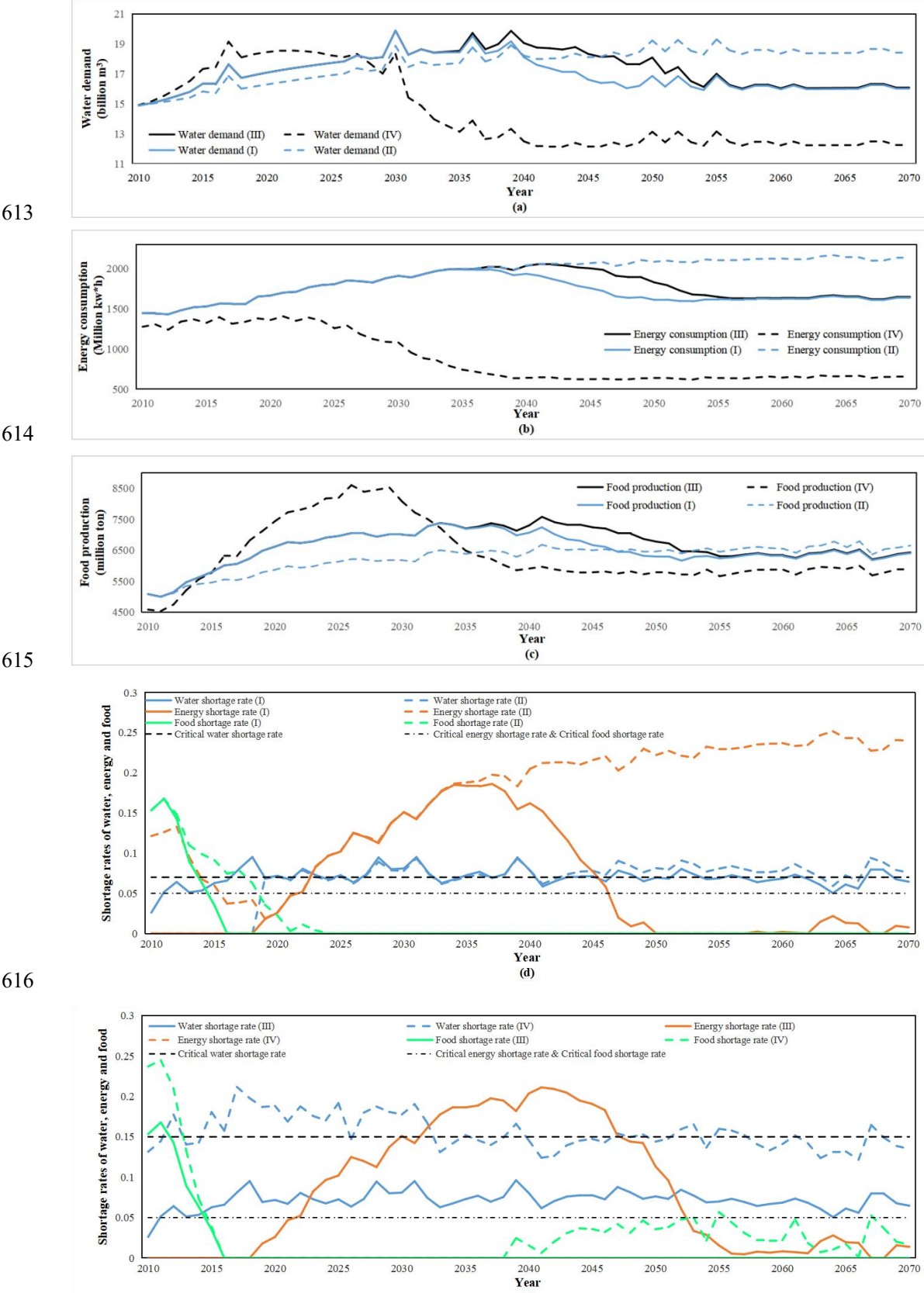





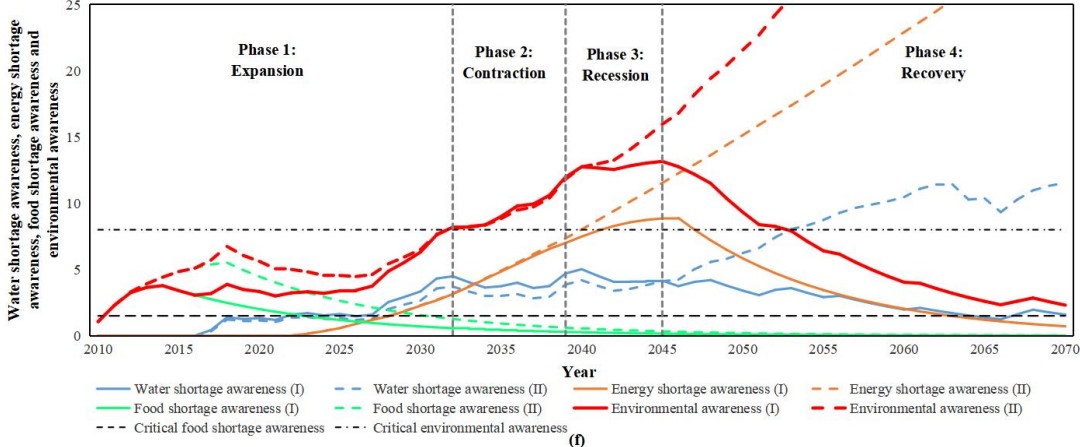

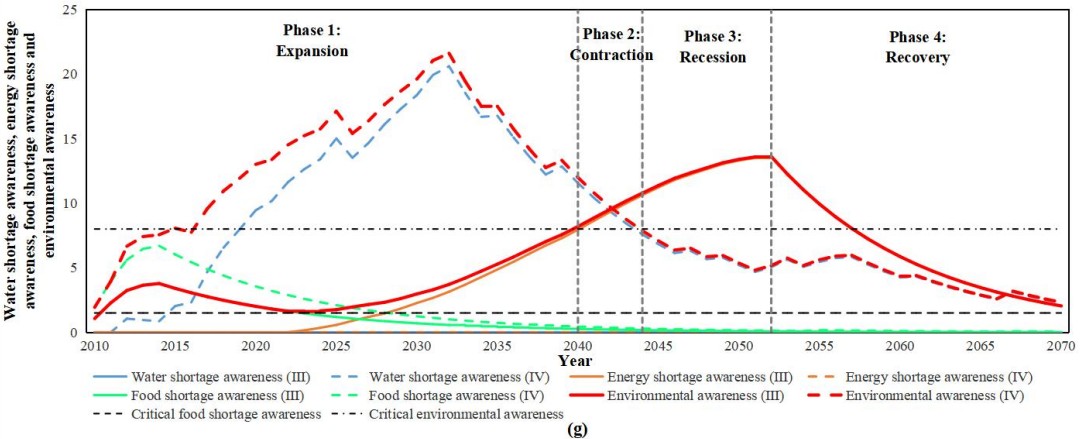



**Figure 6. Trajectories of state variables in WEFS nexus under scenario I, II, III, and IV: (a) water demand; (b) energy consumption; (c) food production; (d) and (e) shortage rates of water, energy, and food; (f) and (g) water shortage awareness, energy shortage awareness, food shortage awareness, and environmental awareness.**

### 4.3.1 WEFS Nexus Response to Environmental Awareness Feedback

Environmental awareness indicates societal perceptions of resources shortages and is the driving factor of feedback on socioeconomic sectors. Both the average annual water demand and energy consumption increased from 16.94 billion $m^3$ and 1,710 million t under scenario I to 17.66 billion $m^3$ and 1,930 million t under scenario II, respectively, as environmental awareness feedback was removed, whereas the food production decreased slightly, from 6,519 million t to 6,248 million t. Specifically, owing to high food shortage in the accelerating expansion phase of food production,

the positive feedback on crop area was triggered by food shortage awareness to
accelerate the increase in crop area. Food production was thus evidently larger when
feedback was considered in Figure 6 (c). Food shortage was then alleviated, and the
average shortage rate decreased from 1.74% to 1.07%. The increasing crop area
meanwhile led to an increase in agricultural water demand (Figure 6 (a)). However, as
the increasing water demand remained within the carrying capacity, little difference in
the water shortage rate existed between scenarios I and II (i.e., 7.03% and 7.44%,
respectively). As the water supply was efficiently ensured, the impacts on urban water
supply and the corresponding energy consumption were negligible. As water demand
and energy consumption increased rapidly in the expansion phase, environmental
awareness increased remarkably owing to the constant water and energy shortages, as
shown in Figure 6 (d) and (f). Negative feedback was triggered to constrain the
socioeconomic expansion. Compared with scenario II, water demand and energy
consumption decreased remarkably under scenario I. The stress on water and energy
supplies was thus relieved, particularly for the energy system, the shortage rate of
which decreased from 17.16% to 5.80%. Therefore, environmental awareness can
efficiently capture resources shortages and regulate the pace of socioeconomic
expansion through feedback, which can maintain the integrated system from constant
resources shortages to sustain the concordant development of the WEFS nexus.
**4.3.2 WEFS Nexus Response to Water Resources Allocation**
Water is considered the major driving factor for the WEFS nexus. Rational water
resources management plays an important role in the sustainable development of the
WEFS nexus. Water resources allocation can regulate the water flow by reservoir
operation, which is considered one of the most effective tools for water resources
management. Based on the Integrated Water Resources Planning of Hanjiang River
Basin (CWRC, 2016), domesticity and ecology water uses should be ensured first.
The priorities for water use from high to low are municipal and rural domesticity,
in-stream ecology, and industrial and agricultural sectors, respectively. The average
annual water demand, supply, and shortage under scenarios III and IV are listed in
Table 5.
**Table 5 Water resources allocation results under scenarios III and IV (million m$^3$).**

| Scenario | Variables | Municipal | Rural | Industry | Agriculture | In-stream ecology | Total |
|---|---|---|---|---|---|---|---|
| III | Demand | 388 | 181 | 6,504 | 6,433 | 3,779 | 17,286 |
| | Supply | 387 | 181 | 5,785 | 6,034 | 3,654 | 16,042 |
| | Shortage | 1 | 0 | 719 | 399 | 124 | 1,244 |
| | Shortage rate | 0.24% | 0.23% | 11.05% | 6.21% | 3.29% | 7.20% |
| IV | Demand | 361 | 170 | 3,330 | 6,720 | 3,779 | 14,359 |
| | Supply | 330 | 155 | 2,622 | 5,658 | 3,312 | 12,077 |
| | Shortage | 31 | 15 | 708 | 1,062 | 466 | 2,282 |
| | Shortage rate | 8.67% | 8.69% | 21.26% | 15.80% | 12.34% | 15.89% |

Despite the increase in water demand from 14,359 to 17,286 million m$^3$ under
scenario III, the water supply also increased from 12,077 to 16,042 million m$^3$. The
total water shortage rate decreased from 15.89% to 7.20% owing to rational water
resources allocation. As more available water resources can be stored in the flood
season and then released in the dry season through reservoir operation, the uneven
temporal and spatial distributions of available water resources were remarkably
relieved, thereby increasing the water supply insurance. For water use sectors, water
shortages were primarily found in industrial and agricultural sectors (719 and 399
million m$^3$, respectively), and other sectors can be satisfied under scenario III. Water
shortage became more serious under scenario IV, as the water shortage rates of these
five sectors increased significantly in Table 5, from 0.24%, 0.23%, 11.05%, 6.21%,
and 3.29% to 8.67%, 8.69%, 21.26%, 15.80%, and 12.34%, respectively. To analyze
the spatial distribution of water shortage rates, Figure 7 shows the water shortage rate
in each operational zone under scenarios III and IV. The water shortage rates of the
study area under scenario IV were evidently higher than those under scenario III,
particularly for the operational zones located at the basin boundaries (e.g., operational
zones Z1, Z2, Z8, Z12, Z13, Z21 and so on). As the boundary zones are far away from
the mainstream of the Hanjiang river and their local water availability is unevenly
distributed, the regulating capacity of the water system is limited and is not
sufficiently strong to ensure the water supply.

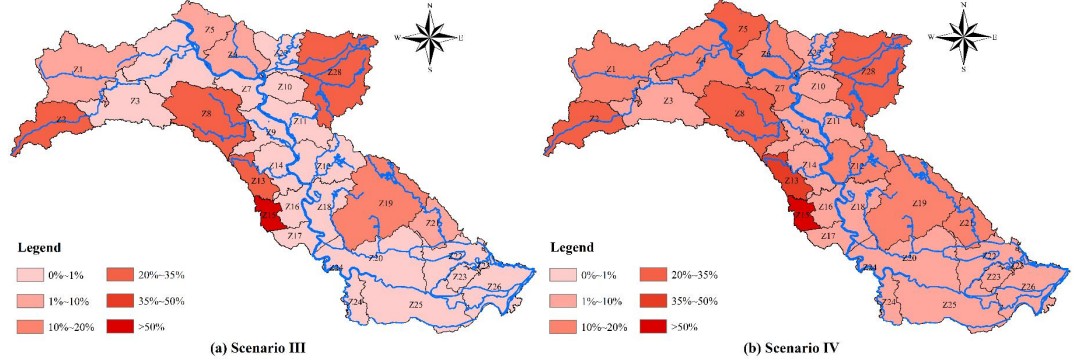

**Figure 7. Distribution of water shortage rates.**
For the co-evolution of WEFS nexus, a remarkable decrease in the average
annual water demand and energy consumption was observed as water resources
allocation was removed from 17.29 billion m³ and 1,761 million t under scenario III
to 14.36 billion m³ and 884 million t under scenario IV, while the food production
also decreased slightly from 6,638 million t to 6,344 million t. Under scenario IV
without considering water resources allocation, the average water shortage rate was
15.89%, exceeding the critical value. Water shortage awareness continued to
accumulate (Figure 6 (g)). As the water supply could not be effectively ensured and
remained at a low level, the energy consumption for urban water supply was small
and always within its planning value. No energy shortage awareness was accumulated
at the beginning of the co-evolution shown in Figure 6 (g). Meanwhile, as agricultural
water demand cannot be ensured, food production was also lowered (Figure 6 (c)).
Higher food shortages then led to higher food shortage awareness (Figure 6 (e), and
(g)). Thus, positive feedback to increase crop area was strengthened. As observed in
Figure 6 (a) and (c), the water demand increased slightly and food production
increased rapidly. As environmental awareness accumulated over its critical value in
2015 and continued to increase, negative feedback to constrain the socioeconomic
expansion was triggered and continued to strengthen. The energy consumption
thereby continued to decrease in Figure 6 (b), accounting for the significant decrease
in the energy shortage rate (i.e., from 8.25% to 0). Environmental awareness increased
and reached the maximum value of 21.6 in 2032 owing to the constant water shortage.
With the strong negative feedback, the water demand and food production decreased
remarkably and remained at a low level, as shown in Figure 6 (a) and (c), which
accounts for the increasing food shortage rate (i.e., from 1.08% to 3.08%).

With water resources allocation taken into account, water shortage was

significantly alleviated under scenario IV, as discussed in the water resources
allocation results (from 15.89% scenario IV to 7.20% under scenario III). The water
shortage rate remained below its critical value in the entire co-evolution process
(Figure 6 (e)). Thus, there was no accumulation of water shortage awareness shown in
Figure 6 (g). Energy consumption continued to increase as the water supply was
ensured. Environmental awareness accumulation was primarily due to energy
shortage.

Overall, water resources allocation can effectively alleviate water shortage to

decrease water shortage awareness by increasing the water supply. The increase in
environmental awareness is primarily due to the constant high-level energy shortage
rate. Therefore, planning energy availability is the primary boundary condition for
sustainable development of the WEFS nexus when water resources allocation is
considered. Under the scenario without considering water resources allocation, the
risk of water shortage is high. Water shortage awareness continues to accumulate and
remains at a high level under scenario IV, which further contributes to high-level
environmental awareness. The energy consumption and food production will be
decreased by negative feedback. Water availability becomes the vital resource
constraining the concordant development of the WEFS nexus.

## 4.4 Sensitivity Analysis for WEFS Nexus

As is discussed above, both environmental awareness feedback and water resources allocation are of great significance to WEFS nexus, the sensitivity analysis of which is conducted to help managers to identify the important parameters and rational water resources allocation schemes for the integrated system.

As environmental awareness feedback is dominated by the critical values and boundary conditions of the WEFS nexus, seven parameters were selected for sensitivity analysis (i.e., parameter 1~7 in Table 6). For water resources allocation, different reservoir operation schemes were adopted by adjusting water release from reservoir. Specifically, a multiplier for water release was added as a parameter to demonstrate the ratio to water release in scenario I (i.e., parameter 8 in Table 6). Each parameter was varied by the given increment, with the other parameters remaining unchanged. The maximum and minimum values, as well as the increments for the seven parameters, are listed in Table 6. Parameter sensitivity analysis was then conducted by analyzing the trajectories of environmental awareness, water demand, energy consumption, and food production, as shown in Figures 8, 9, 10, and 11.

**Table 6 Parameter set for sensitivity analysis.**

| No. | Parameter | Description | Min. | Max. | Increment |
|-----|-----------|-------------|------|------|-----------|
| 1 | $WSRcrit$ | Critical water shortage rate | 0.05 | 0.15 | 0.01 |
| 2 | $ESRcrit$ | Critical energy shortage rate | 0.05 | 0.15 | 0.01 |
| 3 | $FSRcrit$ | Critical food shortage rate | 0.05 | 0.15 | 0.01 |
| 4 | $PEA$ | Planning energy availability | 1,550 | 1,750 | 20 |

| 5 | *TFP* | Target food production | 5,200 | 6,200 | 100 |
| 6 | *FAcrit* | Critical food shortage awareness | 1 | 3 | 0.2 |
| 7 | *Ecrit* | Critical environmental awareness | 5 | 10 | 0.5 |
| 8 | *Qmultiplier* | Multiplier of water release from reservoir | 0.5 | 1.5 | 0.1 |

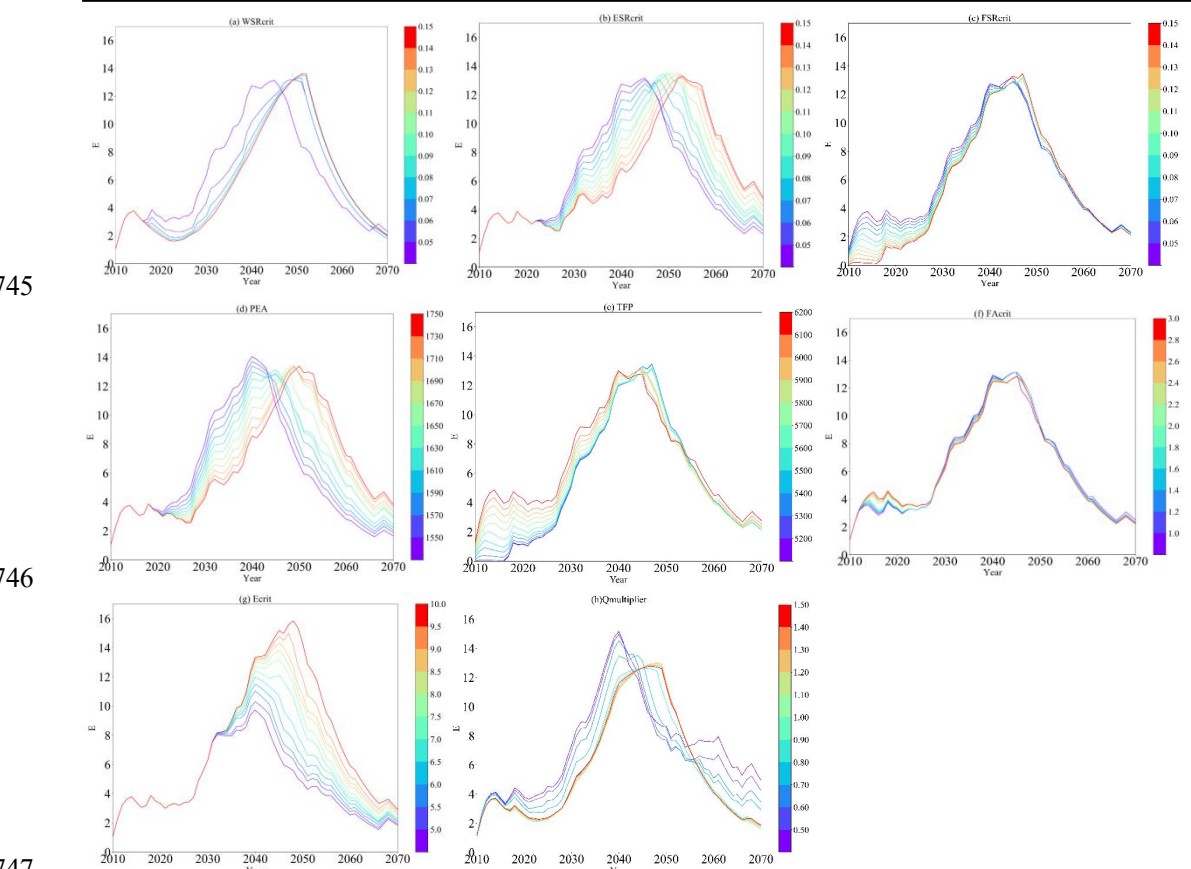

**Figure 8. Trajectories of environmental awareness with varied parameters.**

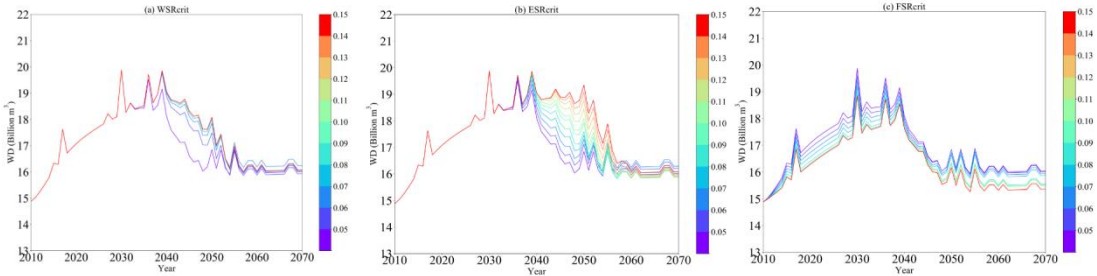

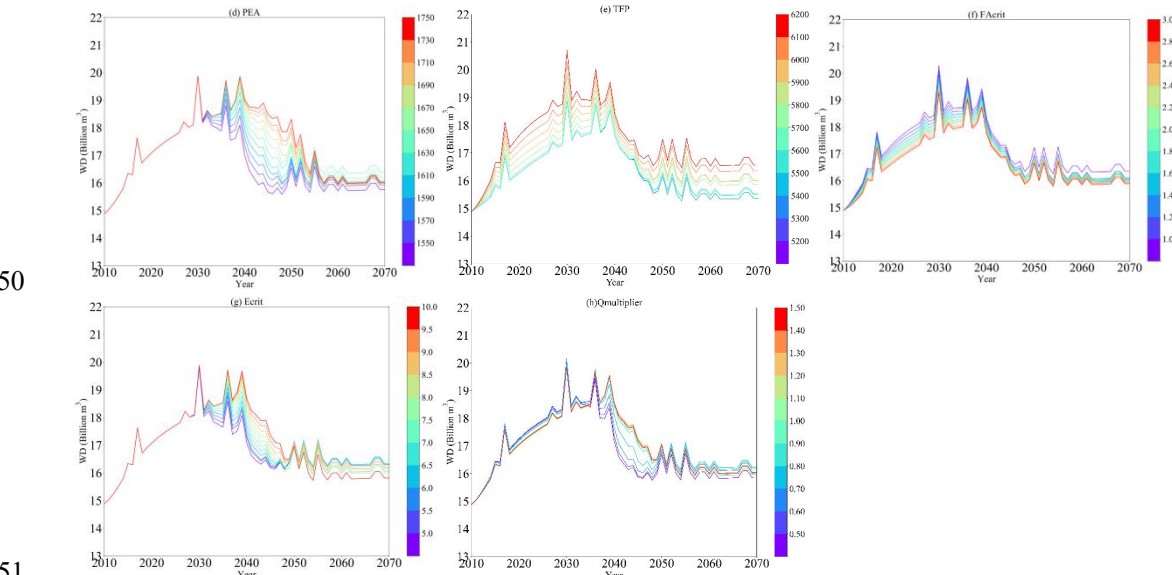



**Figure 9. Trajectories of water demand with varied parameters.**

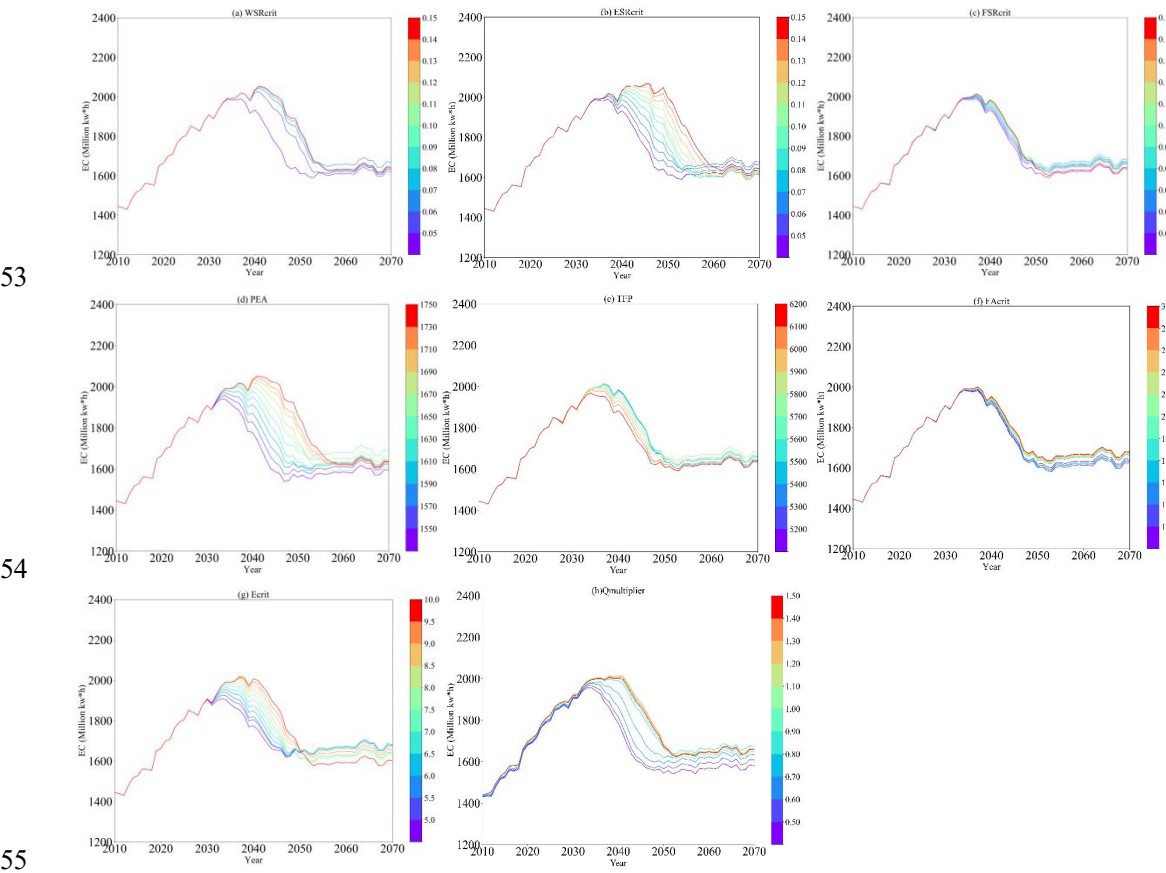




**Figure 10. Trajectories of energy consumption with varied parameters.**

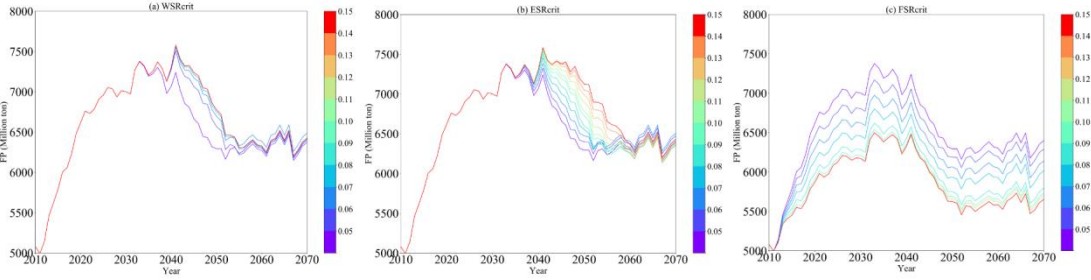


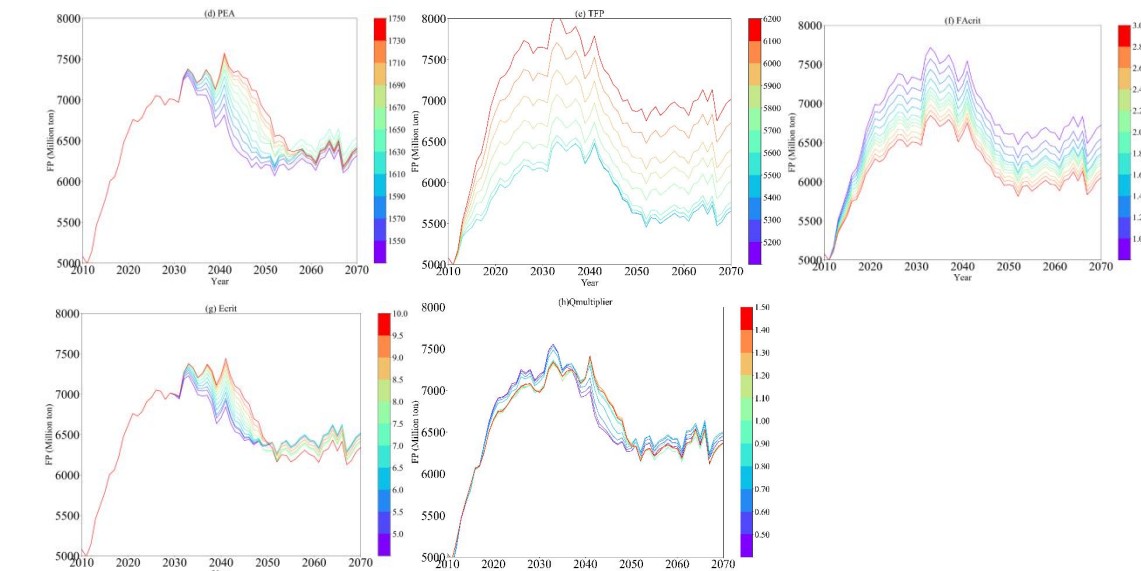

**Figure 11. Trajectories of food production with varied parameters.**

**4.4.1 Sensitivity Analysis of Environmental Awareness Feedback on WEFS Nexus**

The variations in the parameters 1~7 can evidently change the trajectory of environmental awareness shown in Figure 8. The socioeconomic sectors including water demand, energy consumption, and food production were then changed by feedback driven by environmental awareness (Figure 9, 10, and 11), indicating that WEFS nexus is sensitive to the seven parameters.

Specifically, the sensitive responses to parameters *WSRcrit*, *ESRcrit*, *PEA*, and *Ecrit* primarily occurred in the contraction and recession phases of the co-evolution process for WEFS nexus. As demands from water and energy systems can always be ensured by abundant resources availability in the expansion phase, limited water and energy shortages were observed. Environmental awareness accumulated primarily from food shortage awareness but remained below its critical value (Figure 5 (i)). As the feedback due to environmental awareness was not sufficiently strong, the impacts on the co-evolution of WEFS nexus were negligible and were considered as the

insensitivity. However, with social development, water demand and energy
consumption continued to grow and increase over the local carrying capability,
leading an increase in environmental awareness. Negative feedback on socioeconomic
sectors was then triggered. *WSRcrit* and *ESRcrit* are the critical values that determine
the awareness of water and energy shortages to accumulate, and *PEA* indicates the
amount of planning energy availability, which directly determines the energy shortage.
The environmental awareness accumulation can be thereby accelerated by
constraining *WSRcrit*, *ESRcrit*, and *PEA* (Figure 8 (a), (b), and (d)). *Ecrit* is the
threshold for the negative feedback triggering driven by environmental awareness. A
lower *Ecrit* means community is more sensitive to resources shortage and feedback is
easier to trigger (Figure 8 (g)). Therefore, environmental awareness feedback to
constrain socioeconomic expansion can be advanced and strengthened by lowering
*WSRcrit*, *ESRcrit*, *PEA*, and *Ecrit*, accounting for the sensitive response of WEFS
nexus in contraction and recession phases.

*FSRcrit*, *TFP,* and *FAcrit* performed sensitivity during the entire co-evolution

process for WEFS nexus. As food shortages were considerable in the accelerating
expansion phase, food shortage awareness increased rapidly, driving the feedback to
increase crop area. *TFP* can directly determine food shortage, and *FSRcrit* and *FAcrit*
determine thresholds for food shortage awareness accumulation and feedback
triggering by food shortage awareness, respectively. Positive feedback on crop area to
increase food production can thus be advanced and strengthened by constraining
*FSRcrit*, *TFP*, and *FAcrit* (Figure 8 (c), (e), and (f)). The crop area then continued
increasing until environmental awareness feedback was triggered, resulting in the
increases in food production (Figure 11 (c), (e), and (f)) and water demand from
agricultural sector (Figure 9 (c), (e), and (f)). As the agricultural water use was
directly drawn from river system, the energy use quota during water supply was small
and negligible. Energy consumption was thus not sensitive to *FSRcrit*, *FAcrit*, and
*TFP* as shown in Figure 10. Therefore, constraining *FSRcrit*, *FAcrit*, and *TFP* is an
effective way to increase food production by advancing and strengthening the
feedback driven by food shortage awareness, which accounts for the sensitive
responses of environmental awareness, water demand, and food production in
expansion phase.
Simultaneously, it's worth noting that although constraining *WSRcrit*, *ESRcrit*,
*PEA*, and *Ecrit* can maintain the integrated system from constant water shortage and
energy shortage, the over-constrained condition can also sharply increase
environmental awareness (Figure 8 (a), (b), (d), and (e)). Environmental awareness
feedback was remarkably advanced, which shortened the expansion phase and led to
violent degradation of socioeconomic sectors (indicated by drastic decreases of water
demand, energy consumption and food production in Figure 9, 10, and 11,
respectively). The sustainability of WEFS nexus was seriously challenged. Similarly,
despite food production can be effectively increased by constraining *FSRcrit*, *FAcrit*,
and *TFP*, the over-constrained condition will cause a considerable increase in water
demand, as shown in Figure 9 (c), (e), and (f), which will further put stress on the
water supply. Moreover, the regulating capacity of the local system should also be
considered during parameter selection. For example, there was an abrupt decrease
when *WSRcrit* was set to 0.05, as shown in Figure 9 (a), Figure 10 (a), and Figure 11
(a). Violent socioeconomic degradation dominated by environmental awareness
feedback was triggered to decrease environmental awareness, indicating that the
*WSRcrit* was over-constrained and exceeded the regulating capacity of the local water
system. Therefore, a rational parameter setting should be based on the sustainability
of long-term co-evolution for socioeconomic sectors and the regulating capacity of
the local system, which is of great significance for sustaining the stability of the
WEFS nexus.
As each shortage is experienced by different users with different connections to
basin development dynamics (e.g., shortages from water, energy, and food are
aggregated into environmental awareness, despite the food which is planned to be
exported is considered in target food production), it's necessary to discuss the
contributions to environmental awareness from water, energy, and food systems.
Therefore, three weight factors were assigned to shortage awareness of water, energy,
and food in equation (32) to adjust the over-estimated or under-estimated
environmental awareness due to discordant scales. For instance, considering the target
food production comprises inner food demand and exported food, the environmental
awareness within the basin is over-estimated, and the weight factor for food shortage
awareness can be set lower than 1.0 as a reduction factor to decrease current food
shortage awareness. Sensitivity analysis was then conducted. Each weight factor was
varied by given increment, while the other two weight factors were set to 1.0 as
reference. The results are presented in Figure S1, S2, S3, and S4 in supplementary
file.
$$\frac{dE}{dt} = wf_1 * \frac{dWA}{dt} + wf_2 * \frac{dEA}{dt} + wf_3 * \frac{dFA}{dt} \qquad (32)$$

where $wf_1$, $wf_2$, and $wf_3$ are the weight factors for water, energy, and food shortage
awareness, respectively.
WEFS nexus is sensitive to shortage awareness weight factors. Specifically,
weight factors for water and energy shortage awareness can remarkably impact the
recession phases of water demand, energy consumption, and food production. Lower
weight factor can delay environmental awareness accumulation, and thus extend the
contraction phase. However, more violent socioeconomic deterioration was also
accompanied in the later recession phase, which consequently led the slightly smaller
socioeconomic size in recovery phase. Weight factor for food shortage awareness can
effectively dominate the whole evolution of water demand, and energy consumption.
Lower weight factor indicated that smaller food shortage awareness can be
accumulated. Feedback to increase crop area was thereby weakened. Both agriculture
water demand and food production were decreased. As energy use quota for
agricultural water supply is negligible, little response of energy consumption can be
found.
**4.4.2 Sensitivity Analysis of Water Resources Allocation Schemes on WEFS**
**Nexus**
The WEFS nexus in the study area was evidently constrained under water
resources allocation schemes with smaller water release from reservoir. The
decreasing water supply directly increased water shortage, the average annual
shortage rate of which increased from 6.41% to 8.01%. The rapid increase of water
shortage awareness then accelerated environmental awareness accumulation and
further the feedback shown in Figure 8 (h). As the negative feedback on
socioeconomic sectors was strengthened, water demand decreased rapidly in recession
phase (Figure 9 (h)). Water supply was thereby decreased with decreasing water
demand, which accounts for the decreasing energy consumption during water supply
process shown in Figure 10 (h). For food system, decreasing water release notably
altered the stability of food production evolution (Figure 11 (h)). Higher water
shortage rate led smaller food production and further larger food shortage awareness.
Feedback driven by food shortage awareness was strengthened to increase crop area.
Food production thereby increased in expansion phase. However, increasing crop area
was accompanied by increasing agricultural water demand, which brought increases
of water shortage and environmental awareness. With stronger environmental
awareness feedback, food production in recession phase thereby decreased rapidly.

To assess the impacts of water resources allocation schemes in different

operational zones, the spatial distributions of water shortage and socioeconomic
variables including water demand, energy consumption, and food production were
considered. Operational zones were classified into four types as shown in Figure 12.
The zone with small water shortage, and the water shortage rate, and socioeconomic
variables of which perform insensitivity, is defined as type A. If water shortage can be
almost removed and socioeconomic variables are sensitive, the zone is defined as type
B. If water shortage can be partly alleviated and socioeconomic variables are sensitive,
the zone is defined as type C. The zone with considerable water shortage, and the
water shortage rate, and socioeconomic variables of which perform insensitivity, is
defined as type D. Four representative zones including Z9 (Yichengmanhe) in type A,
Z1 (Fangxian) in type B, Z8 (Nanzhang) in type C, and Z13 (Jingmenzhupi) in type D
were selected to study the responses to different water resources allocation schemes.
The water shortages and socioeconomic variables are presented in Figure 13.

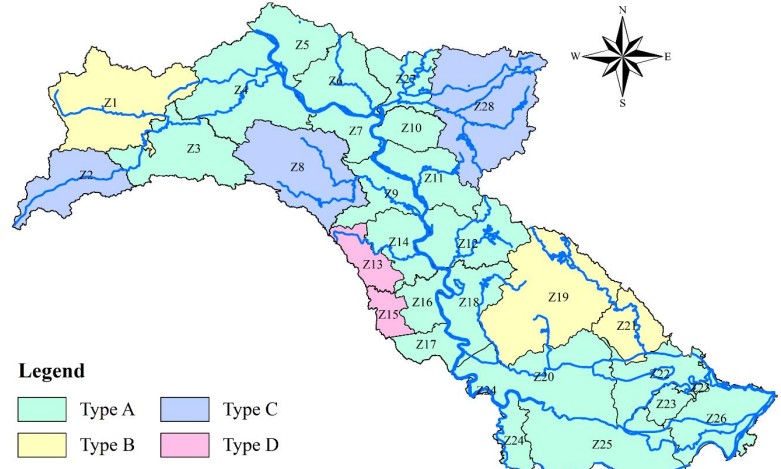


**Figure 12. Spatial distribution of A, B, C, and D types of operational zones.**

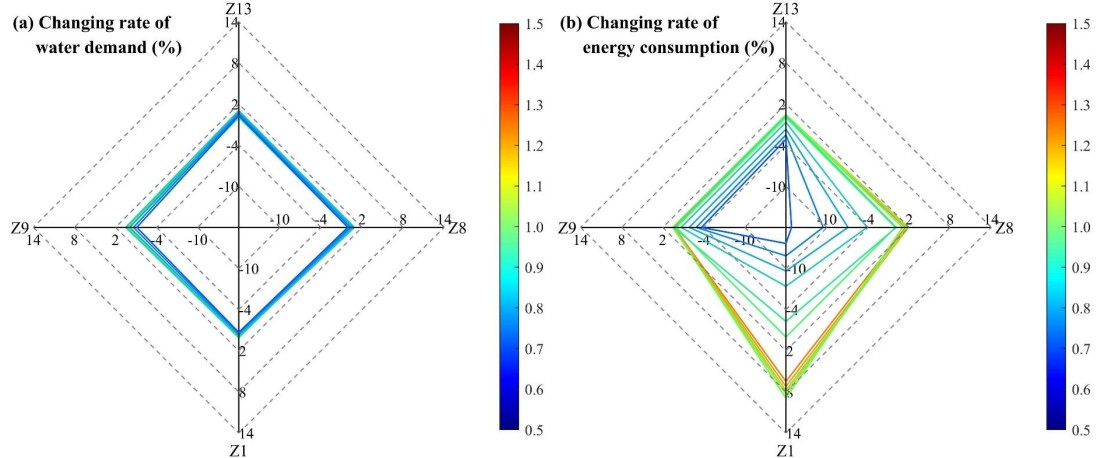


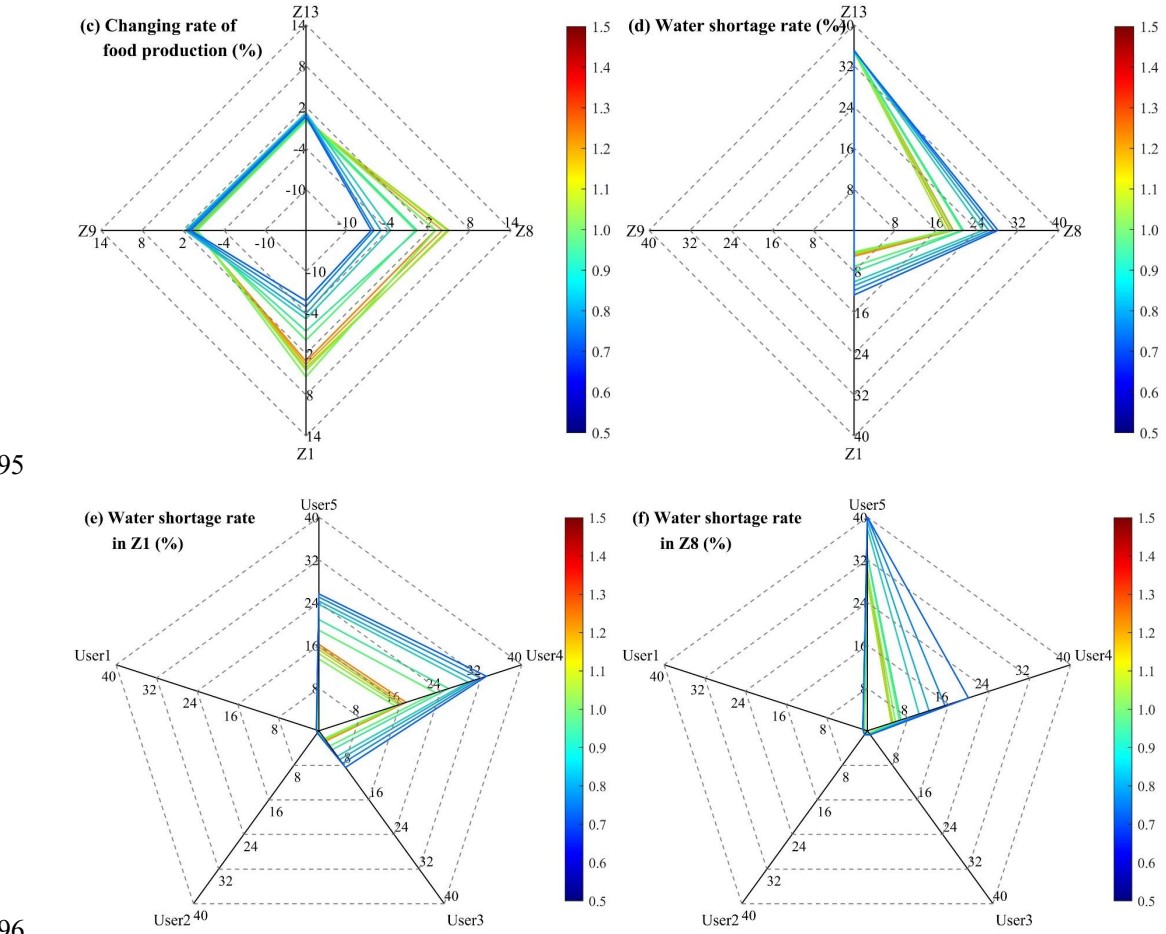



**Figure 13. Socioeconomic variables with varied reservoir release multiplier in Z9, Z1, Z8, and Z13: (a) changing rates of water demand; (b) changing rates of energy consumption; (c) changing rate of food production; (d) water shortage rates; (e) water shortage rates of water users in Z1 (user 1, 2, 3, 4, and 5 are related to municipal, rural, in-stream ecology, industrial, and agricultural users); (f) water shortage rates of water users in Z8.**

As environmental awareness feedback on population, GDP, and crop area was conducted in the entire study area, the water demand variations in Z1, Z8, Z9, and Z13 were similar, and all of them were small (Figure 13 (a)), which indicated that water supply was the primary factor affecting the integrated system.

No water shortage was observed in Z9 under different water resources allocation schemes (Figure 13 (d)), and the energy consumption, and food production also exhibited insensitivity shown in Figure 13 (b), and (c). As Z9 located along the main stream of Hanjiang river, the regulating capacity of water project was strong due to

Danjiangkou reservoir (whose total storage is 33,910 million $m^3$). Despite of the
reduction of water release, the water demand can always be ensured, and the energy
consumption, and food production thereby remained stability. Water shortage rate in
Z1 decreased evidently with the increase of water release (Figure 13 (d)), and the
energy consumption, and food production further increased remarkably, as shown in
Figure 13 (b), and (c). Z1 located at the boundary of study area, the water supply of
which mainly depended on Sanliping reservoir (shown in Figure 3). The regulating
capacity of water project was strong enough to cover most part of water demand.
Therefore, the increasing water release remarkably relived water shortage (water
shortage rate decreased from 12.56% to 4.20%), particularly in industrial and
agricultural users, as shown in Figure 13 (e). Energy consumption during water
supply process thus increased, and food production also increased owing to the
decreasing agricultural water shortage rate. Response of Z8 to water resources
allocation schemes was similar to Z1. The difference was that local reservoirs in Z8
can provide limited regulating capacity, which can only cover part of water demand.
Water shortage was effectively alleviated, but still considerable (water shortage rates
were always more than 18% shown in Figure 13(d)). Z13 was far away from the
mainstream and there was no local reservoir. The regulating capacity of water project
was so weak that no response to water resources allocation schemes was observed.
Water was always the key resource constraining the development of Z13 (Figure 13
(d)).

It's worth noting that it doesn't mean more water release from reservoir can

always promote the development of the integrated system. As shown in Figure 13 (e),

and (f), remarkable decreases of water shortage were no longer observed, since

reservoir release multiplier was more than 1.2. As excessive water release may

decrease reservoir storage in dry season, even more water shortages were found, as

shown in Figure 13 (e), and (f), which further constrained socioeconomic expansion

(Figure 13 (b), and (c)). Therefore, regulating capacity of water project is an

important factor to ensure the stability of water system to sustain WEFS nexus. In the

area equipped with strong regulating capacity of water project, water demand can

always be covered and the integrated system is not sensitive to varied water release

from reservoir. While in the area with certain regulating capacity of water project but

can not totally cover the water demand, regulating the water release from reservoir by

rational water resources allocation schemes can effectively ensure water supply and

thereby contributes to the sustainable development of the integrated system.

**5. Conclusions**

The sustainable management of the WEF nexus remains an urgent challenge, as

human sensitivity and reservoir operation are seldom considered in recent studies.

This study used environmental awareness to capture human sensitivity and

simultaneously incorporated reservoir operation in the form of water resources

allocation model (i.e., IRAS model) into water system to develop a system dynamic

model for the WEFS nexus. The proposed approach was applied to the MLHRB in

China. The conclusions drawn from the study are as follows.
The proposed approach provides a valid analytical tool for exploring the
long-term co-evolution of the nexus across the water, energy, food, and society
systems. Environmental awareness in the society system shows potential to capture
human sensitivity to shortages from water, energy, and food systems. The feedback
driven by environmental awareness can regulate the pace of socioeconomic expansion
to maintain the integrated system from constant resources shortages, which
contributes to the sustainability of the WEFS nexus. The co-evolution of water
demand, energy consumption, and food production can be divided into expansion
(accelerating and natural expansion for food production), contraction, recession, and
recovery phases based on environmental awareness. Rational parameter setting of
boundary conditions and critical values can effectively control environmental
awareness feedback to help managers to keep the socioeconomic sectors from violent
expansion and deterioration in contraction and recession phases. Water resources
allocation can effectively relieve water shortage by increasing water supply. As
high-level environmental awareness led by water shortage is remarkably alleviated,
environmental awareness feedback is weakened and the socioeconomic sectors
develop rapidly. Threats from water shortage on the concordant development of
WEFS nexus are significantly alleviated. Regulating capacity of water project is an
important factor in water resources allocation to ensure the stability of water system
to sustain WEFS nexus. Particularly for the area with certain regulating capacity of
water project but cannot totally cover the water demand, regulating the water release
from reservoir by rational water resources allocation schemes can further ensure water
supply and is of great significance for the sustainable development of the WEFS
nexus.

We acknowledge that environmental awareness feedback functionality remains

to be further improved. Indeed, environmental awareness also has potential to
contribute to socioeconomic expansion by promoting resources-saving technology.
It's the function of the level and duration of environmental awareness, and the sizes of
socioeconomic factors, which will become the focus of our further study. The model
calibration is also challenging, as the data series is not sufficiently long and the forms
and parameters of the feedback function are not prescribed. We consider that
sufficient case studies will gradually emerge over time, which could gradually cover a
range of scenarios and slowly provide reliability in the WEFS nexus modeling.
Moreover, as the primary input of the proposed WEFS nexus model, water availability
was adopted based on the historical scenario in this study. Future climate change has
not been considered for the sake of simplicity. The considerable uncertainties in water
availability can be brought into the water system in the WEFS nexus due to climate
change (Chen et al., 2011). The propagation of the uncertainties can also be
complicated, with interactions among water, energy, food, and society systems during
the co-evolution process. Therefore, more attention should be paid to the uncertainty
analysis on the WEFS nexus under climate change. However, the proposed
framework and our research results not only provide useful guidelines for local
sustainable development but also demonstrate the potential for effective application in
other basins.

**Data availability:** The socioeconomic data used in producing this paper are
available at http://data. cnki.net/

**Author contributions:** Conceptualization, DL and YZ; Methodology, YZ;
Software, YZ; Data Curation, YZ, ZW and LD; Formal analysis, YZ and DL;
Writing-Original Draft preparation, YZ and LD; Writing-Review and Editing, SG, LX,
PL, JY and DL; Funding acquisition, DL.

**Competing interests:** The authors declare that they have no conflict of interest.

**Acknowledgement:** The authors gratefully acknowledge the financial support
from the National Natural Science Foundation of China (Nos. 51879194, 91647106
and 51579183). This work is also partly funded by the Ministry of Foreign Affairs of
Denmark and administered by Danida Fellowship Centre (File number: 18-M01-DTU)
and The Open Innovation Project of Changjiang Survey Planning Design and
Research Co., Ltd. (No. CX2020K03).

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
