# Peer review of "A system dynamic model to quantify the impacts of water"

_Hydrology and Earth System Sciences, 2021_

## Author Comment (AC1)

**Reviewer 1**

This study addresses the phenomenon that the water, energy, and food crises that human society is facing are highly interconnected issues, and their evolutions would further stimulate human response actions, which would in turn (re)shape the evolution trajectories of the FEW systems. In doing so, the authors develop a holistic sociohydrologic model, which not only mimics the water, energy, and food systems but the related human components (e.g., population, GDP, industry, agriculture) are also incorporated endogenously. Overall, the work is interesting and represents a very important direction for extending the scope of sociohydrology, which has been discussed particularly by Di Baldassarre et, al, Sociohydrology: Scientific Challenges in Addressing the Sustainable Development Goals https://doi.org/10.1029/2018WR023901. In this sense, I think this manuscript is a valuable contribution to the scientific progress within the scope of sociohydrology. However, I do have some concerns and suggestions that need to be addressed, which are listed below.

Thank you very much for your positive feedback and valuable comments on our paper. We will thoroughly revise the paper based on the comments. We believe the current comment can greatly help improve the quality of the paper. Here are the responses to your comments:

1. The text and grammar should be revised throughout. There are many places (too many to be listed) where the language is unclear and misleading.
1 Response:

Thanks for your supportive suggestion. We will carefully improve the writing quality in the revised manuscript.

2. I suggest the authors give a more detailed description of figure 1. This figure is very important for understanding the overall feedback relationships between the model variables. Currently, I am not very clear about the feedback relationships.
2 Response:

Thanks for your supportive suggestion. We quite agree with your opinion. We are going to give more details for the description about the primary feedback loop driven by environmental awareness in Figure 1. Description for connection between water system and energy system as well as food system will be improved as follow: The water demands and available water resources are further inputted into water resources allocation model to determine water supply and water shortage for every water use sector in each operational zone. The water supply for these five sectors and agriculture water shortage rates as outputs from water system module are taken as the inputs of energy system module and food system module to determine the energy consumption and food production, respectively. And description for feedback driven by environment awareness will be improved as follow: As environmental awareness accumulates over its critical value, negative feedback on socioeconomic sectors (i.e., population, GDP and crop area) will be triggered to constrain the increase of water demand and further energy consumption and food production to sustain the WEFS nexus.

3. I have some concerns about equations (2)-(5). First, this seems not the Malthus growth model. In the Malthus growth model, the right side of equations 2-5 should be N, G, A, and WQ, respectively, instead of N0, etc. please check if it is a typo. Second, there is an exponential term which the authors call the technology effect, dampening the growth rate of the state variables. This is not very convincing. I believe that technology development would contribute to water conservation activities and thus reduce water use quota, but I do not understand why it would have a negative effect on GDP, population and crop area, this is somewhat counter-intuitive. Third, equation (5). Why is there a negative sign in front of WQ? From table 2, rqwu is already a negative value (i.e., -0.02). If you intend to indicate that the water use quota is decreasing over time, one negative sign needs to be removed. In addition, in this case, the exponential term would dampen the decreasing rate of water use quota. This might not be reasonable, because technology development is always supposed to accelerate the decreasing of water use quota instead of dampening it. Fourth, there is a term representing the effect of GDP on water use quota in equation (5). I assume the rationale is that GDP development would prompt the advancement of water-saving technology. But the effect of technology has already been considered by the exponential term. I think perhaps the equation (5) is over-complex. Fifth, line 155, the

authors claim that this study considers municipal and rural water consumption, industrial water consumption and agricultural water consumption, so I think there should be a distinction of water use quota for each of these types of water use. However, there seems no distinction between the different types of water use in equation (5).

3.1 Response to the first comment:

Thanks for your supportive suggestion. We are going to add the original Malthusian growth equation in revised manuscript. And the forms of equations for population, GDP and crop area will be corrected in equation (3)-(5). As socioeconomic factors in original Malthusian growth model without constraints will explode to infinity in a long-time evolution, the growth rates of population, GDP and crop area are assumed to increase with decreasing rates as time goes. And feedback functions as well as environmental capacities of socioeconomic variables are adopted to constrain the infinity evolution of these socioeconomic variables through equation (3)-(5) (Feng et al., 2016; Hritonenko and Yatsenko, 1999) The equation (4) for GDP simulation is taken as an example here:

$$\begin{cases} \dfrac{dG_t}{dt} = r_{G,t} * G_t \\[2mm] r_{G,t} = \begin{cases} r_{G,0} * (1 + \kappa_G * \exp(-\varphi_G t)) + f_2(E) & G_t \le G_{cap} \\[1mm] \mathrm{Min}(0, r_{G,0} * (1 + \kappa_G * \exp(-\varphi_G t)) + f_2(E)) & G_t > G_{cap} \end{cases} \end{cases} \tag{4}$$

where $G_t$ is the GDP in $t$th year; $GDP_{cap}$, is the environmental capacity of GDP; $r_{G,0}$ is the growth rate of GDP in baseline year, which is observed from history data; $r_{G,t}$ is the growth rates of GDP in $t$th year; $\kappa_G * \exp(-\varphi_G t)$ is used to depict the impacts of technology development on evolution of GDP; $E$ is environmental awareness; $f_2$ is the feedback function.

3.2 Response to the second comment:

Thanks for your supportive suggestion. Taking the GDP simulation as an example, the exponential term (e.g., $\exp(-\varphi_G t)$) is used to depict the impacts of technology development on GDP evolution, and further determine the growth rate of GDP. GDP is assumed to increase but with a decreasing rate, as the difficulty for increasing GDP is increasing as time goes, which can be fitly accounted by the exponential term (i.e., $\exp(-\varphi_G t)$ is non-negative and decrease over time, keeping GDP increasing with a decreasing rate).

3.3 Response to the third comment:

Thanks for your supportive comment. We will take your valuable suggestion and remove the negative sign in equation (6) for water use quota simulation. The exponential term would dampen the decreasing rate of water use quota as time goes, rather than water use quota as discussed in '3.2 Response', as the difficulty of saving water by the advances in technology is increasing over time.

$$\begin{cases} \dfrac{dWQ_{i,j}^{t}}{dt} = WQ_{i,j}^{t} * r_{qwu,t} \\ r_{qwu,t} = r_{qwu,0}(1 - \kappa_{qwu} * \exp(-\varphi_{qwu}t)) \end{cases} \tag{6}$$

where $WQ_{i,j}^{t}$ is the water use quota of $j$th water user in $i$th operational zone in $t$th year; $r_{qwu,0}$ and $r_{qwu,t}$ are the growth rates of water use quotas in baseline year and $t$th year, respectively; $\kappa_{qwu} * \exp(-\varphi_{qwu}t)$ is used to depict the water-saving effect of technology development on evolution of water use quota.

3.4 Response to the fourth comment:

Thanks for your supportive comment. We will take this valuable suggestion and remove the feedback driven by the changing rate of GDP. The model will be re-built and the results will be updated.

3.5 Response to the fifth comment:

Thanks for your supportive suggestion. We have considered the different types of water use in each operational zone for water quota use simulation. We will improve the equation for water use quota by adding subscripts to show the distinctions between the different types of water use in different operational zones.

4. The description of the water resources allocation in section 2.1.2 is too simple. I cannot understand the rationale behind equations 6 and 7. Especially, reservoir operation is an important focus of this study, I suggest the authors give some more detailed descriptions of the water resources allocation processes. Currently, it is difficult to see how the water shortage rate is calculated in equation 7.

4 Response:

Thanks for your supportive comment. We will take your valuable suggestion. More details for Interactive River-Aquifer Simulation (IRAS) water resources allocation model will be given in our manuscript.

Temporal resolutions for IRAS model will be added as follow. "IRAS model runs on a yearly loop. The year is divided into user-defined time step, and each time step is broken into user-defined sub-time-step, base on which water resources allocation conducts."

Detailed descriptions for water shortage estimation will also be added as follow. "Water shortage at demand node should be firstly determined on basis of its water demand and total water supply. The total water supply consists of natural water inflow (i.e., local water availability) and water supply from reservoir. In each sub-time-step (except the first), the average natural water inflow in previous $sts$-1 sub-time-step is estimated as the extrapolated natural water inflow in rest sub-time-steps by equation (7). The water shortage can then be determined by deducting the demand reduction, the total real-time water inflow and the extrapolated natural water inflow from water demand through equation (8). The total water shortage rate can then be determined by equation (9)."

The water shortage at demand node calls for water release from corresponding reservoir node according to their hydrological connections. The amount of water release from reservoir depends on water availability for demand-driven reservoir and operational rules for supply-driven reservoir, respectively, details of which will be given in revised manuscript.

$$WE_{i,j}^{sts} = (\sum_{1}^{sts-1} WTSup_{i,j}^{sts} - \sum_{1}^{sts-1} WRSup_{i,j}^{sts}) * \frac{(Tsts - sts + 1)}{(sts - 1)} \tag{7}$$

$$WS_{i,j}^{sts} = \frac{WD_{i,j}^{ts}(1 - f_{red}) - \sum_{1}^{sts} WTSup_{in}^{sts} - WE_{i,j}^{sts}}{Tsts - sts + 1} \tag{8}$$

$$WSR_{i,j}^{t} = \frac{\sum_{ts} \sum_{sts} WS_{i,j}^{sts}}{\sum_{ts} WD_{i,j}^{ts}} \tag{9}$$

where $ts$ is the current time step; $Tsts$ is the total number of the sub-time-step; $sts$ is the current sub-time-step; $WE_{i,j}^{sts}$ is the extrapolated natural water inflow for $j$th water use sector in $i$th operational zone; $WTSup_{i,j}^{sts}$ is the total water supply; $WRSup_{i,j}^{sts}$ is the water supply from reservoir; $WD_{i,j}^{ts}$ is the water demand; $f_{red}$ is the demand reduction

factor; $WS_{i,j}^{st}$ is the water shortage; $WSR_{i,j}^{t}$ is the water shortage rate in $t$th year.

5. Equation 8 has the same problem as equation 5, please see comment (3).

5 Response:

Thanks for your supportive suggestion. We will improve the equation for energy use quota as discussed in "3.3 Response". The negative sign and feedback driven by changing rate of GDP will be removed.

6. I am a bit confused about how energy consumption is defined in this study. In equation 9, energy consumption is calculated by multiplying water supply by energy use quota, so I assume that energy use quota is defined as the energy demand for supplying per unit of water. In this case, energy consumption in this study means the energy consumed by the water supply sectors only. However, in line 319, the authors introduce the energy consumption by the steel and petrochemical sectors. I think more clarifications are needed. In addition, would the situation of energy shortage have a negative effect on water supply? There is no energy considered in equations 2-7.

6 Response:

Thanks for your supportive comment. We will take the valuable suggestion. We will re-build the WEFS nexus model and update the results by re-defining the energy consumption in Section 2.2

We focus on the energy consumption during water supply process to further help investigate the energy co-benefits of water resources allocation schemes (Zhao et al., 2020; Smith et al., 2016). The energy consumption for water heating and water end use is not included in this study. Energy consumption is determined by energy use quota and the amount of water supply for water use sectors (Smith et al., 2016).

Constant energy shortage can lead the increase of environmental awareness. Once the environmental awareness increases over its critical value, negative feedback on socioeconomic sectors will be triggered. The water demand will thus be decreased, and further water supply will be changed.

7. Equation 11. Similar to comment (3), technology development is supposed to benefit crop yield, but the exponential term here is dampening the crop yield.

7 Response:

Thanks for your supportive suggestion. We will improve the equation for crop yield simulation as is discussed in "3.2 Response". The crop yield is assumed to increase with decreasing rate, as the difficulty of increasing crop yield is increasing over time.

8. Environmental awareness put forward by van Emmerik et al. is intended to capture human sensitivity to environmental deterioration. In this study, the authors quantify environmental awareness by water shortage, food shortage and energy shortage (i.e., equation 14). I feel food shortage and energy shortage are more like social problems rather than environmental problems. It might be better if the authors change a name for this variable.

8 Response:

Thanks for your supportive suggestion. We totally agree with your opinion that "environmental awareness" describes societal perceptions of the environmental degradation within the prevailing value systems. This study is based on the concept of "environmental awareness" proposed by Van Emmerik et al. (2014). We extend water, energy and food as part of environment, which further consists of the environmental awareness in this study. As "environmental awareness" is a popular and recognized terminology in socio-hydrology, it may be difficult for find another terminology to replace "environmental awareness".

9. Equation 18, 19 and 20 should be piecewise equations. I.e., when E is smaller than Ecrit, f(E) should be zero.

9 Response:

Thanks for your supportive suggestion. We will accomplish the piecewise equations for feedback functions.

10. Equation 21-23. If GDP would have an effect on water, food and energy systems, I think it might be more reasonable to use the magnitude of GDP instead of its changing rate.

10 Response:

Thanks for your supportive comment. We will take this valuable suggestion. As the effects of GDP on water use quota, energy use quota and crop yield have been considered by the exponential terms, the feedback function driven by the changing

rate of GDP will be removed as is discussed in "3.4 Response". We will re-build the model and update the results.

11. Section 3. Human response to the issues of water, food and energy shortages is an important aspect of the model. I suggest the authors give some observable evidences to show human adaptive response towards the mismatch between demand for and availability of water resources. for example any policy?

11 Response:

Thanks for your supportive suggestion. We will add the descriptions for human response to the issues from water, energy and food systems in Section 3.1 by citing supportive references as follow: "Due to population expansion, fast urbanization and rapid economic development, the local demands for water, energy and food are going to increase enormously (Zeng et al., 2021; Zhang et al., 2018). The contradictions between the increasing demands and limited resources will be intensified. Improving use efficiencies for water, energy and food in the mid-lower reaches of Hanjiang river basin is needed urgently (Zhang et al., 2018; Liu et al., 2019). The strictest water resources control system for water resources management policy, the total quantity control of water consumed policy and the energy-saving and emission-reduction policy in China are carried out in the mid-lower reaches of Hanjiang river basin to promote the spread of resources-saving technology and further improve the resources use efficiencies in water, energy and food systems. Therefore, impacts of human activities on WEF nexus should be assessed to sustain the collaborative development of the integrated system."

12. A more detailed description of figure 3 is needed.

12 Response:

Thanks for your supportive suggestion. More details of Figure 3 will be added in revised manuscript as follow:

The socioeconomic data (i.e., population, GDP and crop area) for water demand projection are collected based on administrative units, while the hydrological data are often collected on basis of river basins. To ensure the socioeconomic data and the hydrological data consistent in operational zones, the study area is divided into 28 operational zones based on the superimposition of administrative units and sub-basins. Based on the water connections between operational zones and river systems, study

area is sketched in Figure 3, including 2 water transfer project (South-North water transfer project and Changjiang-Hanjiang water transfer project), 17 reservoirs and 28 operational zones.

13. Table 2. These are parameters and they may need to be listed in table 3. In table 2, the authors may need to show the initial conditions of the state variables, i.e., population, GDP, crop area, etc.

13 Response:

Thanks for your supportive comment. We will take your valuable suggestion. The initial conditions so as corresponding descriptions in Table 2 will be accomplished, including population, GDP, crop area, environmental capacities and growth rates of population, GDP and crop area, water use quota, energy use quota, crop yield and their growth rates, planning energy production and planning food production.

14. Table 2 and 3 are too simple. At least the authors need to give a brief description of these parameters, as it is in table 5.

14 Response:

Thanks for your supportive suggestion. We will give more detail to improve Table 2 and 3, including the notations, descriptions, units and values for the parameters.

15. There are only ten years data (i.e., 2010-2019, in yearly time step), but there are 35 parameters that need to be calibrated, which means this is a very complicated overparameterized model. I guess most of the parameters are insensitive. Perhaps an initial sensitivity analysis is needed to screen out those insensitive parameters before conducting calibration.

15 Response:

Thanks for your supportive suggestion. We totally agree your opinion. Indeed, we took the method as mentioned in the comment to calibrate the model.

We will add more details in model calibration in revised manuscript as follow: "Initial parameter sensitivity analysis is adopted to screen out the insensitive parameter, indicating that there are 13 insensitive parameters and 21 sensitive parameters, respectively. The setting of the insensitive parameter is on basis of expert

knowledge and the work of Feng et al. (2019), which has been proved with good performance and suitability. The sensitive parameters in model are then calibrated by fitting the observed data."

16.Section 4.3. The authors explore the system sensitivity to seven parameters. I wonder why these seven parameters are chosen? Especially, all of them are threshold parameters. Are there any management implications obtained? I think it might be more informative if the sensitivities of the parameters related to human management actions are explored.

16 Response:

Thanks for your supportive suggestion. We quite agree your opinion that it's more informative if the sensitivities of the parameters related to human management actions, which indeed motivates us the parameter selection. We will add the description on the motivation for parameter selection in Section 4.3 for sensitivity analysis as follow: "As the critical values and boundary conditions of WEFS nexus are considered as vital factors for policy-makers and managers to control the integrated system so as to achieve the concordant development goals, seven parameters are selected for sensitivity analysis."

17.Table 6. I am a bit confused about how the shortage rate is calculated. In some cases, the shortage rate is derived by dividing shortage by demand, and in some cases it is not. For example, in scenario I, the shortage of rural users is 0, why the shortage rate is 0.23%?

17 Response:

Thanks for your supportive suggestion. The water shortage is 0.347 million m$^3$ (151*0.23%=0.374). And it's rounded down to 0.

Additional minor comments:

18.Line 63. The authors claim that system of systems model and agent-based model do not consider the feedbacks of integrated systems. I do not think this is true. A more appropriate literature review may be needed.

18 Response:

Thank for your supportive suggestion. We quite agree with your opinion that system of systems model and agent-based model have also considered the feedback in

solving WEF nexus. As is discussed in introduction, system dynamic model is a more appropriate and efficient tool to describe the feedback among variables, when compared with system of systems model and the agent-based model, which prefers to focus on optimization and pre-defined rules, respectively.

19.In equation 4, crop area is denoted by A, but in equation 12, it is denoted by CA. please make it consistent.

19 Response:

Thank for your supportive suggestion. The equations for crop area simulation will be improved to keep the notations consistent.

20.Line 251. The authors claim that environmental awareness proposed by van Emmerik et al. is more specific than community sensitivity. This is not the case. In fact, community sensitivity is proposed by Elshafei et al. through a more extensive literature review, and it is considered more sophisticated and is used more widely.

20 Response:

Thank for your supportive suggestion. We will improve the description on social state variable selection in Section 2.4 as follow:

"Environmental awareness describes societal perceptions of the environmental degradation within the prevailing value systems (Feng et al., 2019; Feng et al., 2016; Roobavannan et al., 2018; Van Emmerik et al., 2014). Community sensitivity indicates people's attitudes towards not only the environment control but also the environmental restoration (Chen et al., 2016; Elshafei et al., 2014; Roobavannan et al., 2018). As this study focuses on human sensitivity on environmental degradation, environmental awareness on basis of the concept in the work of Van Emmerik et al. (2014) is adopted as social state variable."

21.Figure 4. Please try not to use abbreviations in the figure. It is very difficult to read.

21 Response:

Thank for your supportive suggestion. Abbreviations in Figure 4 will be avoided.

22.I notice that in some places, the authors use the word "resilience". This is a complex concept, and as it is not the focus of this study, I suggest the authors use some simpler words.

22 Response:

Thank for your supportive suggestion. We will replace "resilience" and "resilient" with other appropriate words in the paper.

**References**

Chen, X., Wang, D., Tian, F., and Sivapalan, M.: From channelization to restoration: Sociohydrologic modeling with changing community preferences in the Kissimmee River Basin, Florida, Water Resour. Res., 52, 1227-1244, 10.1002/2015wr018194, 2016.

Elshafei, Y., Sivapalan, M., Tonts, M., and Hipsey, M. R.: A prototype framework for models of socio-hydrology: identification of key feedback loops and parameterisation approach, Hydrology and Earth System Sciences, 18, 2141-2166, 10.5194/hess-18-2141-2014, 2014.

Feng, M., Liu, P., Li, Z., Zhang, J., Liu, D., and Xiong, L.: Modeling the nexus across water supply, power generation and environment systems using the system dynamics approach: Hehuang Region, China, Journal of Hydrology, 543, 344-359, 10.1016/j.jhydrol.2016.10.011, 2016.

Feng, M., Liu, P., Guo, S., Yu, D. J., Cheng, L., Yang, G., and Xie, A.: Adapting reservoir operations to the nexus across water supply, power generation, and environment systems: An explanatory tool for policy makers, Journal of Hydrology, 574, 257-275, 10.1016/j.jhydrol.2019.04.048, 2019.

Hritonenko, N. and Yatsenko, Y.: Mathematical Modeling in Economics, Ecology and the Environment, Kluwer Academic Publishers, Dordrecht/Boston/London1999.

Liu, D., Guo, S., Liu, P., Xiong, L., Zou, H., Tian, J., Zeng, Y., Shen, Y., and Zhang, J.: Optimisation of water-energy nexus based on its diagram in cascade reservoir system, Journal of Hydrology, 569, 347-358, 10.1016/j.jhydrol.2018.12.010, 2019.

Roobavannan, M., van Emmerik, T. H. M., Elshafei, Y., Kandasamy, J., Sanderson, M. R., Vigneswaran, S., Pande, S., and Sivapalan, M.: Norms and values in sociohydrological models, Hydrology and Earth System Sciences, 22, 1337-1349, 10.5194/hess-22-1337-2018, 2018.

Smith, K., Liu, S., Liu, Y., Savic, D., Olsson, G., Chang, T., and Wu, X.: Impact of urban water supply on energy use in China: a provincial and national comparison, Mitigation and Adaptation Strategies for Global Change, 21, 1213-1233, 10.1007/s11027-015-9648-x, 2016.

Van Emmerik, T. H. M., Li, Z., Sivapalan, M., Pande, S., Kandasamy, J., Savenije, H. H. G., Chanan, A., and Vigneswaran, S.: Socio-hydrologic modeling to understand and mediate the competition for water between agriculture development and environmental health: Murrumbidgee River basin, Australia, Hydrology and Earth System Sciences, 18, 4239-4259, 10.5194/hess-18-4239-2014, 2014.

Zeng, Y., Liu, D., Guo, S., Xiong, L., Liu, P., Yin, J., Tian, J., Deng, L., and Zhang, J.: Impacts of Water Resources Allocation on Water Environmental Capacity under Climate Change, Water, 13, 10.3390/w13091187, 2021.

Zhang, P., Zhang, Y. Y., Ren, S. C., Chen, B., Luo, D., Shao, J. A., Zhang, S. H., and Li, J. S.: Trade reshapes the regional energy related mercury emissions: A case study on Hubei Province based on a multi-scale input-output analysis, Journal of Cleaner Production, 185, 75-85, 10.1016/j.jclepro.2018.03.013, 2018.

Zhao, S., Liu, Y., Liang, S., Wang, C., Smith, K., Jia, N., and Arora, M.: Effects of urban forms on energy consumption of water supply in China, Journal of Cleaner Production, 253, 10.1016/j.jclepro.2020.119960, 2020.

---

## Author Comment (AC2)

**Reviewer 2**

This manuscript presents a new approach for modeling water-energy-food nexus by incorporating social feedback loops driven by environmental awareness and a water resources allocation model into the system. It's a interesting topic for researchers in the related areas, and the proposed approach has potential application value in other basins. The manuscript is clearly organized and the study background is described comprehensively in the Introduction. However, the method is not clearly explained in some places, and there are some detailed errors in words. Below are some detailed comments:

Thank you very much for your positive remarks on our paper. We will thoroughly revise the paper based on your comments. We believe the current comment can greatly help improve the quality of the paper. Here are the responses to your comments:

1. The impact of water supply on energy consumption is related to industrial water, not ecological water or domestic water. Please clearly distinguish the impacts of different types of water supply on energy and food.

1 Response:

Thank for your supportive suggestion. We will give more details to distinguish the impacts of different types of water supply on energy and food in Section 2 as follow: The water supply for socioeconomic water use sectors outputted from water system module is taken as the input of energy system module to determine energy consumption. And the agriculture water shortage rates is taken as the input of food system module to estimate the food production.

2. In Figure 1, is the output of the water resources allocation model a total water supply or water supply of different sectors for every operational zone?

2 Response:

Thanks for your supportive suggestion. The outputs from water resources allocation model are the water supplies for different water use sectors in each operational zone. We will add more details to describe the water supply in Section 2.

3. In the energy system module, water supply not only affects energy consumption, but also energy supply, such as in thermal power, hydro-power and some other sectors. It is need to consider the impact of water supply on planning energy production.

3 Response:

Thanks for your supportive suggestion. We quite agree with your opinion that water supply also plays an important role in energy production. The energy structure in the study area involves thermal power, hydro power, wind power, solar power and biomass power, which brings a great challenge to the data collection and further the energy production simulation. Therefore, as the paper focuses on assessing the impacts of environmental awareness and water resources allocation on WEFS nexus, we simplified the energy production as the boundary conditions of the model (i.e., planning energy production).

4. Please explain why GDP will affect the change of water quota in detail and provide some references for it.

4 Response:

Thanks for your supportive suggestion. We are going to add the references indicating that community wealth, which can be indicated by GDP, is considered as the vital driving factor to promote water-saving technology in Section 2.1.1. Water use quotas are assumed to decrease with the technology advancing due to expansion economy (Blanke et al., 2007; Hsiao et al., 2007). As the difficulty of saving water by the advances in technology is increasing, the changing rate of water use quota is decreasing in equation (6) (Feng et al., 2019).

5. Line 197-202: There are several variables in the equation (6) that are not explained.

5 Response:

Thanks for your supportive suggestion. We will correct equations for water shortage determination in IRAS water resources allocation model. The detailed description for the shortage determination will also be added. Water shortage at demand node should be firstly determined on basis of its water demand and total water supply. The total water supply consists of natural water inflow (i.e., local water availability) and water supply from reservoir. In each sub-time-step (except the first),

the average natural water inflow in previous *sts*-1 sub-time-step is estimated as the extrapolated natural water inflow in rest sub-time-steps by equation (7). The water shortage can then be determined by deducting the demand reduction, the total real-time water inflow and the extrapolated natural water inflow from water demand through equation (8). The total water shortage rate can then be determined by equation (9).

$$WE_{i,j}^{sts} = (\sum_{1}^{sts-1} WTSup_{i,j}^{sts} - \sum_{1}^{sts-1} WRSup_{i,j}^{sts}) * \frac{(Tsts - sts + 1)}{(sts - 1)} \tag{7}$$

$$WS_{i,j}^{sts} = \frac{WD_{i,j}^{ts}(1 - f_{red}) - \sum_{1}^{sts} WTSup_{in}^{sts} - WE_{i,j}^{sts}}{Tsts - sts + 1} \tag{8}$$

$$WSR_{i,j}^{t} = \frac{\sum_{ts}\sum_{sts} WS_{i,j}^{sts}}{\sum_{ts} WD_{i,j}^{ts}} \tag{9}$$

where *ts* is the current time step; *Tsts* is the total number of the sub-time-step; *sts* is the current sub-time-step; $WE_{i,j}^{sts}$ is the extrapolated natural water inflow for *j*th water use sector in *i*th operational zone; $WTSup_{i,j}^{sts}$ is the total water supply; $WRSup_{i,j}^{sts}$ is the water supply from reservoir; $WD_{i,j}^{ts}$ is the water demand; $f_{red}$ is the demand reduction factor; $WS_{i,j}^{st}$ is the water shortage; $WSR_{i,j}^{t}$ is the water shortage rate in *t*th year.

6. For equation (9), why does the energy use quota of an optional zone multiplied by the water use quota of an optional zone equal total energy consumption? What is the definition of energy use quota in the paper? Please explain it.

6 Response:

Thanks for your supportive suggestion. We have carefully read your constructive comments. And we found that water supply plays a more important role in energy production, rather than consumption. Therefore, we will take this valuable suggestion. We are going to re-build the WEFS nexus model by re-defining the energy consumption in Section 2.2 and the results will be updated.

We will focus on the energy consumption during water supply process to further help investigate the energy co-benefits of water resources allocation schemes (Zhao et al., 2020; Smith et al., 2016). The energy consumption for water heating and water

end use will not be included in revised manuscript. Energy consumption is determined by energy use quota and the amount of water supply for water use sectors (Smith et al., 2016), the energy use quota of which indicates the amount of energy consumption when per unit of water is supplied. Despite the amount of energy consumption from water supply process is much smaller than the total amount of energy consumption in the study area, it's still an interesting topic to quantitatively assess the trade-offs between water supply and energy consumption under different water resources allocation schemes.

7. Line 238: the calculation formula of WSR isn't presented in the paper, please add it.
7 Response:

Thanks for your supportive suggestion. We will add the equation for water shortage rate estimation in Section 2.1.2 as is discussed in "5 Response".

8. Line 328-331: Please add references to illustrate the contradictions between the increasing demands and limited resource supply will be aggravated in the study area
8 Response:

Thanks for your supportive suggestion. We will add corresponding references to illustrate that the contradiction between demands and limited resources will be intensified in study area with the impacts of climate change and the fast socioeconomic expansion in Section 3.1. Due to population expansion, fast urbanization and rapid economic development, the local demands for water, energy and food are going to increase enormously (Zeng et al., 2021). The contradictions between the increasing demands and limited resources will be intensified. Improving use efficiencies for water, energy and food in mid-lower reaches of Hanjiang river basin is needed urgently (Zhang et al., 2018; Liu et al., 2019).

9. Are the impact of policy on water supply taken into account in the water allocation model, such as total quantity control of water consumed in the region?
9 Response:

Thanks for your supportive suggestion. Yes, we have indeed taken the "total quantity control of water consumed" policy into account in our study. As the total water demand increase over a give threshold (i.e., 20 billion $m^3$ in study area), the

reduction factor (i.e., 0.95) is used to reduce industrial and agricultural water demand. We will add the description for resources-saving policies in the study area in Section 3.1.

10. Line 358: How long is the data used for parameter calibration? Please add it.

10 Response:

Thanks for your supportive suggestion. The observed data of annual water demand, energy consumption, food production, population, GDP and crop area from 2010 to 2019 are used to calibrate the model. We will add more details for parameter calibration in Section 4.1.

11. The conclusion section is too long now, please make it conciser and highlight the key conclusions.

11 Response:

Thanks for your supportive suggestion. We will carefully improve the conclusion part to make it conciser as follow.

The proposed approach provides a valid analytical tool for exploring the long-term co-evolution of the nexus across water, energy, food and society systems. Environmental awareness in society system can effectively capture the human sensitivity to shortages from water, energy and food systems. The feedback driven by environmental awareness can regulate the socioeconomic expansion to keep the integrated system from constant resources shortage, which contributes to the sustainability of WEFS nexus co-evolution. The co-evolution of water demand, energy consumption and food production can be divided into expansion (accelerating and natural expansions for food production), contraction, recession and recovery phases based on environmental awareness. The co-evolution mode of WEFS nexus functioning strongly depends on the selection of certain parameter values. Rational parameter setting of boundary conditions and critical values is important for managers to keep the socioeconomic sectors from violent expansion and deterioration, especially in contraction and recession phases. Water shortage can be effectively relieved by the increased water supply through reservoirs operation. The high-level environmental awareness lead by water shortage is thus remarkably alleviated. As the negative feedback driven by environmental awareness is weakened, socioeconomic sectors develop rapidly. Water is thus no long the vital factor constraining the

concordant development of WEFS nexus in expansion phase. Therefore, water resources allocation is of great significance for the sustainable development of WEFS nexus.

**technical comments:**

1. Line 124-125: There is no need to use the serial numbers "(1), (2)..." here, please getting rid of them.

12 Response:

    Thanks for your supportive suggestion. We will delete the serial numbers.

2. Line 174: The sentence "...are the of population..."is lack of some words.

13 Response:

    Thanks for your supportive suggestion. We will check and correct the sentence.

3. Figure 1: "Municipal water demand" projected by population is lack of rule water demand, which needs to be added.

14 Response:

    Thanks for your supportive suggestion. We will add "Rural water demand" in Figure 1.

4. The font size of Equation (3)is not consistent with other equation

15 Response:

    Thanks for your supportive suggestion. We will correct the front size of the equation to keep it consistent with others.

5. Figure 4(i) : The text after "phase 1: "is blank.

K

16 Response:

    Thanks for your supportive suggestion. We will add it in Figure 4 (i).

6. Line 404: The word "phase"doesn't need an s.

17 Response:

    Thanks for your supportive suggestion. We will delete the "s".

**References**

Blanke, A., Rozelle, S., Lohmar, B., Wang, J., and Huang, J.: Water saving technology and saving water in China, Agric. Water Manag., 87, 139-150, 10.1016/j.agwat.2006.06.025, 2007.

Feng, M., Liu, P., Guo, S., Yu, D. J., Cheng, L., Yang, G., and Xie, A.: Adapting reservoir operations to the nexus across water supply, power generation, and environment systems: An explanatory tool for policy makers, Journal of Hydrology, 574, 257-275, 10.1016/j.jhydrol.2019.04.048, 2019.

Hsiao, T. C., Steduto, P., and Fereres, E.: A systematic and quantitative approach to improve water use efficiency in agriculture, Irrig. Sci., 25, 209-231, 10.1007/s00271-007-0063-2, 2007.

Liu, D., Guo, S., Liu, P., Xiong, L., Zou, H., Tian, J., Zeng, Y., Shen, Y., and Zhang, J.: Optimisation of water-energy nexus based on its diagram in cascade reservoir system, Journal of Hydrology, 569, 347-358, 10.1016/j.jhydrol.2018.12.010, 2019.

Smith, K., Liu, S., Liu, Y., Savic, D., Olsson, G., Chang, T., and Wu, X.: Impact of urban water supply on energy use in China: a provincial and national comparison, Mitigation and Adaptation Strategies for Global Change, 21, 1213-1233, 10.1007/s11027-015-9648-x, 2016.

Zeng, Y., Liu, D., Guo, S., Xiong, L., Liu, P., Yin, J., Tian, J., Deng, L., and Zhang, J.: Impacts of Water Resources Allocation on Water Environmental Capacity under Climate Change, Water, 13, 10.3390/w13091187, 2021.

Zhang, P., Zhang, Y. Y., Ren, S. C., Chen, B., Luo, D., Shao, J. A., Zhang, S. H., and Li, J. S.: Trade reshapes the regional energy related mercury emissions: A case study on Hubei Province based on a multi-scale input-output analysis, Journal of Cleaner Production, 185, 75-85, 10.1016/j.jclepro.2018.03.013, 2018.

Zhao, S., Liu, Y., Liang, S., Wang, C., Smith, K., Jia, N., and Arora, M.: Effects of urban forms on energy consumption of water supply in China, Journal of Cleaner Production, 253, 10.1016/j.jclepro.2020.119960, 2020.

---

## Author Comment (AC3)

**Reviewer 3**

The authors create a multi-sector system dynamics model, including environmental awareness dynamics and coupled reservoir simulation. The model simulates, among other things, water demand, energy consumption, food production, environmental awareness, and population and GDP growth. The authors apply their model to the Hanjiang river basin and discuss the model simulation results at length. They identify stages of expansion, contraction, recession, and recovery for future water and energy dynamics as well as stages of expansion and stabilization for future food dynamics. The authors conduct a one-at-a-time parameter sensitivity analysis and also show that WEFS (water-energy-food-society) outcomes are strongly impacted by the presence or absence of reservoirs.

While this work aims to contribute in two primary areas – improved understanding of the impact of (1) environmental awareness feedbacks and (2) water supply reservoirs on WEF systems – I believe the work does not achieve these contributions, for the reasons described below:

- It is not clear what exactly about the approach is new. What separates the present study from those WEF studies cited in the introduction, other than the specific context and states modelled? It seems to me that the intended novelty might be coupling a WEF "system-dynamics" model with a detailed reservoir network simulation model, though this is not made clear in the paper. The discussions of model formulation and results do little to emphasize reservoir impacts, though the title suggests that reservoir impacts are central to the paper.

Thank you very much for your insightful and constructive comments on our paper. We will thoroughly revise the paper based on the comments. We believe the current comment can greatly help improve the quality of the paper. Here are the responses to your comments:

We are going to add a new part in Section 4 to study the response of WEFS nexus to environmental awareness feedbacks by setting another two scenarios (i.e., with and without considering the environmental awareness feedbacks, respectively).

And the average annual values of socioeconomic sectors will be counted to contribute to the quantitative assessment on the impacts of environment awareness feedbacks and water resources allocation on WEFS nexus.

- **Socioeconomic model (section 2.1.1, equations (2)-(5)):**

1. The model formulation and justification overlooks well-established growth models subject to resource constraints. Why not use a logistic model for growth?

1 Response:

Thanks for your supportive suggestion. We quite agree with your opinion that logistic model is very popular in growth simulation for socioeconomic sectors. The Malthus growth model adopted in our study is also considered as an effective tool for growth simulation. However, socioeconomic factors in original Malthusian growth model without constraints will explode to infinity in a long-time evolution. Therefore, the growth rates of population, GDP and crop area are assumed to increase with decreasing rates as time goes. And feedback functions as well as environmental capacities of socioeconomic variables are adopted to constrain the infinity evolution of these socioeconomic variables through equation (3)-(5) (Feng et al., 2016; Hritonenko and Yatsenko, 1999). We will add the original Malthusian growth equation in equation (2). And the forms of equations for population, GDP and crop area will be corrected in equation (3)-(5) and interpreted in Section 2.1.2. The equation (4) for GDP simulation is taken as an example here:

$$
\begin{cases}
\dfrac{dG_t}{dt} = r_{G,t} * G_t \\
r_{G,t} = \begin{cases}
r_{G,0} * (1 + \kappa_G * \exp(-\varphi_G t)) + f_2(E) & G_t \le G_{cap} \\
\mathrm{Min}(0, r_{G,0} * (1 + \kappa_G * \exp(-\varphi_G t)) + f_2(E)) & G_t > G_{cap}
\end{cases}
\end{cases}
\tag{4}
$$

where $G_t$ is the GDP in $t$th year; $GDP_{cap}$, is the environmental capacity of GDP; $r_{G,0}$ is the growth rate of GDP in baseline year, which is observed from history data; $r_{G,t}$ is the growth rates of GDP in $t$th year; $\kappa_G * \exp(-\varphi_G t)$ is used to depict the impacts of technology development on evolution of GDP; $E$ is environmental awareness; $f_2$ is the feedback function.

2. Each of these growth rates seem likely to be as or more effected by the *actual* resource limitations (i.e., shortages) than by the "environmental awareness" of those limitations. Yet, the physical limitations are not factored into these equations.

2 Response:

Thanks for your supportive suggestion. We totally agree with your opinion that the physical limitations can affect the socioeconomic growth more quickly and directly, which is of great significance for the short-term socioeconomic growth simulation. However, the physical limitations can't describe the process that human sensitivity responds to the environmental degradation within the prevailing value systems. Therefore, we use environmental awareness to describe societal perceptions of the environmental degradation to further drive the feedback on socioeconomic sectors, which is also an informative approach for the long-term socioeconomic growth simulation.

3. I believe rates of change should be proportional to the state at time t, not the initial condition.

3 Response:

Thanks for your supportive suggestion. We totally agree with your opinion. The changing rates for these socioeconomic variables in the paper are indeed considered changing over time. We will improve the forms of equations for socioeconomic sectors as is discussed in "1 Response", to indicate the changing rate explicitly.

4. The impact of technology development is either formulated unrealistically or discussed inaccurately – current formulation/discussion implies that technology suppresses growth.

4 Response:

Thanks for your supportive suggestion. We will give more descriptions for the impacts of technology development on socioeconomic sectors expansion. The exponential terms $\exp(-\varphi_P t)$, $\exp(-\varphi_G t)$ and $\exp(-\varphi_{CA} t)$ in the equations are used to depict the impacts of technology development on the evolution of population, GDP and crop area, and further determine their growth rates. Population, GDP and crop area are assumed to increase but with decreasing rates, as the difficulty for the increases is increasing as time goes, which can be fitly accounted by the exponential term (i.e., $\exp(-\varphi t)$ is non-negative and decrease over time, keeping socioeconomic sectors increasing with decreasing rates).

5. The water quota dynamics are especially unjustified – an exponential growth/decay model seems ill-fit.

5 Response:

Thanks for your supportive suggestion. We will improve the form of equation (6) for water use quota estimation. The exponential term would dampen the decreasing rate of water use quota (rather than water use quota), as the difficulty of saving water by the advances in technology is increasing over time. We will give more descriptions for water use quota simulation in Section 2.1.1. Water use quotas are assumed to decrease with the technology advancing due to expansion economy (Blanke et al., 2007; Hsiao et al., 2007). As the difficulty of saving water by the advances in technology is increasing, the changing rate of water use quota is decreasing in equation (6) (Feng et al., 2019).

$$\begin{cases} \dfrac{dWQ_{i,j}^{t}}{dt} = WQ_{i,j}^{t} * r_{qwu,t} \\ r_{qwu,t} = r_{qwu,0}(1 - \kappa_{qwu} * \exp(-\varphi_{qwu}t)) \end{cases} \tag{6}$$

where $WQ_{i,j}^{t}$ is the water use quota of $j$th water user in $i$th operational zone in $t$th year; $r_{qwu,0}$ and $r_{qwu,t}$ are the growth rates of water use quotas in baseline year and $t$th year, respectively; $\kappa_{qwu}*\exp(-\varphi_{qwu}t)$ is used to depict the water-saving effect of technology development on evolution of water use quota.

- ## Water shortage model (section 2.1.2, equations (6)-(7)):

1. The index for summation is not declared, making the equations difficult to interpret.

6 Response:

Thanks for your supportive suggestion. We will take your valuable suggestion. We will correct the equations for water shortage determination and give more descriptions on it in Section 2.1.2.

Water shortage at demand node should be firstly determined on basis of its water demand and total water supply. The total water supply consists of natural water inflow (i.e., local water availability) and water supply from reservoir. In each sub-time-step (except the first), the average natural water inflow in previous $sts$-1 sub-time-step is estimated as the extrapolated natural water inflow in rest sub-time-steps by equation (7). The water shortage can then be determined by deducting the demand reduction,

the total real-time water inflow and the extrapolated natural water inflow from water demand through equation (8). The total water shortage rate can then be determined by equation (9).

$$WE_{i,j}^{sts} = (\sum_1^{sts-1} WTSup_{i,j}^{sts} - \sum_1^{sts-1} WRSup_{i,j}^{sts}) * \frac{(Tsts - sts + 1)}{(sts - 1)} \tag{7}$$

$$WS_{i,j}^{sts} = \frac{WD_{i,j}^{ts}(1 - f_{red}) - \sum_1^{sts} WTSup_{in}^{sts} - WE_{i,j}^{sts}}{Tsts - sts + 1} \tag{8}$$

$$WSR_{i,j}^{t} = \frac{\sum_{ts}\sum_{sts} WS_{i,j}^{sts}}{\sum_{ts} WD_{i,j}^{ts}} \tag{9}$$

where $ts$ is the current time step; $Tsts$ is the total number of the sub-time-step; $sts$ is the current sub-time-step; $WE_{i,j}^{sts}$ is the extrapolated natural water inflow for $j$th water use sector in $i$th operational zone; $WTSup_{i,j}^{sts}$ is the total water supply; $WRSup_{i,j}^{sts}$ is the water supply from reservoir; $WD_{i,j}^{ts}$ is the water demand; $f_{red}$ is the demand reduction factor; $WS_{i,j}^{st}$ is the water shortage; $WSR_{i,j}^{t}$ is the water shortage rate in $t$th year.

2. The variable definitions are inconsistent and contradictory. Wdem is said to be water demand in line 201, yet WD also appears in equation (7) and is defined as water demand. There is also a Wd variable which is never defined.

7 Response:

Thanks for your supportive suggestion. We will check and correct the variables definitions of all equations to make them clear and consistent.

3. The temporal resolutions (time step and sub time step) are not explained and are therefore confusing.

8 Response:

Thanks for your supportive suggestion. We will add the description about the temporal resolutions of IRAS water resources allocation model in Section 2.1.2. IRAS model runs on a yearly loop. The year is divided into user-defined time step, and each time step is broken into user-defined sub-time-step, base on which water resources allocation conducts. The temporal resolutions of the established WEFS nexus in the study area will also be given in Section 4. The established WEFS nexus runs on a

yearly loop. Specifically, as water resources allocation model in water system module takes a monthly time step in the study (and the sub-time-step is the default value: 1 day), the annual water supply and water shortage are firstly determined before outputted to energy system module and food system module, respectively. The annual shortage rates of water, energy and food are then used to determine environmental awareness and further the feedback."

4. The distinction between "natural" and "total" water inflow is unclear.

9 Response:

Thanks for your supportive suggestion. We will add more details to describe the water shortage estimation in water resources allocation model, as is discussed in "6 Response". Specifically, the total water supply at a demand node is composed of two parts, the natural water inflow (i.e., local water availability) and water supply from reservoir. The extrapolated natural water flow at $sts$th sub-time-step indicates that the average natural water inflow in previous $sts$-1 sub-time-steps is adopted as the natural water inflow in the rest $Ttst$-$st$+1 sub-time-steps to further estimate the water shortage.

- **Energy system and Food system modules (sections 2.3 and 2.3, equations (8)-(13));**

1. These modules apply opposite approaches, without justification. The energy module simulates energy demand and takes energy production as an input ("planning energy production"). In contrast, the food module simulates food production and takes food demand as an input (misleadingly named "planning food production"). Why not simulated food demand or energy production?

10 Response:

Thanks for your supportive suggestion. We quite agree with your opinion that energy production and food demand play an important role in WEFS nexus.

The model in the study is proposed for WEFS nexus simulation at basin-scale. However, the imports and exports of energy and food for a basin are always quite complex. For instance, the study area (i.e., the mid-lower reaches of Hanjiang river basin) is considered as an important grain producing area, occupying one of the nine major commodity grain bases in China. The local food demand can always be ensured, and most of food production is exported, the total demand of which is hard to be simulated. For energy production, the energy structure in the study area involves

thermal power, hydro power, wind power, solar power and biomass power, which brings a great challenge to the data collection. Moreover, the energy import and export of the study area is complex, as it's under the impacts of Three Gorges (the largest reservoir in China) electric power system, which indicates that the energy production is hard to be determined.

Therefore, as the paper focuses on assessing the impacts of environmental awareness and water resources allocation on WEFS nexus, we simplified the food demand and energy production as the boundary conditions of the model (i.e., planning food production and planning energy production, respectively).

2. No justification is provided for formulating energy demand as a function of water supply, as opposed to population or GDP for instance. Water supply seems like a more important factor for energy production, though energy production is not modelled.

11 Response:

Thanks for your supportive suggestion. Water supply indeed plays a more important role in energy production, rather than consumption. Therefore, we will take this valuable suggestion. We are going to re-build the WEFS nexus model by re-defining the energy consumption in Section 2.2 and the results will be updated.

We will focus on the energy consumption during water supply process to further help investigate the energy co-benefits of water resources allocation schemes (Zhao et al., 2020; Smith et al., 2016). The energy consumption for water heating and water end use will not be included in revised manuscript. Energy consumption is determined by energy use quota and the amount of water supply for water use sectors (Smith et al., 2016). Despite the amount of energy consumption from water supply process is much smaller than the total amount of energy consumption in the study area, it's still an interesting topic to quantitatively assess the trade-offs between water supply and energy consumption under different water resources allocation schemes.

3. I would think that the entire crop yield dynamics are due to technology changes (ignoring water shortage), yet technology change is offered as a single term in equation (11).

12 Response:

Thanks for your supportive comment. We will take the valuable suggestion and improve the equation (20) for crop yield (so as the water use quota and energy use

quota). We are going to re-build the model by removing the feedback driven by the changing rate of GDP. And the results will be updated.

$$\begin{cases} \dfrac{dCY_{i,j}^{t}}{dt} = CY_{i,j}^{t} * r_{pro,t} \\ r_{pro,t} = r_{pro,0} * (1 + \kappa_{pro}\, \exp(-\varphi_{pro}\, t)) \end{cases} \qquad (20)$$

where $CY_{i,j}^{t}$ is the potential crop yield of $j$th crop in $i$th operational zone in $t$th year; $r_{pro,\ 0}$ and $r_{pro,\ t}$ are the growth rates of crop yields in baseline year and $t$th year, respectively; $\kappa_{pro}*\exp(-\varphi_{pro}t)$ is used to depict the impacts of technology development on evolution of crop yield

4. From the results (Section 4, see especially Tables 2 and 5), it seems that a constant energy production and constant food demand are used to drive the model simulation. This seems unrealistic.

13 Response:

Thanks for your supportive suggestion. We quite agree with your opinion that the energy production and food demand keep changing over time. As is discussed in "10 Response", the energy production and food demand are taken as the boundary conditions of the model in our study. We have given a preliminary sensitive analysis on "planning energy production (PEP)" and "planning food production (PFP)" in Section 4.3. The results indicate that PEP and PFP are the sensitive parameters in the co-evolution of WEFS nexus.

Therefore, we think it's an interesting and important topic to taken time-varying energy production and food demand into account under different policies. However, this paper focuses on impacts of environmental awareness feedback and water resources allocation on WEFS nexus. The time-varying energy production and food demand, and so as their simulations will be taken into account in our further study.

- **Model validation (Section 4.1):**

1. The methods used to develop the observed time series are unclear. For instance, how exactly were the agricultural water demand exceedance frequencies used?

14 Response:

Thanks for your supportive suggestion. The observed time series for population, GPD, crop area, water demand, energy consumption and food production from 2010

to 2019 are collected from the yearbook of Hubei province in China (http://data.cnki.net/). The agricultural water demand depends on not only the water use quota and crop area, but also the precipitation frequency. For simplicity, four frequencies (i.e., 95%, 90%, 70%, and 50%) are used to fit the yearly precipitation frequency series. Four types of agriculture water use quotas under the four frequencies (i.e., 95%, 90%, 70%, and 50%) in the baseline year are collected for water demand projection, which will be added as initial condition setup in Table 2.

2. The observed data is not sufficient to validate the model. The observed data cover a short period during the beginning of the simulation during which all states increase approximately linearly. The effects of shortage and environmental awareness are minimal during this period (as stated by the authors in their interpretations); therefore, the observations offer no validation of the awareness dynamics or feedback. That the model matches observed dynamics under this narrow, early set of conditions does not mean that the model can reliably simulate dynamics under drastically different conditions. For instance, a model which predicts perpetual linear growth in all states would seem to match the observations equally well. Given that the data does not validate the model, the model results are only useful to the extent that the model formulation seems true-to-reality. However, little justification is given for the model formulation, and as described above, there are many problematic elements of the model formulation.

15 Response:

Thanks for your supportive suggestion. We quite agree with your opinion that the numerical calibration of the model with such short data is a challenging work. We believe that sufficient case study examples will emerge as time goes, which could cover a range of gradients, and slowly provide confidence in the WEFS nexus modelling.

Our study focuses on the human sensitivity feedback to the environment degradation, which is assumed to be composed of water shortage, energy shortage and food shortage. As the water and energy availability in the beginning of co-evolution can almost cover the demand from water and energy systems in the study area, environmental awareness indeed stays at a low level. Only the parameters for food shortage awareness and its feedback are calibrated due to the food shortage, while the parameters for environmental awareness and its feedback are poorly calibrated.

However, with the fast socioeconomic expansion, the contradictions between demand side and supply side in water, energy and food systems are going to intensify. The society system will then be more sensitive to environment degradation and seek for environment recovery by constraining socioeconomic expansion as feedback. It worth noting that the forms and parameters of feedback function are not prescribed. The forms and parameters are on basis of previous studies (Feng et al., 2019; Van Emmerik et al., 2014), which is proved with good performance and suitability, and combined with expert knowledge to keep the evolution of socioeconomic variables within rational intervals. As the feedback parameters can be used to indicate the response level of community for environment degradation (i.e., the higher level, the stronger feedback), our work can still offer technique support for managers to avoid severe environment degradation from the planning perspective.

- **Model results (Sections 4.2-4.3):**

1. Most of the discussion of the results (co-evolution of WEF system) is a text description of what is seen in the figures. The discussion does little to draw out and emphasize insights.

16 Response:

Thanks for your supportive suggestion. Based on the trajectory of environmental awareness, the co-evolution processes of water demand and energy consumption can be divided into four phases: expansion, contraction, recession and recovery. We will give more detailed discussion and emphasize the current findings. From the discussion, we found that available water and energy are the vital resources constraining the long-term concordant development of the integrated system in the study area. And more attention should be paid to the time lag of community's response to the deterioration WEFS nexus to prevent the integrated system from collapsing, especially after the fast expansion of water demand and energy consumption, which can provide useful support for policy-makers.

2. The sensitivity discussion does little to add understanding. Most interpretations of sensitivity results are vague, such that the same observations could be stated just from the variable definition and model formulation. For example, in lines 551-553, the effect of lowering the food shortage sensitivity threshold level is obvious from its definition.

17 Response:

Thanks for your supportive suggestion. We will update the figures for sensitivity analysis by replacing the black lines with colored lines and color bars so as to give a more informative sensitivity analysis for identifying the explicit variations of state variables with varying parameters. Sensitive analysis on water demand is taken as an example in Figure 6 here.

[Figure]

**Figure 6. Trajectories of water demand with varied parameters. (Figure 7, 8 and 9 for trajectories of energy consumption, food production and environmental awareness will also be updated)**

We find that the co-evolution mode of WEFS nexus functioning strongly depends on the selection of certain parameter values. Rational parameter setting of boundary conditions and critical values is of great significance for managers to keep the socioeconomic sectors from violent expansion and deterioration, especially in contraction and recession phases.

- **Impacts of reservoir system (section 4.4):**

1. The methodology applied here is unclear, what exactly does it mean that one scenario considers allocation and the other doesn't?

18 Response:

Thanks for your supportive suggestion. We will give more details to describe how the methodology applied to the mid-lower reaches of Hanjiang river basin in Section 4. The study area is divided 28 operational zones based on the administrative units and sub-basins. The socioeconomic data (i.e., population, GDP and crop area) for water demand projection are collected based on administrative units, while the hydrological data are often collected on basis of river basins. To ensure the socioeconomic data and the hydrological data consistent in operational zones, the study area is divided into 28 operational zones based on the superimposition of administrative units and sub-basins.

The time resolutions of the model in the study area will also be added to help illustrate how the methodology is applied. The established WEFS nexus runs on a yearly loop. Specifically, as water resources allocation model in water system module takes a monthly time step in the study, the annual water supply and water shortage are firstly determined before outputted to energy system module and food system module, respectively. The annual shortage rates of water, energy and food are then used to determine environmental awareness and further the feedback.

Scenario I considered water resources allocation is based on the real-world reservoir system, while scenario II removes the reservoir system from scenario I, so as to assess the impacts of water resources allocation on WEFS nexus.

2. Nonetheless, it seems that scenario I is running the model with the real-world reservoir network and scenario II is running the model with all reservoirs removed (?). If so, scenario II does not seem like a useful comparison. Is the region considering removing any or all dams in the basin?

17 Response:

Thanks for your supportive suggestion. One of the goals of our study is to assess the impacts of water resources allocation on WEFS nexus, as previous studies haven't considered water resources allocation or significantly simplified reservoirs operational rules in water resources allocation. Compared with scenario I, water resources allocation is removed in scenario II so as to assess the impacts of water resources allocation on WEFS nexus. The results indicate water resources allocation is of great significance in ensuring water supply and further sustaining the WEFS nexus from the planning perspective.

The numbers as well as the operational rules of reservoirs in the study area may change over time in the future. It's also a very interesting topic to investigate the impacts of changing reservoir system on WEFS nexus, which is a very informative study for managers from the planning perspective.

- There are language issues throughout the manuscript – most frequent were typos, poor sentence structure (lots of passive voice that creates confusion about who is the subject and what exactly they are doing), and inappropriate word choice. There are too many to list specifically.

18 Response:

Thanks for your supportive suggestion. We will carefully improve the writing quality in the revised manuscript.

**References**

Blanke, A., Rozelle, S., Lohmar, B., Wang, J., and Huang, J.: Water saving technology and saving water in China, Agric. Water Manag., 87, 139-150, 10.1016/j.agwat.2006.06.025, 2007.

Feng, M., Liu, P., Li, Z., Zhang, J., Liu, D., and Xiong, L.: Modeling the nexus across water supply, power generation and environment systems using the system dynamics approach: Hehuang Region, China, Journal of Hydrology, 543, 344-359, 10.1016/j.jhydrol.2016.10.011, 2016.

Feng, M., Liu, P., Guo, S., Yu, D. J., Cheng, L., Yang, G., and Xie, A.: Adapting reservoir operations to the nexus across water supply, power generation, and environment systems: An explanatory tool for policy makers, Journal of Hydrology, 574, 257-275, 10.1016/j.jhydrol.2019.04.048, 2019.

Hritonenko, N. and Yatsenko, Y.: Mathematical Modeling in Economics, Ecology and the Environment, Kluwer Academic Publishers, Dordrecht/Boston/London1999.

Hsiao, T. C., Steduto, P., and Fereres, E.: A systematic and quantitative approach to improve water use efficiency in agriculture, Irrig. Sci., 25, 209-231, 10.1007/s00271-007-0063-2, 2007.

Smith, K., Liu, S., Liu, Y., Savic, D., Olsson, G., Chang, T., and Wu, X.: Impact of urban water supply on energy use in China: a provincial and national comparison, Mitigation and Adaptation Strategies for Global Change, 21, 1213-1233, 10.1007/s11027-015-9648-x, 2016.

Van Emmerik, T. H. M., Li, Z., Sivapalan, M., Pande, S., Kandasamy, J., Savenije, H. H. G., Chanan, A., and Vigneswaran, S.: Socio-hydrologic modeling to understand and mediate the competition for water between agriculture development and environmental health: Murrumbidgee River basin, Australia, Hydrology and Earth System Sciences, 18, 4239-4259, 10.5194/hess-18-4239-2014, 2014.

Zhao, S., Liu, Y., Liang, S., Wang, C., Smith, K., Jia, N., and Arora, M.: Effects of urban forms on

energy consumption of water supply in China, Journal of Cleaner Production, 253, 10.1016/j.jclepro.2020.119960, 2020.

---

## Author Comment (AC4)

**Reviewer 4**

This study modeled the WEF nexus by incorporating community sensitivity and reservoirs operation, where the co-evolution behaviors of the nexus across the water, energy, food, and society (WEFS) were simulated by the system dynamic model. The proposed approach was applied to the mid-lower reaches of the Hanjiang river basin in China, and the results indicated that water resources allocation could ensure water supply through reservoirs operation and greatly decrease the water shortage rate. This study is an interesting and crucial one for improving resources management. While modeling the WEF nexus in a large watershed is a very challenging problem and difficult to validate its suitability and applicability, especially when there are only limited datasets. This study made a great effort in this direction and proposed a sophisticated methodology with some preliminary analyzed results, which is a valuable contribution to the scientific community. However, I have some concerns and suggestions, which need to be better addressed, listed as follows.

Thank you very much for your positive remarks on our paper. We will thoroughly revise the paper based on your comments. We believe the current comment can greatly help improve the quality of the paper. Here are the responses to your comments:

1. The initial conditions of external variables for the integrated system shown in Table 2 and the calibrated parameters presented in Table 3 should be explained in more details. I am curious why many parameters have the same calibrated value. How to set these values?

1 Response

Thanks for your supportive suggestion. We will add the notations, descriptions, units and values of variables and parameters in Table 2 and 3. And we will give more details for model calibration in Section 4.1. Initial parameter sensitivity analysis is adopted to screen out the insensitive parameter, indicating that there are 13 insensitive parameters and 21 sensitive parameters, respectively. The setting of the insensitive parameter is on basis of expert knowledge and the work of Feng et al. (2019) (the insensitive parameters are thereby set to 0.0856), which has been proved with good

performance and suitability. The sensitive parameters in model are then calibrated by fitting the observed data.

2. How many datasets are used for model calibration? The number of calibrated parameters used for model calibration should be discussed. How to justify the suitability and applicability of the calibrated model should be given.

2 Response

Thanks for your supportive suggestion. The observed data of annual water demand, energy consumption, food production, population, GDP and crop area from 2010 to 2019 are used to calibrate the model. We will add the details for model calibration, as is discussed in "1 Response". 21 sensitive parameters are screened out and calibrated by fitting observed data, the calibration values of which will be listed in Table 3.

We will also give more details for the suitability and applicability of the calibrated model in Section 4.1. The Nash-Sutcliffe Efficiency (NSE) coefficient and percentage bias (PBIAS) are used to calibrate the model. With the NSE more than 0.7 and the absolute value of PBIAS less than 15%, the modelling performance is considered reliable. The NSEs are always more than 0.7 and the corresponding PBIASs are within -15% to 15%, suggesting that the established model is capable to simulate co-evolution of the WEFS nexus.

3. The "Respond links" among the different variables in the WEFS nexus should be explained in much more detail. The terms of feedback functions based on previous work should further explain their suitability.

3 Response

Thanks for your supportive suggestion. The details of respond links will be added in Section 2. Description for connection between water system and energy system as well as food system will be improved as follow: The water demands and available water resources are further inputted into water resources allocation model to determine water supply and water shortage for every water use sector in each operational zone. The water supply for socioeconomic water use sectors outputted from water system module is taken as the input of energy system module to determine energy consumption. And the agriculture water shortage rates is taken as the input of food system module to estimate the food production. And description for feedback

driven by environment awareness will be improved as follow: As environmental awareness accumulates over its critical value, negative feedback on socioeconomic sectors (i.e., population, GDP and crop area) will be triggered to constrain the increases of water demand and further energy consumption and food production to sustain the WEFS nexus.

The description for the suitability of the feedback function will be added in Section 2.5. The terms of feedback functions are based on the work of Feng et al. (2019) and Van Emmerik et al. (2014), which has been proved with good performance and suitability, as it has been successfully applied to simulate the human response to environment degradation in Murrumbidgee river basin (Australia) Hehuang region (China).

4. Figure 4 shows the trajectories of population, GDP, crop area, water demand, energy consumption, food production, shortage rates for water, energy, and food, awareness for water shortage, energy shortage, and food shortage as well as environmental awareness during 2010-2070. The trajectories are the basis of the following analyses. How to get these trajectories should be given in more detail, and their suitability should be discussed?

4 Response

Thanks for your supportive suggestion. The co-evolution of WEFS nexus is conducted based on system dynamics modelling (SDM), which conducts according to the nonlinear ordinary differential equations and dynamic feedback loops as is demonstrated in Section 2.

More details about the application of the established WEFS nexus in the study area will be given in Section 4. The SDM is applied to model the WEFS nexus in the mid-lower reaches of Hanjiang river basin. The established WEFS nexus runs on a yearly loop. Specifically, as water resources allocation model in water system module takes a monthly time step in the study, the annual water supply and water shortage are firstly determined before outputted to energy system module and food system module, respectively. The annual shortage rates of water, energy and food are then used to determine environmental awareness and further the feedback.

5. How to divide the evolution of water demand and energy consumption into four phases should be given?

5 Response

Thanks for your supportive suggestion. The co-evolution processes of water demand and energy consumption can be divided into four phases (i.e., expansion, contraction, recession and recovery) based on the trajectory of environmental awareness.

We will emphasize the phase dividing rules in Section 4.2. With environmental awareness below its critical value, the negative feedback on socioeconomic sectors isn't triggered and water demand as well as energy consumption increases rapidly in the beginning of co-evolution, which is defined as expansion phase. As environmental awareness exceeds its critical value, negative feedback on socioeconomic sectors is triggered and the increases of water demand and energy consumption are constrained, which is defined as contraction phase. The environmental awareness accumulates to the maximum value and water demand as well as energy consumption goes to depress significantly, which is defined as recession phase. As environmental awareness gradually decreases below its critical value, water demand and energy consumption decrease slightly and then tend to stabilization, which is defined as recovery phase.

6. The seven controllable parameters adopted for sensitivity analysis should be discussed in more detail.

6 Response:

Thanks for your supportive suggestion. As the critical values and boundary conditions of WEFS nexus are considered as vital factors for policy-makers and managers to control the integrated system so as to achieve the concordant development goals, seven parameters are selected for sensitivity analysis. We will update the figures for sensitivity analysis by replacing the black lines with colored lines and color bars so as to give a more informative sensitivity analysis for identifying the explicit variations of state variables with varying parameters. Sensitive analysis on water demand is taken as an example in Figure 6 here.

We find that the co-evolution mode of WEFS nexus functioning strongly depends on the selection of certain parameter values. Rational parameter setting of boundary conditions and critical values is of great significance for managers to keep the socioeconomic sectors from violent expansion and deterioration, especially in contraction and recession phases.

[Figure]

**Figure 6. Trajectories of water demand with varied parameters. (Figure 7, 8 and 9 for trajectories of energy consumption, food production and environmental awareness will also be updated)**

7. The conclusion seems like a long summary of the current study. The main contribution with brief (solid) scientific findings extracted from this study might be more interesting to read and easy to learn.

7 Response:

Thanks for your supportive suggestion. We will carefully improve the conclusion part to make it conciser as follow.

The proposed approach provides a valid analytical tool for exploring the long-term co-evolution of the nexus across water, energy, food and society systems. Environmental awareness in society system can effectively capture the human sensitivity to shortages from water, energy and food systems. The feedback driven by environmental awareness can regulate the socioeconomic expansion to keep the integrated system from constant resources shortage, which contributes to the sustainability of WEFS nexus co-evolution. The co-evolution of water demand, energy consumption and food production can be divided into expansion (accelerating

and natural expansions for food production), contraction, recession and recovery phases based on environmental awareness. The co-evolution mode of WEFS nexus functioning strongly depends on the selection of certain parameter values. Rational parameter setting of boundary conditions and critical values is important for managers to keep the socioeconomic sectors from violent expansion and deterioration, especially in contraction and recession phases. Water shortage can be effectively relieved by the increased water supply through reservoirs operation. The high-level environmental awareness lead by water shortage is thus remarkably alleviated. As the negative feedback driven by environmental awareness is weakened, socioeconomic sectors develop rapidly. Water is thus no long the vital factor constraining the concordant development of WEFS nexus in expansion phase. Therefore, water resources allocation is of great significance for the sustainable development of WEFS nexus.

**References**

Feng, M., Liu, P., Guo, S., Yu, D. J., Cheng, L., Yang, G., and Xie, A.: Adapting reservoir operations to the nexus across water supply, power generation, and environment systems: An explanatory tool for policy makers, Journal of Hydrology, 574, 257-275, 10.1016/j.jhydrol.2019.04.048, 2019.

Van Emmerik, T. H. M., Li, Z., Sivapalan, M., Pande, S., Kandasamy, J., Savenije, H. H. G., Chanan, A., and Vigneswaran, S.: Socio-hydrologic modeling to understand and mediate the competition for water between agriculture development and environmental health: Murrumbidgee River basin, Australia, Hydrology and Earth System Sciences, 18, 4239-4259, 10.5194/hess-18-4239-2014, 2014.

---

## Author Response (AR1)

**Cover letter**

Dear professor Murugesu Sivapalan:

We greatly appreciate you and the reviewers for taking time to review this manuscript and provide us with constructive and valuable comments. We have addressed all comments from reviewers and the manuscript has been much improved. Our changes are marked in Blue in the revised manuscript. And our responses to the reviewers are detailed in this response-to-reviewers document submitted with the revised manuscript.

Looking forward to hearing from you.

Sincerely,

Dr. Dedi Liu
Corresponding author: Dedi Liu
Email: dediliu@whu.edu.cn

**Reviewer 1**

This study addresses the phenomenon that the water, energy, and food crises that human society is facing are highly interconnected issues, and their evolutions would further stimulate human response actions, which would in turn (re)shape the evolution trajectories of the FEW systems. In doing so, the authors develop a holistic sociohydrologic model, which not only mimics the water, energy, and food systems but the related human components (e.g., population, GDP, industry, agriculture) are also incorporated endogenously. Overall, the work is interesting and represents a very important direction for extending the scope of sociohydrology, which has been discussed particularly by Di Baldassarre et, al, Sociohydrology: Scientific Challenges in Addressing the Sustainable Development Goals https://doi.org/10.1029/2018WR023901. In this sense, I think this manuscript is a valuable contribution to the scientific progress within the scope of sociohydrology. However, I do have some concerns and suggestions that need to be addressed, which are listed below.

Thank you very much for your positive feedback and valuable comments on our paper. We have thoroughly revised the paper based on the comments. We believe the current comments have greatly helped improve the quality of the paper. Here are the responses to your comments:

We have carefully read the valuable paper in the comments "Sociohydrology: Scientific Challenges in Addressing the Sustainable Development Goals" and cited it in line 82.

1. The text and grammar should be revised throughout. There are many places (too many to be listed) where the language is unclear and misleading.

1 Response:

Thanks for your supportive suggestion. We have carefully improved the writing quality in the revised manuscript.

[Figure]

**Certificate of Elsevier Language Editing Services**

The following article was edited by Elsevier Language Editing Services:
"A system dynamic model to quantify the impacts of water
resources allocation on water-energy-food-society (WEFS) nexus"

Authored by:
**Yujie Zeng**

Date: 08-Dec-2021
Serial number: LE-227542-0C163816CFCE

2. I suggest the authors give a more detailed description of figure 1. This figure is very important for understanding the overall feedback relationships between the model variables. Currently, I am not very clear about the feedback relationships.

2 Response:

Thanks for your supportive suggestion. We agree with your opinion. We have given more details for the description about the primary feedback loop driven by environmental awareness in Figure 1. Description for connection between water system and energy system as well as food system will be improved in line 130~136 "The water demands and available water resources are further inputted into the water resources allocation model to determine the water supply and water shortage for every water use sector in each operational zone. The water supply for socioeconomic water use sectors and agricultural water shortage rates as outputs from the water system module are taken as the inputs of the energy system module and food system module to determine the energy consumption and food production, respectively."

Descriptions for feedback driven by environment awareness have been added in line 144~148. As environmental awareness accumulates over its critical value, negative feedback on socioeconomic sectors (i.e., population, GDP, and crop area)

will be triggered to constrain the increases in water demand, and further energy consumption, and food production to sustain the WEFS nexus.

3. I have some concerns about equations (2)-(5). First, this seems not the Malthus growth model. In the Malthus growth model, the right side of equations 2-5 should be N, G, A, and WQ, respectively, instead of N0, etc. please check if it is a typo. Second, there is an exponential term which the authors call the technology effect, dampening the growth rate of the state variables. This is not very convincing. I believe that technology development would contribute to water conservation activities and thus reduce water use quota, but I do not understand why it would have a negative effect on GDP, population and crop area, this is somewhat counter-intuitive. Third, equation (5). Why is there a negative sign in front of WQ? From table 2, rqwu is already a negative value (i.e., -0.02). If you intend to indicate that the water use quota is decreasing over time, one negative sign needs to be removed. In addition, in this case, the exponential term would dampen the decreasing rate of water use quota. This might not be reasonable, because technology development is always supposed to accelerate the decreasing of water use quota instead of dampening it. Fourth, there is a term representing the effect of GDP on water use quota in equation (5). I assume the rationale is that GDP development would prompt the advancement of water-saving technology. But the effect of technology has already been considered by the exponential term. I think perhaps the equation (5) is over-complex. Fifth, line 155, the authors claim that this study considers municipal and rural water consumption, industrial water consumption and agricultural water consumption, so I think there should be a distinction of water use quota for each of these types of water use. However, there seems no distinction between the different types of water use in equation (5).

3.1 Response to the first comment:

Thanks for your supportive suggestion. We have added the original Malthusian growth equation in these equations. And the forms of equations for population, GDP and crop area have been corrected in equation (2)-(4) and interpreted in line 171~178. As the growth rate in the original Malthusian growth model is adopted as a constant, socioeconomic factors will reach infinity in a long-time evolution. Therefore, we assume that population, GDP, and crop area increase with decreasing rates over time, based on previous studies (He et al., 2017; Lin et al., 2016). And feedback functions,

as well as environmental capacities of socioeconomic variables, are adopted to constrain the infinite evolution of these socioeconomic variables through equations (2)–(4) (Feng et al., 2016; Hritonenko and Yatsenko, 1999).

$$\begin{cases} \dfrac{dN_t}{dt} = r_{P,t} * N_t \\[2mm] r_{P,t} = \begin{cases} r_{P,0}*(1+\kappa_P*\exp(-\varphi_P t))+f_1(E) & N_t \leq N_{cap} \\[1mm] \mathrm{Min}(0, r_{P,0}*(1+\kappa_P*\exp(-\varphi_P t))+f_1(E)) & N_t > N_{cap} \end{cases} \end{cases} \tag{2}$$

$$\begin{cases} \dfrac{dG_t}{dt} = r_{G,t} * G_t \\[2mm] r_{G,t} = \begin{cases} r_{G,0}*(1+\kappa_G*\exp(-\varphi_G t))+f_2(E) & G_t \leq G_{cap} \\[1mm] \mathrm{Min}(0, r_{G,0}*(1+\kappa_G*\exp(-\varphi_G t))+f_2(E)) & G_t > G_{cap} \end{cases} \end{cases} \tag{3}$$

$$\begin{cases} \dfrac{dCA_t}{dt} = r_{CA,t} * CA_t \\[2mm] r_{CA,t} = \begin{cases} r_{CA,0}*(1+\kappa_{CA}*\exp(-\varphi_{CA} t))+f_3(E,FA) & CA_t \leq CA_{cap} \\[1mm] \mathrm{Min}(0, r_{CA,0}*(1+\kappa_{CA}*\exp(-\varphi_{CA} t))+f_3(E,FA)) & CA_t > CA_{cap} \end{cases} \end{cases} \tag{4}$$

where $N_t$, $G_t$, and $CA_t$ are the population, GDP, and crop area in the $t$-th year, respectively; $N_{cap}$, $G_{cap}$, and $CA_{cap}$ denote the environmental capacities of population, GDP, and crop area, respectively; $r_{P,0}$, $r_{G,0}$, and $r_{CA,0}$ represent the growth rates of population, GDP, and crop area in the baseline year, respectively, which are observed from historical data; $r_{P,t}$, $r_{G,t}$, and $r_{CA,t}$ are the growth rates of population, GDP, and crop area in the $t$-th year, respectively; $\kappa_P*\exp(-\varphi_P t)$, $\kappa_G*\exp(-\varphi_G t)$, and $\kappa_{CA}*\exp(-\varphi_{CA} t)$ are used to depict the impacts of technological development on the evolution of population, GDP, and crop area, respectively; $E$ is environmental awareness; $FA$ is food shortage awareness; and $f_1$, $f_2$, and $f_3$ represent the feedback functions.

3.2 Response to the second comment:

Thanks for your supportive suggestion. Taking equation (3) as an example. The exponential term (e.g., $\exp(-\varphi_G t)$) is used to depict the impacts of technology development on GDP evolution, and further determine the growth rate of GDP. GDP is assumed to increase but with a decreasing rate, as the difficulty for increasing GDP is increasing as time goes (He et al., 2017; Lin et al., 2016), which can be fitly accounted by the exponential term (i.e., $\exp(-\varphi_G t)$ is non-negative and decrease over time, keeping GDP increasing with a decreasing rate).

3.3 Response to the third comment:

Thanks for your supportive comment. We have taken your valuable suggestion removed the negative sign in equation (5) for water use quota simulation. The exponential term would dampen the decreasing rate of water use quota (rather than water use quota) as '3.2 Response' discussed, as the difficulty of saving water by the advances in technology is increasing over time. We have given more details for water use quota simulation in line 193~196. Water use quotas are also assumed to decrease with the technological development owing to the expansion economy (Blanke et al., 2007; Hsiao et al., 2007). As the difficulties in saving water by technological advancement are increasing, the changing rate of water use quota is decreasing in equation (5) (Feng et al., 2019).

$$\begin{cases} \dfrac{dWQ_{i,j}^{t}}{dt} = WQ_{i,j}^{t} * r_{qwu,t} \\ r_{qwu,t} = r_{qwu,0}(1 - \kappa_{qwu} * \exp(-\varphi_{qwu}t)) \end{cases} \tag{5}$$

where $WQ_{i,j}^{t}$ denotes the water use quota of the $j$-th water user in the $i$-th operational zone in the $t$-th year; $r_{qwu,0}$ and $r_{qwu,t}$ are the growth rates of water use quotas in the baseline year and $t$-th year, respectively; and $\kappa_{qwu}*\exp(-\varphi_{qwu}t)$ is used to depict the water-saving effect of technological development on the evolution of water use quota.

3.4 Response to the fourth comment:

Thanks for your supportive suggestion. We have taken this valuable suggestion and removed the feedback driven by the changing rate of GDP as is shown in Figure 1. The model has been re-built and the results have been updated.

3.5 Response to the fifth comment:

Thanks for your supportive suggestion. We have considered the different types of water use in each operational zone for water quota use simulation. We have improved equation (5) for water use quota by adding subscripts to show the distinctions between the different types of water use in different operational zones.

4. The description of the water resources allocation in section 2.1.2 is too simple. I cannot understand the rationale behind equations 6 and 7. Especially, reservoir operation is an important focus of this study, I suggest the authors give some more

detailed descriptions of the water resources allocation processes. Currently, it is difficult to see how the water shortage rate is calculated in equation 7.

4 Response:

Thanks for your supportive comment. We have taken your valuable suggestion. More details for Interactive River-Aquifer Simulation (IRAS) water resources allocation model have been added.

Temporal resolutions for IRAS model has been added in line 207~210. IRAS model runs on a yearly loop. The year is divided into user-defined time step, and each time step is broken into user-defined sub-time step, base on which water resources allocation conducts.

Detailed descriptions for water shortage estimation has been added in line 216~225. "The water shortage at the demand node should first be determined based on its water demand and total water supply. The total water supply comprises natural water inflow (i.e., local water availability) and water supply from reservoir. In each sub-time step (except the first), the average natural water inflow in the previous *sts*-1 sub-time steps is estimated as the extrapolated natural water inflow in the remaining sub-time steps using equation (6). The water shortage can then be determined by deducting the demand reduction, total real-time water inflow, and extrapolated natural water inflow from water demand using equation (7). The total water shortage rate can then be determined using equation (8)."

Details for water supply have been added in line 235~241 "The water shortage at the demand node requires water release from the corresponding reservoir nodes according to their hydrological connections. The amount of water released from the reservoir depends on the water availability for demand-driven reservoirs and operational rules for supply-driven reservoirs, respectively. The water release for the supply-driven reservoir is linearly interpolated based on Figure 2 and equations (9)–(15). Additional details on the IRAS model can be found in Matrosov et al. (2011)."

$$WE_{i,j}^{sts} = (\sum_{1}^{sts-1} WTSup_{i,j}^{sts} - \sum_{1}^{sts-1} WRSup_{i,j}^{sts}) * \frac{(Tsts - sts + 1)}{(sts - 1)} \tag{6}$$

$$WS_{i,j}^{sts} = \frac{WD_{i,j}^{ts}(1 - f_{red}) - \sum_{1}^{sts} WTSup_{in}^{sts} - WE_{i,j}^{sts}}{Tsts - sts + 1} \tag{7}$$

$$WSR_{i,j}^{t} = \frac{\sum_{ts}\sum_{sts}WS_{i,j}^{sts}}{\sum_{ts}WD_{i,j}^{ts}} \tag{8}$$

where $ts$ is the current time step; $Tsts$ denotes the total number of the sub-time steps; $sts$ is the current sub-time step; $WE_{i,j}^{sts}$ represents the extrapolated natural water inflow for the $j$-th water use sector in the $i$-th operational zone; $WTSup_{i,j}^{sts}$ is the total water supply; $WRSup_{i,j}^{sts}$ is the water supply from reservoir; $WD_{i,j}^{ts}$ is the water demand; $f_{red}$ is the demand reduction factor; $WS_{i,j}^{st}$ is the water shortage; and $WSR_{i,j}^{t}$ is the water shortage rate in the $t$-th year.

[Figure]

**Figure 2. Water release rule for supply-driven reservoir.**

$$P_{t} = (t - t_{1})/(t_{2} - t_{1}) \tag{9}$$

$$V_{max}^{t} = V_{max}^{b} * (1 - P_{t}) + V_{max}^{e} * P_{t} \tag{10}$$

$$V_{min}^{t} = V_{min}^{b} * (1 - P_{t}) + V_{min}^{e} * P_{t} \tag{11}$$

$$q_{max}^{t} = q_{max}^{b} * (1 - P_{t}) + q_{max}^{e} * P_{t} \tag{12}$$

$$q_{min}^{t} = q_{min}^{b} * (1 - P_{t}) + q_{min}^{e} * P_{t} \tag{13}$$

$$P_{v} = (V^{t} - V_{min}^{t})/(V_{max}^{t} - V_{min}^{t}) \tag{14}$$

$$q^{t} = q_{min}^{t} * (1 - P_{v}) + q_{max}^{t} * P_{v} \tag{15}$$

where $t$, $t_{1},$ and $t_{2}$ are the current time, initial time, and end time in the period, respectively; $P_{t}$ denotes the ratio of current time length to period length; $V_{max}^{t}$, $V_{min}^{t}$, $V_{max}^{b}$, $V_{min}^{b}$, $V_{max}^{e}$, and $V_{min}^{e}$ represent the maximum and minimum storages at the

current time, beginning, and ending of the period, respectively; $q_{max}^t$ , $q_{min}^t$ , $q_{max}^b$ ,

$q_{min}^b$ , $q_{max}^e$ , and $q_{min}^e$ denote the maximum and minimum releases, respectively; $P_v$

is the ratio of current storage; and $q_t$ is the current release.

5. Equation 8 has the same problem as equation 5, please see comment (3).

5 Response:

Thanks for your supportive suggestion. We have improved equation (16) for energy use quota simulation as discussed in "3.3 Response". The negative sign and feedback driven by changing rate of GDP have been removed.

$$\begin{cases} \dfrac{dEQ_{i,j}^t}{dt} = EQ_{i,j}^t * r_{e,t} \\ r_{e,t} = r_{e,0} * (1 - \kappa_e \exp(-\varphi_e t)) \end{cases} \qquad (16)$$

where $EQ_{i,j}^t$ is the energy use quotas of the $j$-th water user in the $i$-th operational zone in the $t$-th year; $r_{e,0}$ and $r_{e,t}$ denote the growth rates of energy use quotas in baseline year and the $t$-th year, respectively; $\kappa_e * \exp(-\varphi_e t)$ depicts the energy-saving effect of technological development

6. I am a bit confused about how energy consumption is defined in this study. In equation 9, energy consumption is calculated by multiplying water supply by energy use quota, so I assume that energy use quota is defined as the energy demand for supplying per unit of water. In this case, energy consumption in this study means the energy consumed by the water supply sectors only. However, in line 319, the authors introduce the energy consumption by the steel and petrochemical sectors. I think more clarifications are needed. In addition, would the situation of energy shortage have a negative effect on water supply? There is no energy considered in equations 2-7.

6 Response:

Thanks for your supportive comment. We have taken your valuable suggestion. We have re-built the WEFS nexus model by re-defining the energy consumption in Section 2.2 (i.e., line 258~263) and updated the results.

The energy system module focuses on the energy consumption during the water supply process for socioeconomic water users to further investigate the energy co-benefits of water resources allocation schemes (Zhao et al., 2020; Smith et al.,

2016). Energy consumption for water heating and water end-use was not included in this study. Energy consumption is determined by the energy use quota and amount of water supply for the water use sectors (Smith et al., 2016).

Constant energy shortage can lead the increase of environmental awareness. Once the environmental awareness increases over its critical value, negative feedback on socioeconomic sectors will be triggered. The water demand will thus be decreased, and further water supply will be changed.

7. Equation 11. Similar to comment (3), technology development is supposed to benefit crop yield, but the exponential term here is dampening the crop yield.

7 Response:

Thanks for your supportive suggestion. We have improved equation (19) for crop yield simulation as discussed in "3.2 Response". The crop yield is assumed to increase with decreasing rate, as the difficulty of increasing crop yield is increasing over time.

$$\begin{cases} \dfrac{dCY_{i,j}^{t}}{dt} = CY_{i,j}^{t} * r_{pro,t} \\ r_{pro,t} = r_{pro,0} * (1 + \kappa_{pro} \exp(-\varphi_{pro} t)) \end{cases} \quad (19)$$

where $CY_{i,j}^{t}$ is the potential crop yields of the $j$-th crop in the $i$-th operational zone in the $t$-th year; $r_{pro,0}$ and $r_{pro,t}$ are the growth rates of crop yields in baseline year and the $t$-th year, respectively; $\kappa_{pro}*\exp(-\varphi_{pro}t)$ depicts the impacts of technological development on the evolution of crop yield

8. Environmental awareness put forward by van Emmerik et al. is intended to capture human sensitivity to environmental deterioration. In this study, the authors quantify environmental awareness by water shortage, food shortage and energy shortage (i.e., equation 14). I feel food shortage and energy shortage are more like social problems rather than environmental problems. It might be better if the authors change a name for this variable.

8 Response:

Thanks for your supportive suggestion. We agree with your opinion that "environmental awareness" describes societal perceptions of the environmental degradation within the prevailing value systems. This study is based on the concept

of "environmental awareness" proposed by Van Emmerik et al. (2014). We extend water, energy and food as part of environment, which further consists of the environmental awareness in this study. As "environmental awareness" has been a popular and recognized terminology in socio-hydrology, it may be difficult for find another terminology to replace "environmental awareness".

9. Equation 18, 19 and 20 should be piecewise equations. I.e., when E is smaller than Ecrit, f(E) should be zero.

9 Response:

Thanks for your supportive suggestion. We have accomplished the piecewise equations in equations (26)-(28).

10. Equation 21-23. If GDP would have an effect on water, food and energy systems, I think it might be more reasonable to use the magnitude of GDP instead of its changing rate.

10 Response:

Thanks for your supportive comment. We have taken this valuable suggestion. As the effects of GDP on water use quota, energy use quota and crop yield have been considered by the exponential terms in equation (5), (16) and (19), the feedback function driven by the changing rate of GDP has been removed as is discussed in "3.4 Response". We have re-built the model and updated the results.

11. Section 3. Human response to the issues of water, food and energy shortages is an important aspect of the model. I suggest the authors give some observable evidences to show human adaptive response towards the mismatch between demand for and availability of water resources. for example any policy?

11 Response:

Thanks for your supportive suggestion. We have added the descriptions for human response to the issues from water, energy and food systems by citing supportive references in line 372~383. Owing to population expansion, rapid urbanization, and economic development, the local demand for water, energy, and food is increasing enormously (Zeng et al., 2021; Zhang et al., 2018). The contradictions between increasing demand and limited resources will be intensified. Therefore, improving use efficiencies for water, energy and food in the mid-lower

reaches of Hanjiang river basin is urgent (Zhang et al., 2018; Liu et al., 2019). The strictest water resources control system for water resources management policy, the total quantity control of water consumed policy, and the energy-saving and emission-reduction policy in China are implemented in the mid-lower reaches of Hanjiang river basin to promote the expansion of resource-saving technology and further improve the resource use efficiencies in water, energy, and food systems.

12. A more detailed description of figure 3 is needed.

12 Response:

Thanks for your supportive suggestion. More details of Figure 4 (i.e., the number of figures has been updated in revised manuscript) have been added in line 384~394.

The socioeconomic data (i.e., population, GDP, and crop area) for water demand projection were collected based on administrative units, whereas the hydrological data were typically collected based on river basins. To ensure that the socioeconomic and hydrological data are consistent in operational zones, the study area was divided into 28 operational zones based on the superimposition of administrative units and sub-basins. Seventeen existing medium or large size reservoirs (the total storage volume is 37.3 billion $m^3$) were considered to regulate water flows. Based on the water connections between operational zones and river systems, the study area is shown in Figure 4, including 2 water transfer projects (the South–North and Changjiang–Hanjiang water transfer projects), 17 reservoirs, and 28 operational zones.

13. Table 2. These are parameters and they may need to be listed in table 3. In table 2, the authors may need to show the initial conditions of the state variables, i.e., population, GDP, crop area, etc.

13 Response:

Thanks for your supportive comment. We have taken your valuable suggestion. The initial conditions so as corresponding descriptions in Table 2 have been accomplished, including population, GDP, crop area, environmental capacities and growth rates of population, GDP and crop area, water use quota, energy use quota, crop yield and their growth rates, planning energy availability and planning food production.

14. Table 2 and 3 are too simple. At least the authors need to give a brief description of these parameters, as it is in table 5.

14 Response:

Thanks for your supportive suggestion. We have given more details to improve Table 2 and 3 in line 426 and 448, including the notations, descriptions, units and values for the parameters.

15. There are only ten years data (i.e., 2010-2019, in yearly time step), but there are 35 parameters that need to be calibrated, which means this is a very complicated overparameterized model. I guess most of the parameters are insensitive. Perhaps an initial sensitivity analysis is needed to screen out those insensitive parameters before conducting calibration.

15 Response:

Thanks for your supportive suggestion. We agree your opinion. Indeed, we took the method as mentioned in the comment to calibrate the model.

We have added the description for the details in model calibration in line 431~436. An initial parameter sensitivity analysis was adopted to screen out the insensitive parameter, which provided distinguishing 13 insensitive and 21 sensitive parameters. The setting of the insensitive parameter was based on expert knowledge and the study of Feng et al. (2019), which has been established to have good performance and suitability. The sensitive parameters in the model were then calibrated based on expert knowledge and the observed data, and the calibrated values are presented in Table 3.

16. Section 4.3. The authors explore the system sensitivity to seven parameters. I wonder why these seven parameters are chosen? Especially, all of them are threshold parameters. Are there any management implications obtained? I think it might be more informative if the sensitivities of the parameters related to human management actions are explored.

16 Response:

Thanks for your supportive suggestion. We agree your opinion that it's more informative if the sensitivities of the parameters related to human management actions, which indeed motivates us the parameter choosing. We have added the description on the motivation for parameter selection in sensitivity analysis in line 573~576 "As the

critical values and boundary conditions of the WEFS nexus are considered vital factors for policymakers and managers to control the integrated system to achieve the concordant development goals, seven parameters were selected for sensitivity analysis (Table 5)."

17. Table 6. I am a bit confused about how the shortage rate is calculated. In some cases, the shortage rate is derived by dividing shortage by demand, and in some cases it is not. For example, in scenario I, the shortage of rural users is 0, why the shortage rate is 0.23%?

17 Response:

Thanks for your supportive suggestion. The water shortage is 0.347 million $m^3$ (151*0.23%=0.374). And it's rounded down to 0.

Additional minor comments:

18. Line 63. The authors claim that system of systems model and agent-based model do not consider the feedbacks of integrated systems. I do not think this is true. A more appropriate literature review may be needed.

18 Response:

Thank for your supportive suggestion. We agree with your opinion that system of systems model and agent-based model have also considered the feedback in solving WEF nexus. As is stated in line 68~71, system dynamic model is a more appropriate and efficient tool to describe the feedback among variables, when compared with system of systems model and the agent-based model, which prefers to focus on optimization and pre-defined rules, respectively.

19. In equation 4, crop area is denoted by A, but in equation 12, it is denoted by CA. please make it consistent.

19 Response:

Thank for your supportive suggestion. The equations for crop area simulation have been improved to keep the notations consistent.

20. Line 251. The authors claim that environmental awareness proposed by van Emmerik et al. is more specific than community sensitivity. This is not the case. In

fact, community sensitivity is proposed by Elshafei et al. through a more extensive literature review, and it is considered more sophisticated and is used more widely.

20 Response:

Thank for your supportive suggestion. We have improved the description on social state variable selection in line 303~310.

Environmental awareness describes societal perceptions of environmental degradation within the prevailing value systems (Feng et al., 2019; Feng et al., 2016; Roobavannan et al., 2018; Van Emmerik et al., 2014). Community sensitivity indicates people's attitudes towards not only the environmental control, but also the environmental restoration (Chen et al., 2016; Elshafei et al., 2014; Roobavannan et al., 2018). As this study focuses on societal perceptions on environmental degradation, environmental awareness based on the concept described in Van Emmerik et al. (2014) was adopted as the social state variable.

21.Figure 4. Please try not to use abbreviations in the figure. It is very difficult to read.

21 Response:

Thank for your supportive suggestion. Abbreviations in Figure 5 (i.e., the number of figures has been updated in revised manuscript) has been avoided.

22.I notice that in some places, the authors use the word "resilience". This is a complex concept, and as it is not the focus of this study, I suggest the authors use some simpler words.

22 Response:

Thank for your supportive suggestion. We have replaced "resilience" and "resilient" with other words in the paper.

**Reviewer 2**

This manuscript presents a new approach for modeling water-energy-food nexus by incorporating social feedback loops driven by environmental awareness and a water resources allocation model into the system. It's a interesting topic for researchers in the related areas, and the proposed approach has potential application value in other basins. The manuscript is clearly organized and the study background is described comprehensively in the Introduction. However, the method is not clearly explained in some places, and there are some detailed errors in words. Below are some detailed comments:

Thank you very much for your positive remarks on our paper. We have thoroughly revised the paper based on your comments. We believe the current comments have greatly helped improve the quality of the paper. Here are the responses to your comments:

1. The impact of water supply on energy consumption is related to industrial water, not ecological water or domestic water. Please clearly distinguish the impacts of different types of water supply on energy and food.

1 Response:

Thank for your supportive suggestion. We have given more details to distinguish the impacts of different types of water supply on energy and food in line 133~136 and Figure 1. The water supply for socioeconomic water use sectors and agricultural water shortage rates as outputs from the water system module are taken as the inputs of the energy system module and food system module to determine the energy consumption and food production, respectively.

[Figure]

**Figure 1. Structure of WEFS nexus model and its feedbacks.**

2. In Figure 1, is the output of the water resources allocation model a total water supply or water supply of different sectors for every operational zone?

2 Response:

Thanks for your supportive suggestion. The outputs from water resources allocation model are the water supplies for different water use sectors in each operational zone. We have added the details to describe Figure 1 in line 130~132. The water demands and available water resources are further inputted into the water resources allocation model to determine the water supply and water shortage for every water use sector in each operational zone.

3. In the energy system module, water supply not only affects energy consumption, but also energy supply, such as in thermal power, hydro-power and some other sectors. It is need to consider the impact of water supply on planning energy production.

3 Response:

Thanks for your supportive suggestion. We agree with your opinion that water supply also plays an important role in energy production. The energy structure in the study area involves thermal power, hydro power, wind power, solar power and biomass power, which brings a great challenge to the data collection and further the energy production simulation. Therefore, as the paper focuses on assessing the impacts of environmental awareness and water resources allocation on WEFS nexus, we simplified the energy production as the boundary condition of the model (i.e., planning energy availability).

4. Please explain why GDP will affect the change of water quota in detail and provide some references for it.

4 Response:

Thanks for your supportive suggestion. We have added the supportive references indicating that community wealth, which can be indicated by GDP, is considered as the vital driving factor to promote water-saving technology in line 193~196. Water use quotas are assumed to decrease with the technological development owing to the expansion economy (Blanke et al., 2007; Hsiao et al., 2007). As the difficulties in saving water by technological advancement are increasing, the changing rate of water use quota is decreasing in equation (5) (Feng et al., 2019).

$$\begin{cases} \dfrac{dWQ_{i,j}^{t}}{dt} = WQ_{i,j}^{t} * r_{qwu,t} \\ r_{qwu,t} = r_{qwu,0}(1 - \kappa_{qwu} * \exp(-\varphi_{qwu} t)) \end{cases} \tag{5}$$

where $WQ_{i,j}^{t}$ denotes the water use quota of the $j$-th water user in the $i$-th operational zone in the $t$-th year; $r_{qwu,0}$ and $r_{qwu,t}$ are the growth rates of water use quotas in the baseline year and $t$-th year, respectively; and $\kappa_{qwu}*\exp(-\varphi_{qwu}t)$ is used to depict the water-saving effect of technological development on the evolution of water use quota.

5. Line 197-202: There are several variables in the equation (6) that are not explained.

5 Response:

Thanks for your supportive suggestion. We have corrected equations (6)-(8) for water shortage determination in IRAS water resources allocation model. The detailed

description for the shortage determination is also added in line 216~225. The water shortage at the demand node should first be determined based on its water demand and total water supply. The total water supply comprises natural water inflow (i.e., local water availability) and water supply from reservoir. In each sub-time step (except the first), the average natural water inflow in the previous $sts$-1 sub-time steps is estimated as the extrapolated natural water inflow in the remaining sub-time steps using equation (6). The water shortage can then be determined by deducting the demand reduction, total real-time water inflow, and extrapolated natural water inflow from water demand using equation (7). The total water shortage rate can then be determined using equation (8).

$$WE_{i,j}^{sts} = (\sum_1^{sts-1} WTSup_{i,j}^{sts} - \sum_1^{sts-1} WRSup_{i,j}^{sts}) * \frac{(Tsts - sts + 1)}{(sts - 1)} \tag{6}$$

$$WS_{i,j}^{sts} = \frac{WD_{i,j}^{ts}(1 - f_{red}) - \sum_1^{sts} WTSup_{in}^{sts} - WE_{i,j}^{sts}}{Tsts - sts + 1} \tag{7}$$

$$WSR_{i,j}^t = \frac{\sum_{ts} \sum_{sts} WS_{i,j}^{sts}}{\sum_{ts} WD_{i,j}^{ts}} \tag{8}$$

where $ts$ is the current time step; $Tsts$ denotes the total number of the sub-time steps; $sts$ is the current sub-time step; $WE_{i,j}^{sts}$ represents the extrapolated natural water inflow for the $j$-th water use sector in the $i$-th operational zone; $WTSup_{i,j}^{sts}$ is the total water supply; $WRSup_{i,j}^{sts}$ is the water supply from reservoir; $WD_{i,j}^{ts}$ is the water demand; $f_{red}$ is the demand reduction factor; $WS_{i,j}^{st}$ is the water shortage; and $WSR_{i,j}^t$ is the water shortage rate in the $t$-th year.

6. For equation (9), why does the energy use quota of an optional zone multiplied by the water use quota of an optional zone equal total energy consumption? What is the definition of energy use quota in the paper? Please explain it.

6 Response:

Thanks for your supportive suggestion. We have carefully read your constructive comments. And we find that it's inappropriate to determine the energy end use based on water use process. Therefore, we have taken this valuable suggestion. We have

re-built the WEFS nexus model by re-defining the energy consumption in line 258~263 and the results have been updated.

The energy system module focuses on the energy consumption during the water supply process for socioeconomic water users to further investigate the energy co-benefits of water resources allocation schemes (Zhao et al., 2020; Smith et al., 2016). Energy consumption for water heating and water end-use was not included in this study. Energy consumption is determined by the energy use quota and amount of water supply for the water use sectors (Smith et al., 2016), the energy use quota of which indicates the amount of energy consumption when per unit of water is supplied.

Despite the amount of energy consumption from water supply process is much smaller than the total amount of energy consumption in the study area, it's still an interesting topic to quantitatively assess the trade-offs between water supply and energy consumption under different water resources allocation schemes.

7. Line 238: the calculation formula of WSR isn't presented in the paper, please add it.

7 Response:

Thanks for your supportive suggestion. We have added the equation (8) to determine water shortage rate in line 228 as is discussed in "5 Response".

8. Line 328-331: Please add references to illustrate the contradictions between the increasing demands and limited resource supply will be aggravated in the study area

8 Response:

Thanks for your supportive suggestion. We have added references to illustrate that the contradiction between demands and limited resources will be intensified in study area with the impacts of climate change and the fast socioeconomic expansion in line 372~376. Owing to population expansion, rapid urbanization, and economic development, the local demand for water, energy, and food is increasing enormously (Zeng et al., 2021; Zhang et al., 2018). The contradictions between increasing demand and limited resources will be intensified. Therefore, improving use efficiencies for water, energy and food in mid-lower reaches of Hanjiang river basin is urgent (Zhang et al., 2018; Liu et al., 2019).

9. Are the impact of policy on water supply taken into account in the water allocation model, such as total quantity control of water consumed in the region?

9 Response:

Thanks for your supportive suggestion. We have indeed taken the corresponding policies for WEFS nexus. We have added the description for the resources management policies in the study area in line 376~381.

The strictest water resources control system for water resources management policy, the total quantity control of water consumed policy, and the energy-saving and emission-reduction policy in China are implemented in the mid-lower reaches of Hanjiang river basin to promote the expansion of resource-saving technology and further improve the resource use efficiencies in water, energy, and food systems.

10. Line 358: How long is the data used for parameter calibration? Please add it.

10 Response:

Thanks for your supportive suggestion. The observed data of annual water demand, energy consumption, food production, population, GDP and crop area from 2010 to 2019 are used to calibrate the model as is shown in line 440~443.

11. The conclusion section is too long now, please make it conciser and highlight the key conclusions.

11 Response:

Thanks for your supportive suggestion. We have carefully improved the conclusion part in line 807~827.

The proposed approach provides a valid analytical tool for exploring the long-term co-evolution of the nexus across the water, energy, food, and society systems. Environmental awareness in the society system can effectively capture human sensitivity to shortages from water, energy, and food systems. The feedback caused by environmental awareness can regulate the pace of socioeconomic expansion to maintain the integrated system from constant resources shortages, which contributes to the sustainability of the WEFS nexus co-evolution. The co-evolution of water demand, energy consumption, and food production can be divided into expansion (accelerating and natural expansion for food production), contraction, recession, and recovery phases based on environmental awareness. The co-evolution mode of the WEFS nexus functioning strongly depends on the selection of certain

parameter values. The rational parameter setting of boundary conditions and critical values is important for managers to keep the socioeconomic sectors from violent expansion and deterioration, particularly in contraction and recession phases. Water shortage can be effectively relieved by the increased water supply through reservoir operation. Thus, the high-level environmental awareness caused by water shortage is remarkably alleviated. As negative feedback due to environmental awareness is weakened, the socioeconomic sectors develop rapidly. Water is no longer the vital factor constraining the concordant development of the WEFS nexus in the expansion phase. Therefore, water resources allocation is of great significance for the sustainable development of the WEFS nexus.

**technical comments:**

1. Line 124-125: There is no need to use the serial numbers "(1), (2)..." here, please getting rid of them.

12 Response:

Thanks for your supportive suggestion. We have deleted the serial numbers in line 126~127.

2. Line 174: The sentence "...are the of population..." is lack of some words.

13 Response:

Thanks for your supportive suggestion. We have corrected the sentence.

3. Figure 1: "Municipal water demand" projected by population is lack of rule water demand, which needs to be added.

14 Response:

Thanks for your supportive suggestion. We have added "Rural water demand" in Figure 1.

4. The font size of Equation (3) is not consistent with other equation

15 Response:

Thanks for your supportive suggestion. We have corrected the front size of the equation to keep it consistent with others.

5. Figure 4(i) : The text after "phase 1: " is blank.

16 Response:

Thanks for your supportive suggestion. We have corrected it in Figure 5 (i) (i.e., the number of figures has been updated in revised manuscript).

6. Line 404: The word "phase"doesn't need an s.

17 Response:

Thanks for your supportive suggestion. We have deleted the "s".

**Reviewer 3**

The authors create a multi-sector system dynamics model, including environmental awareness dynamics and coupled reservoir simulation. The model simulates, among other things, water demand, energy consumption, food production, environmental awareness, and population and GDP growth. The authors apply their model to the Hanjiang river basin and discuss the model simulation results at length. They identify stages of expansion, contraction, recession, and recovery for future water and energy dynamics as well as stages of expansion and stabilization for future food dynamics. The authors conduct a one-at-a-time parameter sensitivity analysis and also show that WEFS (water-energy-food-society) outcomes are strongly impacted by the presence or absence of reservoirs.

While this work aims to contribute in two primary areas – improved understanding of the impact of (1) environmental awareness feedbacks and (2) water supply reservoirs on WEF systems – I believe the work does not achieve these contributions, for the reasons described below:

- It is not clear what exactly about the approach is new. What separates the present study from those WEF studies cited in the introduction, other than the specific context and states modelled? It seems to me that the intended novelty might be coupling a WEF "system-dynamics" model with a detailed reservoir network simulation model, though this is not made clear in the paper. The discussions of model formulation and results do little to emphasize reservoir impacts, though the title suggests that reservoir impacts are central to the paper.

Thank you very much for your insightful and constructive comments on our paper. We have thoroughly revised the paper based on the comments. We believe the current comments have greatly helped improve the quality of the paper. Here are the responses to your comments:

To further assess the impacts of environmental awareness feedbacks and water resources allocation on WEFS nexus, (1) a new Section 4.4.1 have been added to study the response of WEFS nexus to environmental awareness feedbacks by setting another two scenarios in Table 6 and (2) the average values of socioeconomic sectors

have been counted in Table 7 to contribute to the quantitative assessment. The results have been updated in Figure 10.

[revised manuscript text omitted]

We agree with your opinion that logistic model is very popular in growth simulation for socioeconomic sectors. However, socioeconomic variables will always prone to approach their environmental capacity values in logistic model, which makes it harder to distinguish the impacts of environmental awareness feedback on socioeconomic sectors. We assume the growth rate increase with decreasing rate, which is based on the previous studies (He et al., 2017; Lin et al., 2016). The socioeconomic variables thereby keep increasing, the decreases of which are explicitly led by environmental awareness feedback (As is shown in Figure 5 in line 470, population, GDP and crop area have been decreased by high-level environmental awareness).

2. Each of these growth rates seem likely to be as or more effected by the *actual* resource limitations (i.e., shortages) than by the "environmental awareness" of those limitations. Yet, the physical limitations are not factored into these equations.

2 Response:

Thanks for your supportive suggestion. We agree with your opinion that the physical limitations can affect the socioeconomic growth more quickly and directly,

which is of great significance for the short-term socioeconomic growth simulation. However, the physical limitations can't describe the process that human sensitivity responds to the environmental degradation within the prevailing value systems. Therefore, we used environmental awareness to describe societal perceptions of the environmental degradation to further drive the feedback on socioeconomic sectors, which is also an informative approach for the long-term socioeconomic growth simulation.

3. I believe rates of change should be proportional to the state at time t, not the initial condition.

3 Response:

Thanks for your supportive suggestion. We agree with your opinion. The changing rates for these socioeconomic variables in the paper are indeed considered changing over time. We have improved the forms of equation (2), (3), (4), (5), (16) and (19) for population, GDP, crop area, water use quota, energy use quota and crop yield as is discussed in "1 Response", respectively, to indicate the changing rate explicitly.

4. The impact of technology development is either formulated unrealistically or discussed inaccurately – current formulation/discussion implies that technology suppresses growth.

4 Response:

Thanks for your supportive suggestion. That technology development will promote the growth of socioeconomic sectors, but with decreasing rate as is discussed in "1 Response". The exponential terms $\exp(-\varphi_P t)$, $\exp(-\varphi_G t)$ and $\exp(-\varphi_{CA} t)$ in equations (2), (3) and (4) are used to depict the impacts of technology development on the evolution of population, GDP and crop area, and further determine their growth rates. Population, GDP and crop area are assumed to increase but with decreasing rates, as the difficulty for the increases is increasing as time goes (He et al., 2017; Lin et al., 2016), which can be fitly accounted by the exponential term (i.e., $\exp(-\varphi t)$ is non-negative and decrease over time, keeping socioeconomic sectors increasing with a decreasing rate).

5. The water quota dynamics are especially unjustified – an exponential growth/decay model seems ill-fit.

5 Response:

Thanks for your supportive suggestion. We have improved the form of equation (5) for water use quota estimation. The exponential term would dampen the decreasing rate of water use quota (rather than water use quota), as the difficulty of saving water by the advances in technology is increasing over time. We have given more details for water use quota simulation in line 193~196 "Water use quotas are also assumed to decrease with the technological development owing to the expansion economy (Blanke et al., 2007; Hsiao et al., 2007). As the difficulties in saving water by technological advancement are increasing, the changing rate of water use quota is decreasing in equation (5) (Feng et al., 2019)."

$$\begin{cases} \dfrac{dWQ_{i,j}^{t}}{dt} = WQ_{i,j}^{t} * r_{qwu,t} \\ r_{qwu,t} = r_{qwu,0}(1 - \kappa_{qwu} * \exp(-\varphi_{qwu} t)) \end{cases} \tag{5}$$

where $WQ_{i,j}^{t}$ denotes the water use quota of the $j$-th water user in the $i$-th operational zone in the $t$-th year; $r_{qwu,0}$ and $r_{qwu,t}$ are the growth rates of water use quotas in the baseline year and $t$-th year, respectively; and $\kappa_{qwu}*\exp(-\varphi_{qwu}t)$ is used to depict the water-saving effect of technological development on the evolution of water use quota.

- **Water shortage model (section 2.1.2, equations (6)-(7)):**

1. The index for summation is not declared, making the equations difficult to interpret.

6 Response:

Thanks for your supportive suggestion. We have taken your valuable suggestion. Detailed descriptions for water shortage determination have been added in line 216~225. The water shortage at the demand node should first be determined based on its water demand and total water supply. The total water supply comprises natural water inflow (i.e., local water availability) and water supply from reservoir. In each sub-time step (except the first), the average natural water inflow in the previous $sts$-1 sub-time steps is estimated as the extrapolated natural water inflow in the remaining sub-time steps using equation (6). The water shortage can then be determined by deducting the demand reduction, total real-time water inflow, and extrapolated natural

water inflow from water demand using equation (7). The total water shortage rate can then be determined using equation (8).

$$WE_{i,j}^{sts} = (\sum_{1}^{sts-1} WTSup_{i,j}^{sts} - \sum_{1}^{sts-1} WRSup_{i,j}^{sts}) * \frac{(Tsts - sts + 1)}{(sts - 1)} \tag{6}$$

$$WS_{i,j}^{sts} = \frac{WD_{i,j}^{ts}(1 - f_{red}) - \sum_{1}^{sts} WTSup_{in}^{sts} - WE_{i,j}^{sts}}{Tsts - sts + 1} \tag{7}$$

$$WSR_{i,j}^{t} = \frac{\sum_{ts}\sum_{sts} WS_{i,j}^{sts}}{\sum_{ts} WD_{i,j}^{ts}} \tag{8}$$

where $ts$ is the current time step; $Tsts$ denotes the total number of the sub-time steps; $sts$ is the current sub-time step; $WE_{i,j}^{sts}$ represents the extrapolated natural water inflow for the $j$-th water use sector in the $i$-th operational zone; $WTSup_{i,j}^{sts}$ is the total water supply; $WRSup_{i,j}^{sts}$ is the water supply from reservoir; $WD_{i,j}^{ts}$ is the water demand; $f_{red}$ is the demand reduction factor; $WS_{i,j}^{st}$ is the water shortage; and $WSR_{i,j}^{t}$ is the water shortage rate in the $t$-th year.

2. The variable definitions are inconsistent and contradictory. Wdem is said to be water demand in line 201, yet WD also appears in equation (7) and is defined as water demand. There is also a Wd variable which is never defined.

7 Response:

Thanks for your supportive suggestion. We have checked and corrected the variables definitions of all equations to make them clear and consistent.

3. The temporal resolutions (time step and sub time step) are not explained and are therefore confusing.

8 Response:

Thanks for your supportive suggestion. We have added the description about the temporal resolutions of IRAS water resources allocation model in line 207~210. The IRAS model runs on a yearly loop. The year is divided into user-defined time step, and each time step is broken into user-defined sub-time step, based on which water resources allocation conducts.

The temporal resolutions of the established WEFS nexus in the study area have also been added in line 413~419. The established WEFS nexus ran on a yearly loop. Specifically, as the water resources allocation model in the water system module took a monthly time step in the study (and the sub-time step was the default value: 1 day), the annual water supply and water shortage were first determined before being output to the energy system and food system modules, respectively. The annual shortage rates of water, energy, and food were then used to determine environmental awareness and further the feedback.

4. The distinction between "natural" and "total" water inflow is unclear.

9 Response:

Thanks for your supportive suggestion. We have added more details to describe the water shortage estimation in water resources allocation model, as is discussed in "6 Response". Specifically, the total water supply comprises natural water inflow (i.e., local water availability) and water supply from reservoir. In each sub-time step (except the first), the average natural water inflow in the previous $sts$-1 sub-time steps is estimated as the extrapolated natural water inflow in the remaining sub-time steps to further estimate the water shortage.

- **Energy system and Food system modules (sections 2.3 and 2.3, equations (8)-(13));**

1. These modules apply opposite approaches, without justification. The energy module simulates energy demand and takes energy production as an input ("planning energy production"). In contrast, the food module simulates food production and takes food demand as an input (misleadingly named "planning food production"). Why not simulated food demand or energy production?

10 Response:

Thanks for your supportive suggestion. In the study, the "planning energy production" indicates the available energy, while the "planning food production" indicate the target production. We have taken your valuable suggestion to replace "planning energy production" with "planning energy availability" to avoid the misleading.

We agree with your opinion that energy production and food production play an important role in WEFS nexus. The model in the study is proposed for WEFS nexus

simulation at basin-scale. However, the imports and exports of energy and food for a basin are always quite complex. For instance, the study area (i.e., the mid-lower reaches of Hanjiang river basin) is considered as an important grain producing area, occupying one of the nine major commodity grain bases in China. The local food demand can always be ensured, and most of food production is exported, the total demand of which is hard to be simulated. For energy production, the energy structure in the study area involves thermal power, hydro power, wind power, solar power and biomass power, which brings a great challenge to the data collection. As the energy import and export of the study area is also quite complex, the energy production is hard to be determined.

Therefore, as the paper focuses on assessing the impacts of environmental awareness and water resources allocation on WEFS nexus, we simplified the food demand and energy production as the boundary conditions of the model (i.e., planning food production and planning energy availability, respectively).

2. No justification is provided for formulating energy demand as a function of water supply, as opposed to population or GDP for instance. Water supply seems like a more important factor for energy production, though energy production is not modelled.

11 Response:

Thanks for your supportive suggestion. We have carefully read your comment and found that water supply indeed plays a more important role in energy production, rather than consumption. Therefore, we have taken this valuable suggestion. We have re-built the WEFS nexus model by re-defining the energy consumption in Section 2.2 (i.e., line 258~263) and updated all the results.

We focus on the energy consumption during the water supply process for socioeconomic water users to further investigate the energy co-benefits of water resources allocation schemes (Zhao et al., 2020; Smith et al., 2016). Energy consumption for water heating and water end-use was not included in this study. Energy consumption is determined by the energy use quota and amount of water supply for the water use sectors (Smith et al., 2016).

Despite the amount of energy consumption from water supply process is much smaller than the total amount of energy consumption in the study area, it's still an interesting topic to quantitatively assess the trade-offs between water supply and energy consumption under different water resources allocation schemes.

3. I would think that the entire crop yield dynamics are due to technology changes (ignoring water shortage), yet technology change is offered as a single term in equation (11).

12 Response:

Thanks for your supportive comment. We have taken your valuable suggestion and improved equation (19) for crop yield (so as the water use quota and energy use quota). The model has been re-built by removing the feedback driven by the changing rate of GDP, and the results have been updated.

$$\begin{cases} \dfrac{dCY_{i,j}^{t}}{dt} = CY_{i,j}^{t} * r_{pro,t} \\ r_{pro,t} = r_{pro,0} * (1 + \kappa_{pro} \exp(-\varphi_{pro} t)) \end{cases} \quad (19)$$

where $CY_{i,j}^{t}$ is the potential crop yields of the $j$-th crop in the $i$-th operational zone in the $t$-th year; $r_{pro,0}$ and $r_{pro,t}$ are the growth rates of crop yields in baseline year and the $t$-th year, respectively; $\kappa_{pro}*\exp(-\varphi_{pro}t)$ depicts the impacts of technological development on the evolution of crop yield

4. From the results (Section 4, see especially Tables 2 and 5), it seems that a constant energy production and constant food demand are used to drive the model simulation. This seems unrealistic.

13 Response:

Thanks for your supportive suggestion. We agree with your opinion that the energy availability and food demand keep changing over time. As is discussed in "10 Response", the energy availability and food demand are taken as the boundary conditions of the model in our study. We have given a preliminary sensitive analysis on "planning energy availability (PEA)" and "planning food production (PFP)" in Section 4.3. The results indicate that PEA and PFP are the sensitive parameters in the co-evolution of WEFS nexus.

Therefore, we think it's an interesting and important topic to take time-varying energy availability and food demand into account under different policies. However, this paper focuses on impacts of environmental awareness feedback and water resources allocation on WEFS nexus. The time-varying energy production and food demand, and so as their simulations will be taken into account in our further study.

- **Model validation (Section 4.1):**

1. The methods used to develop the observed time series are unclear. For instance, how exactly were the agricultural water demand exceedance frequencies used?

14 Response:

Thanks for your supportive suggestion. The observed time series for population, GPD, crop area, water demand, energy consumption and food production from 2010 to 2019 are collected from the yearbook of Hubei province in China (http://data. cnki.net/). The agricultural water demand depends on not only the water use quota and crop area, but also the precipitation frequency. For simplicity, four frequencies (i.e., 95%, 90%, 70%, and 50%) are used to fit the yearly precipitation frequency series. Four types of agriculture water use quotas under the four frequencies (i.e., 95%, 90%, 70%, and 50%) in the baseline year are collected for water demand projection, which has been added as initial condition setup in Table 2.

2. The observed data is not sufficient to validate the model. The observed data cover a short period during the beginning of the simulation during which all states increase approximately linearly. The effects of shortage and environmental awareness are minimal during this period (as stated by the authors in their interpretations); therefore, the observations offer no validation of the awareness dynamics or feedback. That the model matches observed dynamics under this narrow, early set of conditions does not mean that the model can reliably simulate dynamics under drastically different conditions. For instance, a model which predicts perpetual linear growth in all states would seem to match the observations equally well. Given that the data does not validate the model, the model results are only useful to the extent that the model formulation seems true-to-reality. However, little justification is given for the model formulation, and as described above, there are many problematic elements of the model formulation.

15 Response:

Thanks for your supportive suggestion. We agree with your opinion that the reliability of the model will increase with the extension of observed data series for calibration. However, it's a challenge work to collect such long-term representative data, which thereby requires more descriptions on the justification of model formulation.

As is discussed above, we have improved and given more details for model formulations in Section 2. The forms of feedback functions are on basis of previous studies (Feng et al., 2019; Van Emmerik et al., 2014), which has been proved with good performance and suitability. As the resources availability in the beginning of co-evolution can almost cover the demand in the study area, environmental awareness indeed remains at a low level. The parameters for environmental awareness feedback are thus poorly calibrated by observed data in the beginning. However, with the fast socioeconomic expansion, the contradiction between demand side and supply side is going to intensify. The society system will then be more sensitive to environment degradation and seek for environment recovery by constraining socioeconomic expansion through feedback.

To demonstrate the justification of the environmental awareness feedback, we have given an initial parameter sensitivity analysis on feedback driven by environmental awareness as is discussed in Section 4.1 (line 431~437). With high-level parameters for feedback functions (i.e., $\delta_{rp}^{E}$, $\delta_{rg}^{E}$ and $\delta_{ra}^{E}$), the feedback is strong and may lead violent degradation of socioeconomic sectors. With low-level parameters for feedback functions, the feedback is too weak to constrain the socioeconomic expansion and keeps the constant resources shortages for the integrated system (e.g., the constant water shortage and energy shortage in scenario II as discussed in Section 4.4.1). Therefore, we selected the parameters from appropriate interval based on parameter sensitivity analysis to ensure the rational co-evolution of the integrated system, which is considered as the foreseen scenario from the planning perspective.

We have also added descriptions for current limitation in line 828~832. We acknowledge that the model calibration is challenging, as the data series is not sufficiently long and the forms and parameters of the feedback function are not prescribed. We consider that sufficient case studies will gradually emerge over time, which could gradually cover a range of scenarios and slowly provide reliability in the WEFS nexus modeling.

- **Model results (Sections 4.2-4.3):**

1. Most of the discussion of the results (co-evolution of WEF system) is a text description of what is seen in the figures. The discussion does little to draw out and emphasize insights.

16 Response:

Thanks for your supportive suggestion. Based on the trajectory of environmental awareness, the co-evolution processes of water demand and energy consumption can be divided into four phases: expansion, contraction, recession and recovery. Food production can be divided into five phases based on the trajectory of food shortage awareness: accelerating expansion, natural expansion, contraction, recession and recovery.

The discussion for each phase is conducted with the order as follow: (1) water demand is firstly estimated by socioeconomic sectors (i.e., population, GDP and crop area); (2) water supply and water shortage are determined by water resources allocation; (3) the water supply and agriculture water shortage rate are then used to determine energy consumption and food production, respectively; (4) combined with planning energy availability and planning food production, energy shortage and food shortage can be estimated; (5) the shortage awareness for water, energy and food are then be determined, and further the environmental awareness; (6) the feedback driven by environmental awareness is then triggered to regulate the growth of socioeconomic sectors and further the water demand.

From the results, we find that available water and energy are the vital resources constraining the long-term concordant development of the integrated system in the study area. And more attention should be paid to the time lag of community's response to the deterioration WEFS nexus to prevent the integrated system from collapsing, especially after the fast expansion of water demand and energy consumption, which can provide useful support for policymakers.

2. The sensitivity discussion does little to add understanding. Most interpretations of sensitivity results are vague, such that the same observations could be stated just from the variable definition and model formulation. For example, in lines 551-553, the effect of lowering the food shortage sensitivity threshold level is obvious from its definition.

17 Response:

Thanks for your supportive suggestion. We have updated Figure 6, 7, 8 and 9 by replacing the black lines with colored lines and color bars so as to give a more informative sensitivity analysis for identifying the explicit variations of state variables

with varying parameters. Sensitive analysis on water demand is taken as an example in Figure 6 here.

[Figure]

**Figure 6. Trajectories of water demand with varied parameters. (Figure 7, 8 and 9 for trajectories of energy consumption, food production and environmental awareness have also updated)**

We find that the co-evolution mode of WEFS nexus functioning strongly depends on the selection of certain parameter values. Rational parameter setting of boundary conditions and critical values is of great significance for managers to keep the socioeconomic sectors from violent expansion and deterioration, especially in contraction and recession phases.

- **Impacts of reservoir system (section 4.4):**

1. The methodology applied here is unclear, what exactly does it mean that one scenario considers allocation and the other doesn't?

18 Response:

Thanks for your supportive suggestion. We have given more details to describe how the methodology applied to the mid-lower reaches of Hanjiang river basin in line 384~389. The study area is divided 28 operational zones based on the administrative

units and sub-basins. The socioeconomic data (i.e., population, GDP, and crop area) for water demand projection were collected based on administrative units, whereas the hydrological data were typically collected based on river basins. To ensure that the socioeconomic and hydrological data are consistent in operational zones, the study area was divided into 28 operational zones based on the superimposition of administrative units and sub-basins. Based on the water connections between operational zones and river systems, water resources allocation is conducted and further the WEFS nexus simulation.

The time resolutions of the model in the study area have also been added to help illustrate how the methodology is applied in line 413~419. The established WEFS nexus ran on a yearly loop. Specifically, as the water resources allocation model in the water system module took a monthly time step in the study (and the sub-time step was the default value: 1 day), the annual water supply and water shortage were first determined before being output to the energy system and food system modules, respectively. The annual shortage rates of water, energy, and food were then used to determine environmental awareness and further the feedback.

Scenario I considered water resources allocation is based on the real-world reservoir system, while scenario II removes the reservoir system from scenario I, so as to assess the impacts of water resources allocation on WEFS nexus.

2. Nonetheless, it seems that scenario I is running the model with the real-world reservoir network and scenario II is running the model with all reservoirs removed (?). If so, scenario II does not seem like a useful comparison. Is the region considering removing any or all dams in the basin?

17 Response:

Thanks for your supportive suggestion. One of the goals of our study is to assess the impacts of water resources allocation on WEFS nexus, as previous studies haven't considered water resources allocation or significantly simplified reservoirs operational rules in water resources allocation. Compared with scenario I, water resources allocation is removed in scenario II so as to assess the impacts of water resources allocation on WEFS nexus. The results indicate water resources allocation is of great significance in ensuring water supply and further sustaining the WEFS nexus from the planning perspective.

The numbers as well as the operational rules of reservoirs in the study area may change over time in the future. It's also a very interesting topic to investigate the impacts of changing reservoir system on WEFS nexus, which is a very informative study for managers from the planning perspective.

- There are language issues throughout the manuscript – most frequent were typos, poor sentence structure (lots of passive voice that creates confusion about who is the subject and what exactly they are doing), and inappropriate word choice. There are too many to list specifically.

18 Response:

Thanks for your supportive suggestion. We have carefully improved the writing quality in the revised manuscript.

[Figure]

ELSEVIER

**Certificate of Elsevier Language Editing Services**

The following article was edited by Elsevier Language Editing Services:
"A system dynamic model to quantify the impacts of water resources allocation on water-energy-food-society (WEFS) nexus"

Authored by:

**Yujie Zeng**

Date: 08-Dec-2021
Serial number: LE-227542-0C163816CFCE

**Reviewer 4**

This study modeled the WEF nexus by incorporating community sensitivity and reservoirs operation, where the co-evolution behaviors of the nexus across the water, energy, food, and society (WEFS) were simulated by the system dynamic model. The proposed approach was applied to the mid-lower reaches of the Hanjiang river basin in China, and the results indicated that water resources allocation could ensure water supply through reservoirs operation and greatly decrease the water shortage rate. This study is an interesting and crucial one for improving resources management. While modeling the WEF nexus in a large watershed is a very challenging problem and difficult to validate its suitability and applicability, especially when there are only limited datasets. This study made a great effort in this direction and proposed a sophisticated methodology with some preliminary analyzed results, which is a valuable contribution to the scientific community. However, I have some concerns and suggestions, which need to be better addressed, listed as follows.

Thank you very much for your positive remarks on our paper. We have thoroughly revised the paper based on your comments. We believe the current comments have greatly helped improve the quality of the paper. Here are the responses to your comments:

1. The initial conditions of external variables for the integrated system shown in Table 2 and the calibrated parameters presented in Table 3 should be explained in more details. I am curious why many parameters have the same calibrated value. How to set these values?

1 Response

Thanks for your supportive suggestion. We have added the notations, descriptions, units and values of variables and parameters in Table 2 and 3 in line 426 and 448.

We have added more details for model calibration in line 431~437. An initial parameter sensitivity analysis was adopted to screen out the insensitive parameter, which provided distinguishing 13 insensitive and 21 sensitive parameters. The setting of the insensitive parameter was based on expert knowledge and the study of Feng et al. (2019), which has been established to have good performance and suitability. The

sensitive parameters in the model were then calibrated based on expert knowledge and the observed data, and the calibrated values are presented in Table 3 (insensitive parameters are set to 0.0856).

2. How many datasets are used for model calibration? The number of calibrated parameters used for model calibration should be discussed. How to justify the suitability and applicability of the calibrated model should be given.

2 Response

Thanks for your supportive suggestion. The observed data of annual water demand, energy consumption, food production, population, GDP and crop area from 2010 to 2019 are used to calibrate the model as is shown in line 440~443. We have added the details for model calibration, as is discussed in "1 Response",. 21 sensitive parameters are screened out and calibrated by fitting observed data, the calibration values of which are listed in Table 3.

We have also given more details for the suitability and applicability of the calibrated model in line 437~447. The Nash-Sutcliffe Efficiency (NSE) coefficient and percentage bias (PBIAS) are used to calibrate the model. When the NSE was >0.7 and absolute value of PBIAS was <15%, the modeling performance was considered reliable. The NSEs are always more than 0.7 and the corresponding PBIASs are within -15% to 15%, suggesting that the established model is reliable for simulating the co-evolution of the WEFS nexus.

3. The "Respond links" among the different variables in the WEFS nexus should be explained in much more detail. The terms of feedback functions based on previous work should further explain their suitability.

3 Response

Thanks for your supportive suggestion. The details of respond links have been added Section 2. Description for connection between water system and energy system as well as food system will be improved in line 130~136. The water demands and available water resources are further inputted into the water resources allocation model to determine the water supply and water shortage for every water use sector in each operational zone. The water supply for socioeconomic water use sectors and agricultural water shortage rates as outputs from the water system module are taken as

the inputs of the energy system module and food system module to determine the energy consumption and food production, respectively.

Description for feedback driven by environment awareness has been improved in line 144~148. As environmental awareness accumulates over its critical value, negative feedback on socioeconomic sectors (i.e., population, GDP, and crop area) will be triggered to constrain the increases in water demand, and further energy consumption, and food production to sustain the WEFS nexus.

The description for the suitability of the feedback function have been added in line 332~336. The terms of feedback functions are based on the studies of Feng et al. (2019) and Van Emmerik et al. (2014), which have been established to have good performance and suitability, as they have been successfully applied to simulate the human response to environmental degradation in the Murrumbidgee river basin (Australia) and Hehuang region (China).

4. Figure 4 shows the trajectories of population, GDP, crop area, water demand, energy consumption, food production, shortage rates for water, energy, and food, awareness for water shortage, energy shortage, and food shortage as well as environmental awareness during 2010-2070. The trajectories are the basis of the following analyses. How to get these trajectories should be given in more detail, and their suitability should be discussed?

4 Response

Thanks for your supportive suggestion. The co-evolution of WEFS nexus is conducted based on system dynamic modeling (SDM), which conducts according to the nonlinear ordinary differential equations and dynamic feedback loops as is demonstrated in Section 2

More details about the application of the established WEFS nexus in the study area have been added in line 413~419. The SDM is applied to the mid-lower reaches of Hanjiang river basin. The established WEFS nexus ran on a yearly loop. Specifically, as the water resources allocation model in the water system module took a monthly time step in the study (and the sub-time step was the default value: 1 day), the annual water supply and water shortage were first determined before being output to the energy system and food system modules, respectively. The annual shortage rates of water, energy, and food were then used to determine environmental awareness and further the feedback.

5. How to divide the evolution of water demand and energy consumption into four phases should be given?

5 Response

Thanks for your supportive suggestion. The co-evolution processes of water demand and energy consumption can be divided into four phases (i.e., expansion, contraction, recession and recovery) based on the trajectory of environmental awareness.

We have emphasized the phase dividing rules in Section 4.2.

In line 477~479: With environmental awareness below its critical value, the negative feedback on socioeconomic sectors is not triggered, and water demand, as well as energy consumption, increases rapidly, which is defined as the expansion phase (2010–2032);

In line 502~504: As environmental awareness exceeds its critical value, negative feedback on socioeconomic sectors is triggered, and the increase in water demand and energy consumption is constrained, which is defined as the contraction phase (2033–2039);

In line 517~519: Environmental awareness accumulates to the maximum value and water demand, and energy consumption decrease significantly, which can be defined as the recession phase (2040–2045);

In line 527~529: As environmental awareness gradually decreases below its critical value, water demand and energy consumption decrease slightly and then tend to stabilize, which is defined as the recovery phase (2046–2070).

6. The seven controllable parameters adopted for sensitivity analysis should be discussed in more detail.

6 Response:

Thanks for your supportive suggestion. We have added more details for the parameter selection in sensitivity analysis in line 573~576. As the critical values and boundary conditions of WEFS nexus are considered as vital factors for policymakers and managers to control the integrated system so as to achieve the concordant development goals, seven parameters are selected for sensitivity analysis.

We have also updated the figures for sensitivity analysis by replacing the black lines with colored lines and color bars so as to give a more informative sensitivity

analysis for identifying the explicit variations of state variables with varying parameters. Sensitive analysis on water demand is taken as an example in Figure 6 here.

[Figure]

**Figure 6. Trajectories of water demand with varied parameters. (Figure 7, 8 and 9 for trajectories of energy consumption, food production and environmental awareness have also been updated)**

We find that the co-evolution mode of WEFS nexus functioning strongly depends on the selection of certain parameter values. Rational parameter setting of boundary conditions and critical values is of great significance for managers to keep the socioeconomic sectors from violent expansion and deterioration, especially in contraction and recession phases.

7. The conclusion seems like a long summary of the current study. The main contribution with brief (solid) scientific findings extracted from this study might be more interesting to read and easy to learn.

7 Response:

Thanks for your supportive suggestion. We have carefully improved the conclusion part in line 807~827.

[revised manuscript text omitted]

---

## Author Response (AR2)

**Cover letter**

Dear professor Murugesu Sivapalan:

We greatly appreciate you and the reviewers for taking time to review this manuscript and provide us with constructive and valuable comments. We have addressed all comments from reviewers and the manuscript has been much improved. Our changes are marked in Blue in the revised manuscript. And our responses to the reviewers are detailed in this response-to-reviewers document submitted with the revised manuscript.

Looking forward to hearing from you.

Sincerely,

Dr. Dedi Liu

Corresponding author: Dedi Liu

Email: dediliu@whu.edu.cn

**Reviewer 1**

I appreciate the efforts the authors have made to address my comments on the early version of the manuscript. However, as I read throughout the revised manuscript, I think there are still some technical issues with the model formulation and lack of robustness of model assumptions and simulation experiments, listed as follows.

Thank you very much for your valuable comments on our paper. We believe current comments have greatly helped improve the quality of the paper. Here are the responses to your comments:

1. Equation (2)-(4). What is the definition of "environmental capacities of socioeconomic variables"? If $N_t > N_{cap}$, $N$ is mathematically forced to decrease (or stay constant), even if the community has no awareness of environmental deterioration (i.e., $f(E)=0$). Why?

1. Response:

Thanks for your supportive comment. "environmental capacities of socioeconomic variables" indicates the maximum value that can be carried by the system, which is used to constrain the socioeconomic variables within their maximum values. Even if negative feedback driven by environmental awareness is not triggered (i.e., $f(E)=0$), $N$ is not allowed to increase when $N_t$ is larger than $N_{cap}$.

To make it clearer, we have replaced "environmental capacities of socioeconomic variables" with "environmental carrying capacities of socioeconomic variables" in the paper.

Thanks.

2. The authors argue that "As the growth rate in the original Malthusian growth model is adopted as a constant, socioeconomic factors will reach infinity in a long-time evolution. Therefore, we assume that population, GDP, and crop area increase with decreasing rates over time" (Line 172).

In fact, although a decreasing, exponential term is added to the growth rate equation, the variables can still reach infinity in a long run, as long as the value of the growth rate is positive. Therefore, it cannot justify why you have to add the exponential term.

And I still cannot appreciate the assumption that technology development will slow down the growth of population, GDP, and crop area. I suggest the authors consider using another statement instead of "technology development" to explain this term. In fact, in the model, the growth rate is just a function of time, and there is no technology involved.

2. Response:

Thanks for your supportive comment. We have given more details for the improved Malthusian growth model in line 174-179. According to previous studies, the socioeconomic expansion in China will slow down (He et al., 2017; Lin et al.,

2016), the growth rate of which will decrease. The constant growth rate in the original

Malthusian growth model is thereby not applicable for socioeconomic simulation.

Therefore, we used exponential terms (i.e., $\exp(-\varphi t)$) to simulate the evolution of socioeconomic variables, which increases with decreasing rate.

We agree with your opinion that the variables can still reach infinity in a long run with exponential term (e.g., scenario II in Figure 10). Therefore, we add the feedback function driven by environmental awareness (i.e. $f(E)$) to the equation to regulate the socioeconomic expansion. As is shown in Figure 10, socioeconomic variables keep increasing under the scenario without considering environmental awareness feedback (i.e., scenario II), while the over-speed socioeconomic expansion is effectively constrained under the scenario considering environmental awareness feedback (i.e., scenario I), which exactly indicates that environmental awareness is of great significance for the sustainable development of WEFS nexus.

We have replaced "technological development" with "social development" (i.e.,

$\kappa_P \exp(-\varphi_P t)$, $\kappa_G \exp(-\varphi_G t)$, $\kappa_{CA} \exp(-\varphi_{CA} t)$ are used to depict the impacts of social development on the evolution of population, GDP, and crop area, respectively).

Thanks.

3. In fact, I believe the growth rate of the socioeconomic variables should be related to the size of the variables. It might be less reasonable to simply assume that the growth rate is just a function of time.

3. Response:

Thanks for your supportive comment. We agree with your opinion that growth rate of socioeconomic variable is not only the function of time, but also related to the sizes of these variables. As the interconnections between water, energy, food and society systems are being intensified, the evolution of socioeconomic variable is not only related to its own size, but also impacted by the status of other systems.

Environmental awareness, which takes resources demand, supply and shortage in water, energy and food systems into account, is considered as a comprehensive indicator to sustain the WEFS nexus system. Therefore, we assume that growth rate of socioeconomic variable is the function of time and environmental awareness, as is shown in equation (2)-(4).

Thanks.

4. If Nt<Ncap and f(E)=0, rpt would decrease with time, and its minimum value is rp0. Why? rp0 represents the growth rate of population in the baseline year. Does the baseline year mean the first year? If so, when t=1, why rpt does not equal rp0?

4. Response:

Thanks for your supportive comment. We find a typo in the equation and we have revised them as follow:

$$\begin{cases} \dfrac{dN_t}{dt} = r_{P,t} * N_t \\ r_{P,t} = \begin{cases} r_{P,\,0} * \kappa_P * \exp(-\varphi_P t) + f_1(E) & N_t \le N_{cap} \\ \mathrm{Min}(0, r_{P,0} * \kappa_P * \exp(-\varphi_P t) + f_1(E)) & N_t > N_{cap} \end{cases} \end{cases} \qquad (2)$$

where $N_t$ is the population in the $t$-th year; $N_{cap}$ denotes the environmental capacities of population; $r_{P,\,0}$ is the growth rate of population from historical observed data before the baseline year; $r_{P,\,t}$ is the growth rate of population in the $t$-th year;

$\kappa_P * \exp(-\varphi_P t)$ is used to depict the impacts of social development on the evolution of population; $E$ is environmental awareness; and $f_1$ represents the feedback function.

As discussed above, we assume that the socioeconomic variables are going to increase with decreasing rate according to previous studies (He et al., 2017; Lin et al., 2016). The ideal minimum growth rate will decrease approaching to 0 in the revised equation ($r_{*,0}$ in previous equation), as the socioeconomic variables are considered to keep increasing with decreasing rate until reaching their environmental carrying capacities if no environmental awareness feedback is triggered. However, the ideal conditions may be difficult to be satisfied in the foreseeable future and can give little information for the planning (except for scenario II removed feedback function, environmental awareness feedback is triggered under another three representative scenarios shown in Figure 10).

The baseline year is the first year. We have re-described the definition of $r_{p,0}$ to eliminate the misleading as follow: $r_{p,0}$ is the growth rate of population from historical observed data before the baseline year.

Thanks.

5. The assumption that an increase in environmental awareness will have a negative effect on GDP growth is less robust. In fact, if the community feels the water shortage issues, they usually replace the water-intensive industrial sectors with some less water-intensive ones. And the latter (e.g., some high-tech industries) might contribute even more to GDP growth, with a relatively smaller volume of water consumption.

5. Response:

Thanks for your supportive comment. We agree that long-term high-level environmental awareness can also promote the advancement of resource-saving technology and further increase GDP, besides the constraints on socioeconomic variables. Simultaneously taking the positive and negative impacts of environmental awareness on GDP may be more reasonable. However, it will also make it difficult to distinguish the negative feedback on GDP driven by environmental awareness, which is the focus of our study. The process that environmental awareness promotes the advancement of resource-saving technology and further increase GDP can be quite complex, which is related to not only the level and duration of environmental awareness but also the sizes of various socioeconomic factors. It's really an interesting topic and will become the focus of our further study.

And we have taken it as a limitation and future work as discussed in line 914-919

in conclusion section "We acknowledge that environmental awareness feedback functionality remains to be further improved. Indeed, environmental awareness also has potential to contribute to socioeconomic expansion by promoting resources-saving technology. It's the function of the level and duration of environmental awareness, and the sizes of socioeconomic factors, which will become the focus of our further study".

Thanks.

6. Equation (5) and equation (16). The growth rates of water use quota and energy use quota are always negative, which means the water use quota and energy use quota would decrease to a negative value in a long run. This is not true. Maybe a minimum value is needed to constrain the variable.

6. Response:

Thanks for your supportive comment. We have added the minimum value as constraint in equation (5), (16), and Table 2.

$$\begin{cases} \dfrac{dWQ_{i,j}^{t}}{dt} = WQ_{i,j}^{t} * r_{qwu,t} \\[2mm] r_{qwu,t} = \begin{cases} r_{qwu,0} * \kappa_{qwu} * \exp(-\varphi_{qwu}\,t) & WQ_{i,j}^{t} > WQ_{i,j}^{min} \\[1mm] 0 & \text{else} \end{cases} \end{cases} \tag{5}$$

where $WQ_{i,j}^{t}$ denotes the water use quota of the $j$-th water user in the $i$-th operational zone in the $t$-th year; $r_{qwu,\,0}$ and $r_{qwu,\,t}$ are the growth rates of water use quotas from historical observed data and $t$-th year, respectively; $WQ_{i,j}^{min}$ is the minimum value of water use quota; and $\kappa_{qwu}*\exp(-\varphi_{qwu}t)$ is used to depict the water-saving effect of social development on the evolution of water use quota.

Thanks.

7. Equation (19) has the same problem as equation (2).

7. Response:

Thanks for your supportive comment. We have revised equation (19) as discussed above.

Thanks.

8. In my previous comments, I suggested the authors give some observable evidence to show human adaptive responses. I want to know if there is any evidence to show that the community will take measures to constrain the growth of population, GDP, and crop area? This is important, because the data points during the modeling period (2010-2019) cannot validate the model assumptions about the feedback from environmental awareness to population, GDP, and crop area.

8. Response:

Thanks for your supportive comment. We believe our manuscript have been greatly benefited from this valuable suggestion. We have added more observed evidence to show human adaptive response to resources shortage from water, energy and food systems in line 461-481.

The environmental awareness is the key factor to drive the feedback. However, as environmental awareness is a subjective variable, there are no empirical observed data to calibrate it, which requires more evidences to show adaptive human response to environmental awareness. Hepburn et al. (2010) have reviewed studies on environmentally related human behavioral economics. Substantial studies indicate that environmental awareness is considered as an important factor in modelling socioeconomic decisions and policies for water, energy and food systems (Li et al.,

2019; Li et al., 2021; Lian et al., 2018; Rockson et al., 2013; Xiong et al., 2016). For instance, Xiong et al. (2016) investigated the evolution newspaper coverage of water issues in China based on water-related articles in a major national newspaper, *People's*

*Daily*. They found that economic development was the primary target of China before

2000. With the conflict between water demand and supply being intensified, concerns about water security arisen in the newspaper since 2000, which indicated that environmental awareness towards water shortage emerged. Related policies (e.g., the strictest water resources control system for water resources management policy) were thereby implemented to constrain the over-speed socioeconomic expansion and further ensure water security.

Thanks.

9. Model calibration. Results about parameter sensitivity analysis are not shown. I

cannot understand how the insensitive parameters are identified (based on expert knowledge is not a rigorous way). In addition, the 13 insensitive parameters are used in different equations for different variables. Why can all of their values be set to

0.0856?

9. Response:

Thanks for your supportive comment. We have added more details for parameter calibration in line 440-444. As is shown in Table 3, some parameters in the model are adopted as auxiliary parameters, which are not equipped with exactly physical definitions. It indicates there is no independent empirical data to calibrate these parameters. Therefore, by reviewing previous studies (Feng et al., 2019; Feng et al.,

2016; Van Emmerik et al., 2014) and expert knowledge, we evaluated the order of magnitudes and rational boundaries for these parameters. The initial parameter sensitivity analysis was then conducted to identify the sensitive and insensitive parameters. As the insensitive parameters are not able to remarkably alter the system (Taking insensitive parameter $\varphi_P$ as an example, as shown below), the empirical values in previous studies (Feng et al., 2019; Feng et al., 2016) were adopted.

Thanks.

[Figure]

[Figure]

**Figure 1. Trajectories of environmental awareness, water demand, energy consumption, and food production with varied $\varphi_P$.**

10. The modeling period is extended to 2070. How are the external drivers assumed for the future prediction?

10. Response:

Thanks for your supportive comment. We have added more details to describe how the water availability and water demand extended to 2070 in line 420-423. "water availability from 1956 to 2016 was adopted as the future water availability, while dynamic water demand was projected in water system module, both of which were inputted into water resources allocation model."

Thanks.

11. Line 647. "constraining WSRcrit, ESRcrit, PEA, and Ecrit can maintain the integrated system from constant water shortage and energy shortage…". What can we do to constrain these parameters? All of these parameters are boundary conditions, and they are not directly associated with human adaptive actions.

11. Response:

Thanks for your supportive comment. This study aims to assess the impacts of environmental awareness feedback and water resources allocation on WEFS nexus. Therefore, we only taken these parameters as static boundary conditions, while the dynamics with human adaptive actions haven't considered yet. However, we have evaluated the sensitivity of these parameters in Section 4.4, and found that these parameters can effectively regulate evolution of socioeconomic variables and are of great significance for the sustainable development of WEFS nexus, which has laid the basis for further study on the dynamics of these parameters with human adaptive actions.

Thanks.

12. Line 32-36. The statement is somewhat misleading. I firstly thought the energy shortage rate decreased from 17.16% to 5.80% during some time period. But table 7

shows that this is a comparison between two modeling scenarios. This cannot be used as a conclusion because once you set a different scenario (e.g., different parameter values), the values of energy shortage rate will change as well.

12. Response:

Thanks for your supportive comment. One of our goals is to assess the impacts of environmental awareness feedback on WEFS nexus. Therefore, we set scenario I

considering environmental awareness feedback, while scenario II doesn't, as is shown in Table 6. The impacts of environmental awareness feedback on WEFS nexus can be studied by comparing the difference between scenario I and II. And we found that environmental awareness can effectively capture human sensitivity to resources shortage and keep the integrated system from constant resources shortage by regulating socioeconomic expansion (e.g., the average annual energy shortage rate under scenario II decreased from 17.16% to 5.80% under scenario I). Differences between different periods under the same scenario can be used to assess the evolution phases division of socioeconomic variables, as is discussed in Section 4.2.

To remove the misleading, we have added the scenarios and their corresponding values in abstract in line 35-36. "Rational water resources allocation can ensure water supply through reservoir operation, decreasing the water shortage rate from 15.89%

under scenario IV to 7.20% under scenario III."

Thanks.

Overall, I think the model is a bit over-complexed. Perhaps, the authors may consider modeling water demand directly. It seems that it is not necessary to model population,

GDP, crop area, and water use quota, individually. According to equation (17) and equation (20), both energy consumption and food production are just related to water supply, and have nothing to do with population, GDP, and crop area. Perhaps using water demand as a state variable might largely simplify the model structure and still maintain the completeness of the WFE system story.

Response:

Thanks for your valuable and constructive suggestion. Directly considering water demand as a state variable without simulating population, GDP, crop area, and water use quota can no doubt effectively simplify the model structure. However, it will be difficult to distinguish different types of water demand (i.e. municipal, rural, industrial and agricultural water demand), as population, GDP, and crop area are considered as important factors for water demand projection in quota method. Furthermore, it will also challenge the estimations of energy consumption and food production, as energy use quotas are different in different sectors (shown in equation (17) and Table 2), and food production is determined by crop area and agricultural water shortage rate (shown in equation (20)). Therefore, eliminating the simulation of population, GDP, crop area, and water use quota may directly bring risks to water and food systems, and further WEFS nexus system.

Thanks.

**Reviewer 3**

The authors have made a considerable effort the revise the manuscript. While some concerns have been adequately addressed, other have not, and the revisions and authors responses also raise new concerns. The new and remaining concerns are described below. Because the major concerns (1-3 below) are such that they permeate the entire paper, I believe the paper should not be accepted.

Thank you very much for your valuable comments on our paper. We believe the current comments have greatly improved the quality of the paper. Here are the responses to your comments:

(1) Co-evolution of WEF nexus – calibration and discussion (primarily sections 4.1 and 4.2):

• It is misleading to claim that the model is "reliable for simulating the co-evolution of the WEFs nexus" (lines 446-447). As expressed before, the data only covers the early, "expansion phase" of the simulation and is thus insufficient to validate the model. It is also misleading to present the NSE and percent bias of the calibrated model without clarifying that the data only cover a short window and a single phase of the co-evolution. In section 4.1 and throughout the manuscript, all language which asserts or implies that the model has been validated should be eliminated or revised to be accurate. For instance, in the abstract, it is misleading to claim that "The results show that environmental awareness can effectively capture the human sensitivity to shortages from water, energy, and food systems". Prior comments regarding this issue have not been adequately addressed.

1. Response:

Thanks for your supportive comment. We believe our manuscript has greatly benefited from this valuable suggestion. We have given more details to explicitly emphasize that the observed data can only cover the initial expansion phase of WEFS nexus co-evolution in line 461-464. To further demonstrate the reliability of the established WEFS nexus, evidence of human adaptive response to shortages from water, energy, and food systems is added in line 464-480. The state "reliable for simulating the co-evolution of the WEFS nexus" has been revised as "the established model still has potential to simulate the co-evolution of WEFS nexus." in line 480-481.

As environmental awareness stays at a low level and the feedback is not triggered in initial expansion phase, the feedback driven by high-level environmental awareness hasn't been calibrated yet. However, as environmental awareness is a subjective variable, there are no empirical data to calibrate it, which requires more evidences to show adaptive human response to environmental awareness. Hepburn et al. (2010) have reviewed studies on environmentally related human behavioral economics. Substantial studies indicate that environmental awareness is considered as an important factor in modelling socioeconomic decisions and policies for water, energy and food systems (Li et al., 2019; Li et al., 2021; Lian et al., 2018; Rockson et al., 2013; Xiong et al., 2016). For instance, Xiong et al. (2016) investigated the evolution newspaper coverage of water issues in China based on water-related articles in a major national newspaper, *People's Daily*. They found that economic development was the primary target of China before 2000. With the conflict between water demand and supply being intensified, concerns about water security arisen in the newspaper since 2000, which indicated that environmental awareness towards water shortage emerged. Related policies (e.g., the strictest water resources control system for water resources management policy in China) were thereby implemented to constrain the over-speed socioeconomic expansion and further ensure water security. Therefore, the established model still has potential to simulate the co-evolution of WEFS nexus.

Thanks.

• The authors state in their responses (see response 15 especially), that they selected parameters "to ensure the rational co-evolution of the integrated system". This sounds like the "phases" of co-evolution were imposed on the model, rather than emerging from the model. Thus, the entire perspective of section 4.2, which discusses the phases as an insightful model result, is misleading.

2. Response:

Thanks for your supportive comment. The state should be corrected. We selected the parameters from appropriate intervals based on parameter sensitivity analysis to ensure "socioeconomic variables in WEFS nexus with rational intervals", rather than

"the rational co-evolution of the integrated system". As is shown in Figure 8, 9, 10, and 11 for parameter sensitivity analysis, despite the amounts of socioeconomic variables have changed with varied parameters, the phases dividing rule is still valid for the co-evolution of WEFS nexus.

Therefore, the phases emerge from the model, rather than imposed.

Thanks.

• Analysis of co-evolution is, by nature, analysis of how states *change over time*.

The use of static energy production and food target are therefore troublesome. These values are critical to shortages and therefore awareness and all other state co-evolution. In their responses, the authors claim this simplification is acceptable since the focus of the study is as stated above; this response may be acceptable, but the paper's attention given to the co-evolution and its phases should be minimized accordingly (i.e. dramatically cut and revise section 4.2).

3. Response:

Thanks for your supportive comment. We agree with your opinion that WEFS

nexus co-evolution is not the focus of the study and thereby needs simplification. We have greatly simplified Section 4.2 for WEFS nexus co-evolution, and only the phase dividing, and their primary properties are emphasized. (the length of Section 4.2

decreases from 1,329 to 789 words).

Thanks.

• The phases of co-evolution are essentially the same as those presented in Feng et al.

2016. Reference to that study should be made, and the discussion in the section 4.2

shortened significantly.

4. Response:

Thanks for your supportive comment. We have cited the reference in line 490, and greatly shortened Section 4.2.

Thanks.

(2) In light of the authors response, the scales of the energy, food, and water systems appear to be very different (sections 2.1, 2.2, 2.3). Thy system is only energy use for water supply within the basin; the water system is all water users within the basin; and the food system goes beyond the basin boundaries (exporting food was provided as justification for food target/demand being an external driver). However, the energy, food, and water shortage awareness are all aggregated into a single "environmental awareness" which drives population, GDP, and crop area. The discordant scales should be accounted for, or at a minimum discussed, when aggregating environmental awareness. In addition, the different scales would seem to have implications for how the model results are interpreted, but no such discussion is provided in the results.

5. Response:

Thanks for your supportive comment. Water demand in each water user in each operational zone is projected in water system module. Water supply for every water user is then simulated according to water resources allocation model. Once the water supply is determined, the energy consumption at each water user during water supply process can be estimated. The food production in every operational zone is also determined, with the agricultural water shortage rate outputted from water resources allocation model. Therefore, the water demand, water supply, energy consumption, energy supply, and food production are within the basin, except for the target food production (or food demand).

As current observed crop area and food production structure are formed with the target food production considering food exported, it's hard to distinguish the crop area for food export from the total crop area. If the target food production doesn't consider the food export, food production will be remarkably larger than food demand, and the current crop area will keep rapidly shrinking with environmental awareness feedback, which is not consistent with the fact. Therefore, we still consider the exported food as part of target food demand of the basin, and further laid the basis for environmental awareness estimation.

Simultaneously, we find that target food production is an important parameter for WEFS nexus, particularly for food system, as shown in Figure 8 (e), 9 (e), 10 (e), and 11 (e). Therefore, it's exactly more reasonable to separate local food demand and exported food demand, to further assess their impacts on WEFS nexus, respectively, which is really an interesting top. However, it's not the focus of this paper, and it will be studied in our next work.

Thanks.

(3) The discussion of all results still needs re-organization and narrowing of focus. The abstract and author comments indicate that the focus of the paper is (1) impacts of environmental awareness feedbacks and (2) impacts of reservoir storage. It seems that the novelty of the present study (especially in comparison to Feng et al. 2016 and 2019) comes from section 4.4 – toggling the environmental awareness feedback on/off, and using a detailed reservoir simulation rather than a simplified reservoir model. The results discussion should be cut down to indeed focus on these contributions. Perhaps all of section 4.2 should be eliminated (more on this section above). Section 4.3 (sensitivity analysis) should be revised to focus on the two main contributions, rather than a comprehensive description of sensitivity.

6. Response:

Thanks for your supportive comment. We agree with your opinion and our manuscript has greatly benefited from this valuable suggestion. To explicitly emphasize the contribution of the study, we re-organized the discussion structure: Section 4.1 model calibration, Section 4.2 co-evolution of WEFS nexus, Section 4.3 impacts of environmental awareness feedback and water resources allocation on WEFS nexus, and Section 4.4 sensitivity analysis for WEFS nexus.

Specifically, for Section 4.2, as the WEFS nexus co-evolution is not the focus of the study, but still considered as the basis for Section 4.3 and 4.3, we greatly simplified Section 4.2, rather than eliminated.

The updated sensitivity analysis in Section 4.4 consists of two parts: 4.4.1 sensitivity analysis of **environmental awareness feedback** on WEFS nexus, and newly added 4.4.2 sensitivity analysis of **water resources allocation schemes** on WEFS nexus. For sensitivity analysis of environmental awareness feedback, seven parameters related to the boundary conditions and critical values are selected. Results indicate that these parameters can dominate the environmental awareness evolution, and further regulate the pace of socioeconomic expansion by environmental awareness feedback, which is of significance to keep the integrated system from violent deterioration in contraction and recession phases. For sensitivity analysis of water resources allocation schemes, varied multipliers for water release from reservoir based on reference scenario are adopted. According to the response to different water resources allocation schemes, operational zones were classified into four types as shown in Figure 12. And we found that regulating capacity of water project is an important factor in water resources allocation to ensure the stability of water system to sustain WEFS nexus. Particularly for the area with certain regulating capacity of water project but cannot totally cover the water demand, regulating the water release from reservoir by rational water resources allocation schemes can further ensure water supply and contribute to the sustainable development of the WEFS nexus.

The newly added results for sensitivity analysis of water resources allocation schemes are shown in Figure 8, 9, 10, 11, 12, and 13.

Thanks.

[Figure]

**Figure 8 (h). Trajectories of environmental awareness with varied parameters.**

[Figure]

**Figure 9 (h). Trajectories of water demand with varied parameters.**

[Figure]

**Figure 10 (h). Trajectories of energy consumption with varied parameters.**

[Figure]

**Figure 11 (h). Trajectories of food production with varied parameters.**

[Figure]

**Figure 12. Spatial distribution of A, B, C, and D types of operational zones.**

[Figure]

[Figure]

**Figure 13. Socioeconomic variables with varied reservoir release multiplier in Z9, Z1, Z8, and Z13: (a) changing rates of water demand; (b) changing rates of energy consumption; (c) changing rate of food production; (d) water shortage rates; (e) water shortage rates of water users in Z1 (user 1, 2, 3, 4, and 5 are related to municipal, rural, in-stream ecology, industrial, and agricultural users); (f) water shortage rates of water users in Z8.**

(4) Less permeating issues

• The new water quota formulation (eqn 5) seems to imply that the water quota will go to zero as time goes to infinity (assuming parameter values as provided in section 4).

7. Response:

Thanks for your supportive comment. We have added the minimum value as constraint in equation (5), (16), and Table 2.

$$
\begin{cases}
\dfrac{dWQ_{i,j}^{t}}{dt} = WQ_{i,j}^{t} * r_{qwu,t} \\[2mm]
r_{qwu,t} = \begin{cases}
r_{qwu,0} * \kappa_{qwu} * \exp(-\varphi_{qwu}t) & WQ_{i,j}^{t} > WQ_{i,j}^{min} \\
& \text{else}
\end{cases}
\end{cases}
\tag{5}
$$

where $WQ_{i,j}^{t}$ denotes the water use quota of the $j$-th water user in the $i$-th operational zone in the $t$-th year; $r_{qwu,0}$ and $r_{qwu,t}$ are the growth rates of water use quotas from historical observed data and $t$-th year, respectively; $WQ_{i,j}^{min}$ is the minimum value of water use quota; and $\kappa_{qwu}*\exp(-\varphi_{qwu}t)$ is used to depict the water-saving effect of social development on the evolution of water use quota.

Thanks.

• There is still no index provided for the summations in equations 6-8. Also, in equation 7, if total inflow includes natural inflow, is natural inflow being subtracted twice? Why? And why isn't the demand reduction factor included in the denominator of equation 8?

8. Response:

Thanks for your supportive comment.

(1) We have added the summation for total water shortage rate in equations (8).

$$WE_{i,j}^{sts} = (\sum_{1}^{sts-1} WTSup_{i,j}^{sts} - \sum_{1}^{sts-1} WRSup_{i,j}^{sts}) * \frac{(Tsts - sts + 1)}{(sts - 1)} \tag{6}$$

$$WS_{i,j}^{sts} = \frac{WD_{i,j}^{ts}(1 - f_{red}) - \sum_{1}^{sts} WTSup_{in}^{sts} - WE_{i,j}^{sts}}{Tsts - sts + 1} \tag{7}$$

$$WSR_t = \frac{\sum_{i,j} WSR_{i,j}^{t}}{f_{red} * \sum_{i,j} WD_{i,j}^{t}} = \frac{\sum_{ts}\sum_{sts} WS_{i,j}^{sts}}{f_{red} * \sum_{ts} WD_{i,j}^{ts}} \tag{8}$$

where $ts$ is the current time step; $Tsts$ denotes the total number of the sub-time steps; $sts$ is the current sub-time step; $WE_{i,j}^{sts}$ represents the extrapolated natural water inflow for the $j$-th water use sector in the $i$-th operational zone; $WTSup_{i,j}^{sts}$ is the total water supply; $WRSup_{i,j}^{sts}$ is the water supply from reservoir; $WD_{i,j}^{ts}$ is the water demand; $f_{red}$ is the demand reduction factor; $WS_{i,j}^{st}$ is the water shortage; $WSR_{i,j}^{t}$ is the water shortage rate in the $t$-th year; and $WSR_t$ is the total water shortage rate.

(2) The total water supply ($WTSup_{i,j}^{sts}$) comprises natural water inflow and water supply from reservoir. In each sub-time step (except the first), the average natural water inflow (i.e., $WE_{i,j}^{sts}$) in the previous $sts$-1 sub-time steps is estimated as the extrapolated natural water inflow in the remaining $Tsts$-$sts$+1 sub-time steps using equation (6). In equation (7), $\sum_{1}^{sts} WTSup_{i,j}^{sts}$ covers the total natural water inflow in the previous $sts$-1 sub-time steps, while $WE_{i,j}^{sts}$ is considered as the total natural water inflow in the remaining $Tsts$-$sts$+1 sub-time steps. Therefore, natural water inflow is not subtracted twice.

(3) Thanks for your reminder. It's a typo and we have added reduction factor into equation (8).

Thanks.

• What does it mean that IRAS runs on a yearly "loop" (line 208)?

9. Response:

Thanks for your supportive comment. The operational rule period of IRAS

model is yearly-based. Once the length of time step is given, the number of time steps in a year can be determined and IRAS model will split the long-term series into yearly series to adapt to the operational rule. For instance, both the water availability and water demand are long term series from 2010 to 2070 in the study, while the reservoir operational rules are not. The reservoir operational rules repeat every year (in the study, there are 12 time steps in a year, as the length of time step is a month) to regulate the water release during the long term series. More details can be found in

Matrosov et al. (2011).

As the state "IRAS runs on a yearly loop" can give little information and may mislead readers, we have removed it from the paper.

Thanks.

• "Planning food production" should be renamed to indicate that it is target production (or demand) and authors say in responses (line 297). Section 2.3 should also state that the food production target is an external driver because food production is largely exported (per author responses).

10. Response:

Thanks for your supportive comment. We have renamed "Planning food production" with "Target food production" in the paper. The state indicating the target food production has considered the exported food has been added in line290-292

"With the target food production which has considered the local and exported food demands of basin, the food shortage rate can then be estimated using equations (20)

and (21)".

Thanks.

• It is still unclear how agricultural water demand is implemented in the model. How are exceedance frequencies/demands assigned to years of the simulation? (lines

406-410)

11. Response:

Thanks for your supportive comment. We have given more details for agricultural water demand frequency in line 413-416. Agricultural water demand depends on agricultural water use quota and crop area. Agricultural water use quota is related to water availability frequency. The historical water availability during

1956~2016 is adopted as the water availability of simulation years in the paper, based on which the frequencies can be determined. As the water availability empirical frequencies between years during the long term (i.e., 1956~2016) are different, the agricultural water demands are also different between years. However, it's hard to collect all the agricultural water use quotas under all the frequencies. Therefore, four typical frequencies P = 50%, 75%, 90%, and 95% (corresponding agricultural water use quotas data are available), are selected to indicate wet, normal, dry, and extreme dry years to simplify the agricultural water demand series (If $P_t \leq 60\%$, the year is defined as wet year; if $60\% < P_t \leq 80\%$, the year is defined as normal year; if

$80\% < P_t \leq 90\%$, the year is defined as dry year; if $90\% \leq P_t$, the year is defined as extreme year).

Thanks.

**References**

Feng, M., Liu, P., Li, Z., Zhang, J., Liu, D., and Xiong, L.: Modeling the nexus across water supply, power generation and environment systems using the system dynamics approach: Hehuang Region, China, Journal of Hydrology, 543, 344-359, 10.1016/j.jhydrol.2016.10.011, 2016.

Feng, M., Liu, P., Guo, S., Yu, D. J., Cheng, L., Yang, G., and Xie, A.: Adapting reservoir operations to the nexus across water supply, power generation, and environment systems: An explanatory tool for policy makers, Journal of Hydrology, 574, 257-275, 10.1016/j.jhydrol.2019.04.048, 2019.

He, S. Y., Lee, J., Zhou, T., and Wu, D.: Shrinking cities and resource-based economy: The economic restructuring in China's mining cities, Cities, 60, 75-83, 10.1016/j.cities.2016.07.009, 2017.

Hepburn, C., Duncan, S., and Papachristodoulou, A.: Behavioural Economics, Hyperbolic Discounting and Environmental Policy, Environmental & Resource Economics, 46, 189-206, 10.1007/s10640-010-9354-9, 2010.

Li, B., Sivapalan, M., and Xu, X.: An Urban Sociohydrologic Model for Exploration of Beijing's Water Sustainability Challenges and Solution Spaces, Water Resour. Res., 55, 5918-5940, 10.1029/2018wr023816, 2019.

Li, X. Y., Zhang, D. Y., Zhang, T., Ji, Q., and Lucey, B.: Awareness, energy consumption and pro-environmental choices of Chinese households, Journal of Cleaner Production, 279, 10.1016/j.jclepro.2020.123734, 2021.

Lian, X. B., Gong, Q., and Wang, L. F. S.: Consumer awareness and ex-ante versus ex-post environmental policies revisited, International Review of Economics & Finance, 55, 68-77, 10.1016/j.iref.2018.01.014, 2018.

Lin, J. Y., Wan, G., and Morgan, P. J.: Prospects for a re-acceleration of economic growth in the PRC, J. Comp. Econ., 44, 842-853, 10.1016/j.jce.2016.08.006, 2016.

Matrosov, E. S., Harou, J. J., and Loucks, D. P.: A computationally efficient open-source water resource system simulator - Application to London and the Thames Basin, Environmental Modelling & Software, 26, 1599-1610, 10.1016/j.envsoft.2011.07.013, 2011.

Rockson, G., Bennett, R., and Groenendijk, L.: Land administration for food security: A research synthesis, Land Use Policy, 32, 337-342, 10.1016/j.landusepol.2012.11.005, 2013.

Van Emmerik, T. H. M., Li, Z., Sivapalan, M., Pande, S., Kandasamy, J., Savenije, H. H. G., Chanan, A., and Vigneswaran, S.: Socio-hydrologic modeling to understand and mediate the competition for water between agriculture development and environmental health: Murrumbidgee River basin, Australia, Hydrology and Earth System Sciences, 18, 4239-4259, 10.5194/hess-18-4239-2014, 2014.

Xiong, Y. L., Wei, Y. P., Zhang, Z. Q., and Wei, J.: Evolution of China's water issues as framed in Chinese mainstream newspaper, Ambio, 45, 241-253, 10.1007/s13280-015-0716-y, 2016.

---

## Author Response (AR3)

**Cover letter**

Dear professor Murugesu Sivapalan:

We greatly appreciate you and the reviewers for taking time to review this manuscript and provide us with constructive and valuable comments. As reviewer 3 showed major concerns on the model conceptualization (particularly for socioeconomic projection) and the discordant scale of WEFS nexus, we have devoted ourselves to improve corresponding sections in Method and Discussion: (1) we applied Logistic model for socioeconomic projection in WEFS nexus as reviewer 3 recommended, and differences between the results of Malthusian model and Logistic model were discussed in the revised manuscript; (2) weight factors for water, energy, and food shortage awareness were added, the sensitivity analysis on which was conducted to investigate the contributions to environmental awareness from water, energy, and food systems with discordant scale. We believe the manuscript has been much improved. Our changes are marked in Blue in the revised manuscript. And our responses to the reviewers are detailed in this response-to-reviewers document submitted with the revised manuscript.

If you have any queries, please don't hesitate to contact me at the address below.

Looking forward to hearing from you.

Sincerely,

Dr. Dedi Liu

Corresponding author: Dedi Liu

Email: dediliu@whu.edu.cn

**Reviewer1**

The authors have made substantial revisions to the manuscript, and most of my concerns are addressed or clarified. I only have three very minor suggestions that I think the authors should further do to improve their paper.

Thank you for your positive feedback and valuable comments on our paper. We have carefully revised our manuscript according to the comments. Here are the responses to your comments:

1) It is misleading to simply write "scenario I, scenario II …" in the abstract. Readers will not understand what these scenarios are, without reading the whole paper. Therefore, the authors should replace "scenario I, scenario II …" with some specific description languages.

1. Response:

Thanks for your supportive comment. We have added specific description about scenarios in abstract in line 32~37.

"The annual average energy shortage rate thereby decreased from 17.16% to 5.80% by taking environmental awareness feedback, contributing to the sustainability of the WEFS nexus. Rational water resources allocation can ensure water supply through reservoir operation. The annual average water shortage rate decreased from 15.89% to 7.20% as water resources allocation was considered."

2) Some of my comments should be better clarified in the main text instead of just in the Response document. For example, the definition of environmental carrying capacity should be given in the manuscript Line 179.

2. Response:

Thanks for your supportive comment. We have checked the manuscript and added the definition in line 180~182. "environmental carrying capacities of socioeconomic variables (indicating the maximum socioeconomic size that can be carried by the system)"

3) After two rounds of revision, the current version of the manuscript is a bit too long.

Maybe it is better to move some of the contents, such as Table 1, to a supplementary document. And the language can be more concise throughout the manuscript.

3. Response:

Thanks for your supportive comment. We have added a supplementary document to simplify the manuscript, including tables for reservoir characteristics, and the calibrated parameters, and figures for sensitivity analysis of shortage awareness weight factors.

**Reviewer 3**

Though the authors have clearly invested time developing this manuscript, I still believe that it does not meet the standards of publication in this journal. Ultimately, as stated in the first review comments, for an abstract model which cannot be validated with real-world data, trustworthy insights require a well-reasoned model conceptualization. Major issues are still present with the model formulation and presentation, and the abundance of issues captured throughout the review process does not inspire confidence that the model will reach a reliable form.

Thank you very much for your critical but supportive comments, from which we have benefited a lot. We have tried hard to investigate the reliability of model conceptualization, and the discordant scale of WEFS nexus. Here are the responses to your comments:

**Lines 170-182 and equations 2-4:**

I maintain that a logistic model (sometimes called a Verhulst model) is more appropriate than the model proposed here. In a logistic model, proximity to the carrying capacity slows down growth (or exceeding the carrying capacity causes decay) rather than *time* slowing down growth. Why should time inherently slow growth? For instance, this oversight would seem to be the reason that population, GDP, etc. never resume growing after the year 2050.

1. Response:

Thanks for your supportive comment. We agree that logistic model is also popular in growth simulation for socioeconomic sector, as is claimed in the first round response. To quantitatively assess the differences between the Malthusian model and the Logistic model, **we applied Logistic model for WEFS nexus simulation**.

Model conceptualization for Logistic model was added in line 173~189.

[revised manuscript text omitted]

For Malthusian model, the socioeconomic variables evolution can be divided into four phases: expansion, contraction, recession, and recovery, as was discussed in the manuscript.

One of the major differences between results of Malthusian model and Logistic model is that state variable evolution in logistic model **fluctuates remarkably and performs periodicity**. However, it's worth noting that the socioeconomic expansion in the future will **slow down and tend to stabilization** (He et al., 2017; Lin et al., 2016), **the growth rate of which will thereby decrease as time goes**. Moreover, the economic development in the study area is also expected to gradually grow and then remains stable according to the Integrated Water Resources Planning of Hanjiang River Basin (CWRC, 2016). As the periodic fluctuation for WEFS nexus evolution through Logistic model is not consistent with the slowed socioeconomic expansion in foreseeable future and cannot fitly satisfy the planning in the study area, Logistic model is not adopted. Malthusian model can fitly meet the demand mentioned above, which is thereby applied for further analysis on WEFS nexus in our study.

**Equations 6-8:**

Equations 6 and 7 still don't provide what index is being summed over (only the
bounds of summation, 1 to sts-1, are provided), despite mention in the first two
rounds of comments. Also, it should be made clear that the variable WE, as
formulated in equation 6, is not the natural water inflow during the current time step,
but rather the *projected* natural inflow *for the rest of the simulation*.

More importantly, the reasoning behind equation 7 seems seriously flawed: water
shortage is defined as the *current* step water demand minus the reservoir inflow
from *all preceding steps* minus the natural inflow from *all steps* (preceding and
projected to follow), then divided by the remaining time steps. Why is current water
shortage not just current demand minus total current supply?

The right-most expression in equation 8 is very unclear – the numerator is summed
over two different time indices (ts and sts) yet only one time index is present within
the summation (sts). Also, how are the two expressions in equation 8 equivalent? One
sums shortages and demands overs all users/districts and the other over all time
steps…

2. Response:

Thanks for your supportive comment.

First, as is claimed in line 221~223, in IRAS model, each year is divided into *ts*
time steps, and each time step is further split into *sts* sub-time steps. Equation (9) and
(10) are used to estimate the water shortage of *j*th water user in *i*th operational zone
during *sts* sub-time step. Total water shortage in the study area is summed by equation
(11).

Second, we agree that the description of "WE" should be clearer. "extrapolated
natural water inflow" has been replaced by "projected natural water inflow for the rest
*Tsts-sts*+1 sub-time steps" in line 235~244.

Third, water inflow for water user comprises natural water inflow and reservoir
release. Specifically, reservoir release is directly related to water shortage from
corresponding water users. Directly taking current shortage by deducting total current
supply from current demand means that reservoir release in current sub-time step is
always related to water shortage in last sub-time step, while the information from natural water inflow is not used. As the temporal distribution of natural water inflow is uneven (i.e., natural water inflow is different in different sub-time step), water supply will be risked, and water resources allocation efficiency will be decreased. Equation (9) and (10) project natural water inflow in the rest sub-time steps based on natural water inflow in previous sub-time steps. Reservoir release in each sub-time step always considers natural water inflow in previous sub-time step, which can effectively improve water resources allocation efficiency.

Fourth, thanks for reminding us. We have corrected equation (11) by summing water user $j$ and operational zone $i$ in line 242.

$$WSR_t = \frac{\sum\limits_{i,j} WSR_{i,j}^t}{f_{red} * \sum\limits_{i,j} WD_{i,j}^t} = \frac{\sum\limits_{i,j} \sum\limits_{ts} \sum\limits_{sts} WS_{i,j}^{sts}}{f_{red} * \sum\limits_{i,j} \sum\limits_{ts} WD_{i,j}^{ts}} \tag{11}$$

**413-416:**

the authors have rephrased the statement and provided the years of data used to assess precipitation frequencies, but have not answered my previous question – how are precipitation frequencies assigned to years within the simulation? That is, how is the time series of future precipitation exceedances constructed? Also, why are all the precipitation exceedance frequencies used above 50% - this doesn't really capture wet years… (despite the text calling 50% "wet" and 75% "normal")

3. Response:

Thanks for your supportive comment.

First, as is claimed in line 413~415, historical discharge series from 1956 to 2016 is adopted, rather than future precipitation. The frequency series is determined by empirical frequency method.

Second, when the precipitation frequency is less than 50%, the year is considered as wet year, and agriculture water use quota with exceedance frequency 50% is adopted for agricultural water demand projection. It means the water demand is over-estimated. More water shortage can be exposed, which **further ensures the water supply safety**.

My prior comment regarding discordant scales where shortages are experienced has not been addressed. The author's response merely restates the information within the manuscript, describing what the scales are. I will try to rephrase my comment: water, energy, and food shortages are all aggregated into one "environmental awareness"

variable, however each shortage is experienced by different users with (in reality)

different connections to basin development dynamics. Energy users are defined as a sub-set of individuals/firms within the basin, being only those using energy to supply water. Water users are the full set of individuals/firms within the basin. Finally, food users are both within and outside of the basin. So, *shortages* are experienced discordantly by (1) a subset of those within the basin, (2) all those within the basin, and (3) those outside the basin; yet, these shortages are all aggregated into one

"environmental awareness". Therefore, via environmental awareness, energy shortage experienced by water suppliers directly constrains crop area; or, food shortage experienced by people living outside the basin directly constrains population growth within the basin. Even if the model formulation is not updated, some acknowledgement and discussion is necessary. Perhaps most concerning, between the first and second versions of the manuscript, the model was reformulated from simulating *all* energy consumption to just energy consumption *by water suppliers*.

However, none of the discussion of results was changed. The parameter values were updated and new values for results were pasted in, but none of the substance of discussion was updated. A drastic change in model scope occurred and yet there were no implications for the interpretation of results?

4. Response:

Thanks. We have greatly benefited from this valuable suggestion.

First, we have added the discussion on the impacts of discordant scale on WEFS

nexus in line 829~859.

As each shortage is experienced by different users with different connections to basin development dynamics (e.g., shortages from water, energy, and food are aggregated into environmental awareness, despite the food which is planned to be exported is considered in target food production), it's necessary to discuss the contributions to environmental awareness from water, energy, and food systems. Therefore, three weight factors were assigned to shortage awareness of water, energy, and food in equation (32) to adjust the over-estimated or under-estimated environmental awareness due to discordant scales. For instance, considering the target food production comprises inner food demand and exported food, the environmental awareness within the basin is over-estimated, and the weight factor for food shortage awareness can be set lower than 1.0 as a reduction factor to decrease current food shortage awareness. Sensitivity analysis was then conducted. Each weight factor was varied by given increment, while the other two weight factors were set to 1.0 as reference. The results are presented in Figure S1, S2, S3, and S4 in supplemental file.

$$\frac{dE}{dt} = wf_1 * \frac{dWA}{dt} + wf_2 * \frac{dEA}{dt} + wf_3 * \frac{dFA}{dt} \tag{32}$$

where $wf_1$, $wf_2$, and $wf_3$ are the weight factors for water, energy, and food shortage awareness, respectively.

[Figure]

**Figure S1. Trajectories of water demand with varied shortage awareness weight factors.**

[Figure]

**Figure S2. Trajectories of energy consumption with varied shortage awareness weight factors.**

[Figure]

**Figure S3. Trajectories of food production with varied shortage awareness weight factors.**

[Figure]

**Figure S4. Trajectories of environmental awareness with varied shortage awareness weight factors.**

WEFS nexus is sensitive to shortage awareness weight factors. Specifically, weight factors for water and energy shortage awareness can remarkably impact the recession phases of water demand, energy consumption, and food production. Lower weight factor can delay environmental awareness accumulation, and thus extend the contraction phase. However, more violent socioeconomic deterioration was also accompanied in the later recession phase, which consequently led the slightly smaller socioeconomic size in recovery phase. Weight factor for food shortage awareness can effectively dominate the whole evolution of water demand, and energy consumption. Lower weight factor indicated that smaller food shortage awareness can be accumulated. Feedback to increase crop area was thereby weakened. Both agriculture water demand and food production were decreased. As energy use quota for agricultural water supply is negligible, little response of energy consumption can be found.

Second, we redefined the energy consumption in the first round of response according to your first review comments. We focused on the energy consumption during the water supply process for socioeconomic water users to further investigate the energy co-benefits of water resources allocation schemes. Simultaneously, boundary conditions for energy system was also updated (e.g., planning energy availability, energy use quotas). Results indicated the phase dividing rule was still valid for the nexus co-evolution, despite the amount of energy consumption was indeed decreased significantly. However, environmental awareness feedback on socioeconomic factors was determined by shortage rate, rather than the amount of shortage. As there were small differences in energy shortage rate evolution process with redefined energy consumption, former discussion on the impacts of energy system on WEFS nexus was still valid.

Malthus, T.: An Essay on the Principle of Population, Penguin, Harmondsworth, England1798.